# FEDSAGD: FEDERATED LEARNING WITH STABLE AND ACCELERATED CLIENT GRADIENT DESCENT

## ABSTRACT

Federated Learning (FL) has become a promising paradigm for distributed machine learning. However, FL often suffers from degraded generalization performance due to the inconsistency between local and global optimization objectives and client-side overfitting. In this paper, we introduce global-update stability as an analytical tool to study generalization error and derive the stability bounds of mainstream FL optimization algorithms under non-convex settings. Our analyses reveal how the number of global update steps, data heterogeneity, and update rules influence their stability. We observe that momentum-based FL acceleration methods do not improve stability. To address this issue, we propose FedSAGD, a new FL algorithm that leverages the global momentum acceleration mechanism and a hybrid proximal term to enhance stability. This design ensures updates follow a globally consistent descent direction while retaining the benefits of acceleration. Theoretical analysis shows that FedSAGD achieves an advanced stability upper bound of $\mathcal{O}(1 - (1 - \Gamma)^T)(0 < \Gamma < 1)$ and attains a convergence rate of $\mathcal{O}(\frac{1}{\sqrt{sKT}})$ on non-i.i.d. datasets in the non-convex settings. Extensive experiments on real-world datasets demonstrate that FedSAGD significantly outperforms multiple baseline methods under standard FL settings, achieving faster convergence and state-of-the-art performance.

## 1 INTRODUCTION

Federated Learning (FL) enables distributed training without sharing raw data (McMahan et al., 2017), but it struggles with two core challenges in realistic cross-device scenarios: data heterogeneity and limited client participation rate. These factors induce local inconsistency (i.e., divergence between a client's local optimization objective and the global optimization goal) and local overfitting (i.e., client-side models overfit to local data distribution), leading to a phenomenon known as client drift (Sun et al., 2023a;b). This hinders both the convergence speed and the generalization performance of the global model (Karimireddy et al., 2020; Charles & Konečný, 2021; Malinovskiy et al., 2020). Consequently, the severe client drift caused by local overfitting to inconsistent solutions may lead to the degradation of the global model into merely an average of the client models (Sun et al., 2023a). This challenge becomes even more pronounced when the participation rate of clients is low during each communication round.

Several methods have attempted to address this challenge by incorporating momentum mechanisms (e.g., FedAdam (Reddi et al., 2021), FedAvgM (Hsu et al., 2019)), variance reduction techniques (e.g., SCAFFOLD (Karimireddy et al., 2020)) or regularization-based constraints (e.g., FedProx (Li et al., 2020), FedDyn (Acar et al., 2021)). Momentum-based and variance reduction methods mainly focus on reducing optimization error to accelerate convergence, while regularization-based methods aim to improve generalization by constraining the deviation between local and global models during local updates. However, these approaches often fail to significantly enhance generalization in highly non-IID scenarios with low client participation. We attribute this limitation to the absence of explicit attention to the stability of federated optimization, encompassing both its convergence behavior and its sensitivity to data heterogeneity and participation variability.

To study this, we adopt the framework of algorithmic stability (Hardt et al., 2016; Kuzborskij & Lampert, 2018; Bousquet & Elisseeff, 2002; Zhang et al., 2022), which provides a formal link between the sensitivity of an algorithm to data perturbations and its generalization ability. We

leverage global-update stability to investigate the sensitivity of FL algorithms to client replacement and establish a formal link to their generalization ability. We analyze the stability of common FL algorithms and show that momentum-based FL acceleration methods do not improve stability.

## 1.1 MAIN CONTRIBUTIONS

We propose FedSAGD, a new FL algorithm that leverages the global momentum acceleration mechanism and a hybrid proximal term to enhance stability. This design ensures that the updates always follow a globally consistent descent direction while controlling client drift, thereby improving the stability of global updates and retaining the benefits of acceleration. We prove that FedSAGD achieves a tighter stability bound of $\mathcal{O}(1 - (1 - \Gamma)^T)$ than existing momentum-based FL algorithms such as FedAdam (Reddi et al., 2021), FedAvgM (Hsu et al., 2019), and FedANAG (Zhang et al., 2025), which exhibit an upper bound of $\mathcal{O}(T)$, while also maintaining a same favorable convergence rate of $\mathcal{O}(\frac{1}{\sqrt{sKT}})$. We conduct extensive experiments on real-world vision and language datasets, including CIFAR-10/100, EMNIST, and Shakespeare. Experimental results show that the proposed FedSAGD consistently outperforms existing baselines in terms of test generalization and convergence speed. Our contributions are summarized below.

- We propose FedSAGD, a federated acceleration algorithm for non-convex FL with partial participation, combining client-side gradient-corrected momentum and a hybrid proximal term. Compared with previous momentum-based FL methods, FedSAGD achieves improved stability while maintaining fast convergence.

- We provide theoretical analysis of FedSAGD's stability and convergence under non-convex settings, showcasing its state-of-the-art performance.

- We empirically validate our theoretical findings by comparing FedSAGD with several established baselines. Extensive experiments confirm that our method consistently outperforms strong baselines in convergence speed, generalization, and stability.

## 2 RELATED WORK

**Generalization and stability in centralized learning.** Bousquet & Elisseeff (2002) introduced the concept of algorithmic stability and showed that it can be used to derive bounds on the generalization error. On-average stability is proposed by Shalev-Shwartz et al. (2010) and further studied by Kuzborskij & Lampert (2018). Stability-based generalization analysis was introduced into stochastic gradient-based methods (Hardt et al., 2016; Richards & Rebeschini, 2020). Sun et al. (2021); Zhu et al. (2022) extended the results to D-SGD and discussed the relationship between generalization performance and communication topology. Although stability analysis has proven to be an effective tool for characterizing the generalization performance of algorithms in centralized learning, due to issues such as data heterogeneity across clients and client dropout in federated learning, these conclusions cannot be directly extended to the federated setting.

**Generalization and stability in federated learning.** For the generalization bounds in FL, Hu et al. (2023) characterized the generalization error for both participating and non-participating clients in the training process. Liu et al. (2025) proposed the first algorithm-dependent generalization analysis with uniform stability for the typical personalized FL method. Sun et al. (2024) demonstrated that the generalization performance is closely related to the data heterogeneity and the convergence behaviors through on-average stability. Zhang et al. (2025) focus on the local-update stability of FL, i.e., how client variations affect the model after $K$ local update. To better model the fact that client participation in FL is partial and random, we propose global-update stability, which aims to investigate how variations in the participating client sets affect the difference in models trained by the FL algorithm.

**Federated optimization.** McMahan et al. (McMahan et al., 2017) first introduced FedAvg to address major challenges such as massively distributed clients and partial client participation. With the rapid development of FL, many optimization methods have been proposed to mitigate the divergence between clients and the global model caused by data heterogeneity. FedProx (Li et al., 2020) restricts the offset of local updates by adding a proximal term. FedDyn (Acar et al., 2021) dynamically modifies the device objective with a penalty term. SCAFFOLD (Karimireddy et al., 2020) employs

variance reduction techniques to correct the client drift in local updates. MimeLite (Karimireddy et al., 2021) uses a combination of control variates and server-level optimizer states (e.g., momentum) in each client update step. As classic optimization methods, momentum methods can be traced back to Polyak's heavy ball method (Polyak, 1964) and Nesterov's accelerated method (NAG) (Nesterov, 2013). With the flourishing of momentum mechanisms in machine learning research (Liu & Belkin, 2020; Liu et al., 2020; Assran & Rabbat, 2020), this motivates researchers to incorporate momentum methods into FL settings. FedAvgM (Hsu et al., 2019) and FedAdam (Reddi et al., 2021) apply momentum to the server side, with the latter utilizing an adaptive optimizer. FedCM (Xu et al., 2021) introduces global momentum into the local update. FedLNAG (Yu et al., 2019) employs NAG to accelerate each local iteration. FedACG (Kim et al., 2024) uses the global model with a lookahead gradient in the penalty term to regularize the local updates. FedANAG (Zhang et al., 2025) incorporates both the global and local analog NAG, while avoiding degrading stability and the communication overhead of uploading additional local momentum. However, these methods only enhance the empirical performance without improving stability relative to FedAvg. Therefore, we aim to design an algorithm with better stability that translates to improved empirical performance.

## 3 PROBLEM FORMULATION

### 3.1 PRELIMINARIES

Throughout this paper, we mainly consider the typical cross-device FL setting, which involves one server and a large number of clients. Let $m$ be the total number of clients, we use $[M]$ to denote the set $\{1, 2, \ldots, m\}$. We denote the set of active clients at round $t$ as $\mathcal{S}_t$, with $|\mathcal{S}_t| = s$. $T$ is the total communication rounds, $K$ represents the number of local updates per communication round. $x$ is the model parameter. $\langle \cdot, \cdot \rangle$ denotes the inner product for two vectors, and $\| \cdot \|$ the Euclidean norm.

We consider minimizing the following optimization problem of the form:

$$f(x) = \frac{1}{m} \sum_{i=1}^{m} F_i(x), \tag{1}$$

where the $F_i(x) = \mathbb{E}_{\xi_i \sim \mathcal{P}_i}[F_i(x, \xi_i)]$ is the local loss function of the client $i \in [M]$, $\xi_i$ represents the random data samples drawn from the distribution $\mathcal{P}_i$ and $n_i$ is the number of local samples. In FL settings, $\mathcal{P}_i$ may differ across the local clients, i.e., for clients $i$ and $j$, their data distribution might be significantly different. We state some standard assumptions (Reddi et al., 2021; Karimireddy et al., 2021; Zhang et al., 2025) as follows.

**Assumption 3.1.** *(L-smooth). $F_i$ is L-smooth for all client $i$, i.e., $\|\nabla F_i(x) - \nabla F_i(y)\| \leq L\|x - y\|$, $\forall x, y \in R^d$.*

**Assumption 3.2.** *(Unbiased estimator of local gradient with bounded variance). The local gradient estimate $g_{t,k}^i = \nabla F_i(x_{t,k}^i, \xi_i)$ with randomly sampled data $\xi_i$ is unbiased, i.e., $\mathbb{E}[g_{t,k}^i] = \nabla F_i(x_{t,k}^i)$. The variance of local gradients is bounded, i.e., $\mathbb{E}[\|g_{t,k}^i - \nabla F_i(x_{t,k}^i)\|^2] \leq \sigma_l^2$, $\forall i \in [M]$ and $\forall t$.*

**Assumption 3.3.** *(Bounded heterogeneity). The variance of local gradients at all clients is bounded, i.e., $\mathbb{E}[\|\nabla F_i(x_t) - \nabla f(x_t)\|^2] \leq \sigma_g^2$, $\forall i \in [M]$ and $\forall t$.*

**Assumption 3.4.** *(Z-Lipschitz) $F_i$ is Z-Lipschitz for all client $i$, i.e., $\|F_i(x) - F_i(y)\| \leq Z\|x - y\|$, $\forall x, y \in R^d$.*

Assumption 3.1 guarantees a Lipschitz continuity and Assumption 3.2 guarantees the stochastic gradient is bounded by zero mean and constant variance. Although in practical FL the samples are not i.i.d., they are still sampled from distributions that are not entirely unrelated (Li et al., 2020). Thus, it is reasonable to bound the dissimilarity between local functions in Assumption 3.3. Assumption 3.4 is widely used in the stability analysis (Hardt et al., 2016; Lei & Ying, 2020).

### 3.2 STABILITY BOUNDS ANALYSIS

We introduce the notion of algorithmic stability to provide an upper bound on the generalization error. In particular, we formally improve the on-average stability proposed by Sun et al. (2024), While Sun et al. (2024) characterizes sensitivity to single data point perturbations within a client,

this perspective does not fully capture the dominant source of variability in FL, i.e., partial client participation. In practice, client participation is partial and random, and models must generalize to unseen clients whose data distributions may differ significantly. To better model this scenario, we extend the stability analysis from sample-level perturbations to client-level perturbations, leading to the following definition.

**Definition 3.5.** *(global-update stability for FL). Let $\mathcal{A}(\mathcal{S})$ denote the model output by algorithm $\mathcal{A}$ on client set $\mathcal{S}$. A FL algorithm $\mathcal{A}$ is said to satisfy $\epsilon$-global update stability if given any two client sets $\mathcal{S}$ and $\mathcal{S}^{(i)}$, then for any $j \in [M]$*

$$\max_{i \in [M]} \mathbb{E}_{\mathcal{A}, \mathcal{S}, \xi_j}\big[|F_j(\mathcal{A}(\mathcal{S}), \xi_j) - F_j(\mathcal{A}(\mathcal{S}^{(i)}), \xi_j)|\big] \leq \epsilon,$$

*where two neighboring sets $\mathcal{S}$ and $\mathcal{S}^{(i)}$ only differ by one client $i$. Here, the expectation is also taken over the randomness of the active client set $\mathcal{S}$.*

Global-update stability means that any change in the participating FL clients will not cause a significant difference in the model trained by the FL algorithm in expectation. To better capture this partial participation setting, we adopt the concept of semi-empirical risk from Yuan et al. (2021) and Zhang et al. (2025) to define generalization error $\epsilon_{gen}$. The unparticipating expected risk as follows

$$R_u = \mathbb{E}_{i \sim \mathcal{C}}[\mathbb{E}_{\xi_i \sim \mathcal{P}_i}[F_i(\mathcal{A}(\mathcal{S}), \xi_i)]], \tag{2}$$

where $\mathcal{C}$ is a meta distribution from which the active clients are drawn. $R_u$ denotes the expected loss of the model over all clients in the meta-distribution. And the semi-empirical risk is defined by

$$R_e = \frac{1}{|\mathcal{S}|} \sum_{i \in \mathcal{S}}[\mathbb{E}_{\xi_i \sim \mathcal{P}_i}[F_i(\mathcal{A}(\mathcal{S}), \xi_i)]]. \tag{3}$$

$R_e$ represents the expected loss of the model over the clients participating in the training. Note that, to characterize the partial and random participation of clients, we adopt semi-empirical risk (Yuan et al., 2021; Zhang et al., 2025) rather than the traditional empirical risk. This replaces the empirical loss within each client with the expected loss over that client's local data distribution. Traditional generalization error measures the performance of a model on unseen data, while the generalization error we define here measures the performance of the model on the data of unseen clients. To better assess the generalization error for unparticipating clients, we define the generalization error of a model output $\mathcal{A}(\mathcal{S})$ is

$$\epsilon_{gen} := \mathbb{E}_{i \sim \mathcal{C}}[\mathbb{E}_{\xi_i \sim \mathcal{P}_i}[F_i(\mathcal{A}(\mathcal{S}), \xi_i)]] - \frac{1}{|\mathcal{S}|} \sum_{i \in \mathcal{S}}[\mathbb{E}_{\xi_i \sim \mathcal{P}_i}[F_i(\mathcal{A}(\mathcal{S}), \xi_i)]], \tag{4}$$

a smaller $\epsilon_{gen}$ implies the model $\mathcal{A}(\mathcal{S})$ has a better generalization performance on the data of unparticipating clients. The following theorem shows that generalization error of the model can be bounded by global-update stability. The proof is given in Appendix B.

**Theorem 3.6.** *Suppose that a FL algorithm $\mathcal{A}$ satisfies the $\epsilon$-global-update stability. Then,*

$$\mathbb{E}_{\mathcal{S}}\mathbb{E}_{\mathcal{A}}[\epsilon_{gen}] \leq \mathbb{E}_{\mathcal{S}}\mathbb{E}_{\mathcal{A}}[|R_u - R_e|] \leq \epsilon.$$

Theorem 3.6 establishes that if the algorithm exhibits high global update stability (i.e., a small stability upper bound), the generalization gap is guaranteed to be small. This indicates that the model not only fits the participating clients but also generalizes well to unseen clients. Therefore, improving stability becomes a key objective in designing effective FL algorithms. Based on Definition 3.5, we analyze the stability of existing methods. Following Hardt et al. (2016), we denote $x_T$ and $x'_T$ as the global models trained by two neighboring active client sets, and, under the Assumption 3.4, we relate the model difference $\mathbb{E}\|x_T - x'_T\|$ to stability. Supposing the same model initialization, we present in Table 1 the upper bounds of the resulting models for different FL algorithms. See Appendix C for the detailed proof.

As proposed in Sun et al. (2024), the model differences of the algorithms increase linearly with the heterogeneity of the client data distributions, are proportional to the sampling variance of the gradients, and also depend on the learning rates chosen during the local training process. Although Zhang et al. (2025) also analyzed the stability of FL algorithms, which they termed local-update stability, their study only considered the effect of client changes on the model difference after $K$

local training steps, essentially corresponding to stability at the local update level. In contrast, our work investigates global-update stability, namely, the impact of changes in client participation on the model difference of FL algorithms over $T$ communication rounds. Moreover, the conclusion on local momentum stability in Zhang et al. (2025) is limited to the setting of quadratic functions. As shown in Table 1, our stability results demonstrate that, in the non-convex setting, the global stability of existing momentum-based algorithms grows linearly with $T$. Furthermore, applying Nesterov momentum locally does not significantly increase the upper bound of model discrepancy, and relying solely on global momentum is also insufficient to improve the stability of FL algorithms.

Analysis reveals that FedProx and FedDyn exhibit better stability due to the incorporation of proximal terms in the local update process. Based on the discussion in Appendix C.2, after introducing the proximal term, the local-update stability of FedProx and FedDyn is reduced from the linear upper bound $\mathcal{O}(K)$ to a finite upper bound $\mathcal{O}(1 - (1 - \lambda\eta_l)^K)$. However, this cannot improve the global-update stability of the FL algorithm, which remains at $\mathcal{O}(T)$. To address this limitation, we propose a hybrid proximal term: $\frac{1}{2(\mu+\lambda)} \|(\mu + \lambda) x_{t,K} - \lambda x_t\|^2$. Then we can get the following conclusion

**Theorem 3.7.** *Suppose Assumption 3.1-3.4 hold and consider FedSAGD. Let $x_T$ and $x'_T$ be two model results obtained by neighboring active client-sets $\mathcal{S}$ and $\mathcal{S}'$, respectively. Under the assumption that $F_i$ is a non-convex and L-smooth function, then*

$$\mathbb{E}\|F_i(x_T) - F_i(x'_T)\| \leq Z\mathbb{E}\|x_T - x'_T\| \leq \frac{1 - (1 - \Gamma)^T}{\Gamma} \frac{2 + 2\beta}{\lambda + \mu}(\sigma_l^2 + 2\sigma_g^2 + 2Z^2)^{\frac{1}{2}} Z,$$

where $\Gamma := 1 - [\frac{\lambda}{\lambda+\mu} + \frac{\mu}{\lambda+\mu}(1 - \eta_l(\lambda + \mu))^K] \in (0, 1)$. By combining the proximal term with weight decay, this design transforms the accumulated historical gradients over $T$ rounds into an exponential moving average, thereby reducing the global stability of the algorithm from a linear upper bound $\mathcal{O}(T)$ to a finite upper bound $\mathcal{O}(1 - (1 - \Gamma)^T)$. The detailed proof can be found in Appendix C.11.

**Learning Rate and Convergence.** The stability analysis of various algorithms demonstrates that, under non-convex objective functions, stability is influenced by the learning rate. As the number of global iterations $T$ increases, the bias of the FL model in non-convex settings accumulates and becomes progressively larger. To guarantee convergence in such scenarios, the learning rate $\eta_l$ is typically set to be inversely proportional to the square root of the global rounds $T$, that is, $\eta_l = \mathcal{O}\left(\frac{1}{LK\sqrt{T}}\right)$. This choice also effectively compensates for the stability discrepancies in non-convex settings. Indeed, this convergence requirement has been validated in many existing studies (Yang et al., 2021; Yu et al., 2019; Zhang et al., 2025). Nevertheless, even with this compensation introduced by learning rate decay, the stability of existing algorithms remains at $\mathcal{O}(\sqrt{T})$, which is weaker than that of our proposed algorithm, achieving $\mathcal{O}(1 - (1 - \Gamma)^T)$.

Table 1: Summary of stability bounds of FL algorithms. All results are the ones under non-convex and Assumptions 3.1-3.4. We define $\Theta := (\sigma_l^2 + 2\sigma_g^2 + 2Z^2)^{\frac{1}{2}}$ for simplicity.

| Method | Non-convex | Historical gradient[a] | Proximal term | Communication[b] |
|---|---|---|---|---|
| FedAvg | $2\eta_l K\Theta T$ | - | - | $1\times$ |
| FedProx[c] | $2\lambda^{-1}\Theta[1 - (1 - \lambda\eta_l)^K]T$ | - | ✓ | $1\times$ |
| SCAFFOLD | $6\eta_l K\Theta T$ | Variance reduction | - | $2\times$ |
| FedDyn | $4\lambda^{-1}\Theta[1 - (1 - \lambda\eta_l)^K]T$ | - | ✓ | $1\times$ |
| FedLNAG | $2\eta_l K\Theta T$ | Client side-momentum | - | $2\times$ |
| FedAdam | $2\tau^{-1}\eta_l K\Theta T$ | Server side-momentum | - | $1\times$ |
| FedCM | $2\Theta T$ | Client side-momentum | - | $1 \sim 1.5\times$ |
| MimeLite | $2\eta_l K\Theta T$ | - | - | $2\times$ |
| FedANAG | $2(1 - \beta)^{-1}\eta_l K\Theta T$ | Client side-momentum | - | $1 \sim 1.5\times$ |
| FedACG | $(4 - 2\lambda)(1 - \lambda)^{-1}\eta_l K\Theta T$ | Server side-momentum | ✓ | $1\times$ |
| FedSAGD(ours) | $(\lambda + \mu)^{-1}\Gamma^{-1}(2 + 2\beta)\Theta[1 - (1 - \Gamma)^T]$ | Client side-momentum | ✓ | $1 \sim 1.5\times$ |

[a] This column refers to the utilization of historical gradients $g_{r,\tau}^i$ for $\tau < k$ and $r \leq t$ during the $k$-th local update.

[b] Here, communication refers to amount of data transmitted w.r.t. to FedAvg.

[c] $\lambda, \mu, \beta$ are the hyper-parameters of different algorithms.

# 4 METHODOLOGY

## 4.1 FEDSAGD ALGORITHM

We introduce FedSAGD, as shown in Algorithm 1, to improve stability and generalization while accelerating convergence, by coupling global momentum with client-side acceleration and a hybrid proximal term. Together, these components enable FedSAGD to reduce the negative effects of heterogeneity while achieving faster convergence with fewer communication rounds.

At the beginning of each round $t$, a subset of clients $\mathcal{S}_t$ are randomly selected to participate in the current training process (line 4). The global server will communicate the parameter $x_t$ and global momentum $v_t(v_0 = 0)$ to the active clients for local training (line 6). Each active client perform three stages: (1) computing the unbiased stochastic gradient $g_{t,k}^i = \nabla F_i(x_{t,k}^i, \xi_i)$ with local data $\xi_i$ (line 8); (2) computing the local momentum $m_{t,k}^i$ through combining $v_t$ and $g_{t,k}^i$ (line 9); (3) executing the gradient descent step with the local momentum $m_{t,k}^i$ and the hybrid proximal term (line 10). After $K$ iterations, local updates are completed, and the clients communicate the offset of the model parameters $x_{t,K}^i - x_{t,0}^i$ to the server for aggregation (line 12). On the global server, the global momentum $v_t$ will be updated as the weighted sum of the current average local offset $\Delta x_t$ and the historical global momentum (line 15). Then, the server performs a simple update to derive the global model $x_{t+1}$ (line 16).

---

**Algorithm 1** FedSAGD Algorithm Framework

---

1: **Input:** model parameters $x_0$, total communication rounds $T$, number of local iterations $K$, momentum coefficient $\beta$, proximal term $\lambda, \mu$, local learning rate $\eta_l$, global learning rate $\eta$.
2: **Output:** model parameters $x_T$.
3: **for** $t = 0, 1, 2, \ldots, T-1$ **do**
4:     select active clients-sets $\mathcal{S}_t$ at round t
5:     **for** client $i \in \mathcal{S}_t$ in parallel **do**
6:         communicate $x_t$ and $v_t$ to local client $i$ and initialize $x_{t,0}^i = x_t$
7:         **for** $k = 0, 1, 2, \ldots, K-1$ **do**
8:             compute unbiased stochastic gradient: $g_{t,k}^i = \nabla F_i(x_{t,k}^i, \xi_i)$
9:             update the momentum: $m_{t,k}^i = \beta v_t + g_{t,k}^i$
10:           update the gradient step: $x_{t,k+1}^i = x_{t,k}^i - \eta_l(m_{t,k}^i + ((\lambda + \mu)x_{t,k}^i - \lambda x_t))$
11:         **end for**
12:         $\Delta x_t^i = x_{t,K}^i - x_{t,0}^i$ and communicate $\Delta x_t^i$ to server
13:     **end for**
14:     $\Delta x_t = \frac{1}{s}\sum_{i \in \mathcal{S}_t} \Delta x_t^i$
15:     update global momentum: $v_{t+1} = \frac{\beta}{1+\beta}v_t - \frac{\Delta x_t}{(1+\beta)K\eta_l}$
16:     update global model: $x_{t+1} = x_t + \eta \Delta x_t$
17: **end for**

---

**Hybrid proximal term.** In traditional optimization problems, penalized proximal term are typically added to the objective function to improve the model's generalization ability and prevent overfitting (Krogh & Hertz, 1991; Ghojogh & Crowley, 2019). In FedProx (Li et al., 2020), the prox term is used to enhance local consistency in FL, while FedDyn (Acar et al., 2021) utilizes a dynamic regularization term to align the global and local optima. Through analysis, we find that the proximal term can improve the local-update stability $\mathcal{O}(1 - (1 - \eta_l \lambda)^K)$, but fail to enhance the global updates stability $\mathcal{O}(T)$. Therefore, we propose a hybrid proximal term: $\frac{1}{2(\mu+\lambda)}\|(\mu + \lambda)x_{t,K} - \lambda x_t\|^2$, which further improves the stability of global updates to $\mathcal{O}(1 - (1 - \Gamma)^T)$. As shown in the local update in Algorithm 1 (Line.10), for all $i, j \in [M], i \neq j$ we have:

$$\mathbb{E}\|x_{t,K}^i - x_{t,K}^j\| \leq (1 - \Gamma)\mathbb{E}\|x_t - x_t'\| + \zeta(2 + 2\beta)\Theta, \tag{5}$$

where $\zeta = \frac{1-(1-\eta_l(\lambda+\mu))^K}{\lambda+\mu}$. Proof details can be referred to in Appendix C.11. From equation 5, it can be observed that after incorporating the hybrid proximal term, the accumulated historical gradients

over $T$ rounds are transformed into an exponential moving average, which differs fundamentally from FedProx, as shown in Fig 6b. This also explains why our method achieves improved stability.

**Momentum update.** In FL settings, using local momentum can lead the update direction toward local optima rather than the global optimum, thus hindering global convergence. This issue becomes particularly pronounced in the presence of data heterogeneity. Therefore, we use global momentum locally to accelerate convergence while preserving stability. In fact, according to Line 8 of Algorithm 1 , the accumulated local update of FedSAGD can be written as:

$$x_{t,K}^i - x_{t,0}^i = -\eta_l \sum_{k=0}^{K-1} \alpha_k \hat{g}_{t,k}^i - \alpha\beta\eta_l v_t, \tag{6}$$

where $\alpha_k := (1 - \eta_l\lambda)^{K-1-k}$ and $\alpha := \sum_{k=0}^{K-1} \alpha_k$. $v_t$ denotes the global momentum. The detailed derivation is provided in Appendix D.2. FedSAGD achieves the same local-update stability upper bound as FedProx, but introduces an additional update term involving the global momentum $v_t$. In other words, at each local iteration, FedSAGD differs from FedProx by an extra term $\beta\frac{1-\alpha^K}{\lambda}v_t$. To maintain the same maximum model discrepancy as FedProx, the local update direction of FedSAGD is forced to be closer to the global optimum, as shown in Fig 6a. Consequently, FedSAGD is able to accelerate the training process while preserving high local-update stability.

Note that the overall communication cost of the proposed FedSAGD scheme consists of two parts: the downlink cost of broadcasting the global model $x_t$ and global momentum $v_t$ from the server, and the uplink cost of unicasting $s$ local model updates from the clients. In wireless communication, broadcasting a file to $s$ destinations is significantly more efficient than unicasting $s$ copies of the same file to each destination. Therefore, the communication overhead of FedSAGD is approximately $\frac{2+s}{1+s}$ times that of FedAvg, where $s$ typically represents a large number in cross-device FL (Kairouz et al., 2021).

## 4.2 CONVERGENCE ANALYSIS

We rigorously prove the convergence of FedSAGD for non-convex settings, assuming partial participation, i.e.$|\mathcal{S}_t| < m, \forall t \in [T]$. Our analysis can be directly extended to full participation. We provide a proof sketch in the following. For the full proof, please refer to Appendix D.

**Proof sketch.** To facilitate the handling of the momentum terms in the proof, we introduce an auxiliary sequence $z_t$. Denote $G_t^i = \sum_{k=0}^{K-1} \alpha_k \hat{g}_{t,k}^i$, $\alpha' = \frac{\alpha}{K}$ and $\gamma = 1 + \alpha'$. The form of $z_t$ is as follows:

$$z_t = \begin{cases} x_t, & t = 0; \\ \frac{1+\beta}{1-\alpha'\beta}x_t - \frac{\gamma\beta}{1-\alpha'\beta}x_{t-1} + \frac{\beta\eta\eta_l}{(1-\alpha'\beta)s}\sum_{i\in\mathcal{S}_{t-1}} G_{t-1}^i, & t \geq 1. \end{cases} \tag{7}$$

The $z_t$ update is:

$$z_{t+1} = z_t - \frac{\eta\eta_l}{(1-\alpha'\beta)s}\sum_{i\in\mathcal{S}_t} G_t^i.$$

From the expression of $z_t$, it can be seen that, as $t$ tends to positive infinity, $z_t$ and $x_t$ both converge to optimum $x^*$. Furthermore, after mapping $x_t$ to $z_t$, the entire update process will be simplified to an SGD-type method with the gradient $\hat{g}$. We subsequently introduce the convergence result for the FedSAGD algorithm.

**Theorem 4.1.** *Under Assumption 3.1-3.3 and loss function is non-convex, we define $D_0 := \mathbb{E}(\hat{f}(z_0) - \hat{f}(z_T))$. When the learning rate satisfies $\eta_l \leq \frac{1}{16KL}$ and*

$$\eta\eta_l \leq \frac{m(s-1)}{s(m-1)}min\left\{\frac{(1-\alpha'\beta)^2\sqrt{K}}{4\alpha\beta L\sqrt{3\alpha}}, \frac{(1-\alpha'\beta)\eta}{6\beta L\sqrt{2\alpha}}, \frac{1-\alpha'\beta}{2\alpha L}\right\}.$$

*Then the auxiliary sequence $z_t$ in equation equation 7 generated by executing the FedSAGD satisfies:*

$$\frac{1}{T}\sum_{t=0}^{T-1}\|\nabla\hat{f}(x_t)\| \leq \mathcal{O}\left(\frac{LD_0}{\sqrt{sKT}} + \frac{\sigma_l^2}{\sqrt{sKT}} + \frac{1}{T}\left(\frac{\sigma_l^2}{K^2} + K\sigma_l^2 + \frac{\sigma_l^2}{sK} + \frac{\sigma_g^2}{K}\right) + \Psi_l + \Psi_g\right)$$

*where*

$$\Psi_l = \left(\frac{1}{T} + \frac{K}{sT}\right)\frac{(m-s)}{m-1}\left(\frac{1}{T} + \frac{1}{sK^2T} + \frac{1}{\sqrt{sKT}}\right)\sigma_l^2,$$

$$\Psi_g = \left(\frac{K}{T} + K\right)\frac{(m-s)}{m-1}\left(\frac{1}{T} + \frac{1}{sK^2T} + \frac{1}{\sqrt{sKT}}\right)\sigma_g^2.$$

**Remark 4.2.** *To obtain the above results, it is also necessary to suppose that $\eta = \mathcal{O}(\sqrt{sK})$ and $\eta_l = \mathcal{O}(\frac{1}{LK\sqrt{T}})$. This is consistent with the assumptions on the convergence rate in Reddi et al. (2021); Yang et al. (2021). However, we do not make the assumption of bounded gradients as in Reddi et al. (2021). When $T$ is sufficiently large, the dominant term of convergence bound of algorithm is $\mathcal{O}(\frac{1}{\sqrt{mKT}})$ for full participation and $\mathcal{O}(\frac{\sqrt{K}}{\sqrt{sT}})$ for partial participation, which matches with the convergence rate of current FL algorithms (Yang et al., 2021; Zhang et al., 2025).*

**Remark 4.3.** *Though uniform sampling can effectively approximate the distribution of all working nodes in expectation, resulting in a structurally similar convergence rate between partial and full participation, the distribution deviation due to the fewer participating clients could destabilize the training process, particularly in highly non-i.i.d. cases (Yang et al., 2021). Our proposed algorithm reduces the upper bound of stability from $\mathcal{O}(T)$ to $\mathcal{O}(1 - (1 - \Gamma)^T)$ while still achieving a linear speedup $\mathcal{O}(\frac{\sqrt{K}}{\sqrt{sT}})$ with partial worker participation under heterogeneous datasets.*

## 5 EXPERIMENTAL ANALYSIS ON REAL WORLD DATASETS

### 5.1 SETUP

**Datasets and models.** We adopt standard federated learning benchmarks following the same train/test splits as prior works (McMahan et al., 2017; Li et al., 2020), including CIFAR-10 and CIFAR-100 (Krizhevsky, 2009), a subset of EMNIST (Cohen et al., 2017) referred to as EMNIST-L, and the Shakespeare dataset (Shakespeare, 1907). To comprehensively evaluate performance across diverse federated scenarios, the experiments adopt three settings similar to those in previous work (Zhang et al., 2025): Setting I (CIFAR-10 and CIFAR-100): 200 clients with a 2% participation rate per round; Setting II (EMNIST-L): 500 clients with a 1% participation rate; Setting III (Shakespeare): 100 clients with a 3% participation rate. To simulate data heterogeneity, we follow the approach of Hsu et al. (2019), sampling label distributions from a Dirichlet distribution with concentration parameter 0.3. For the IID setting, training data is randomly assigned to clients. In the non-IID setting, label distributions follow the sampled Dirichlet proportions. The model architectures include multinomial logistic regression, convolutional neural networks (CNNs), and recurrent neural networks (RNNs), standard ResNet-18 network, aligned with configurations used in previous studies (Sun et al., 2023a; Acar et al., 2021; Kim et al., 2024; Zhang et al., 2025). Due to space limitations, further details on datasets and models are provided in the Appendix A.

**Baseline methods.** We compare against a broad set of classical and efficient methods designed to address local inconsistency and client drift in federated learning, including FedAvg (McMahan et al., 2017), FedProx (Li et al., 2020), FedAdam (Reddi et al., 2021), SCAFFOLD (Karimireddy et al., 2020), FedCM (Xu et al., 2021), FedDyn (Acar et al., 2021), FedACG (Kim et al., 2024), FedANAG (Zhang et al., 2025), FedAvgM (Hsu et al., 2019), MimeLite (Karimireddy et al., 2021), and FedLNAG (Yu et al., 2019). FedAdam improves global updates via adaptive optimization. FedCM, FedACG, FedANAG, and FedAVG-M leverage global momentum to correct local updates. SCAFFOLD and FedDyn mitigate client heterogeneity by aligning local and global loss functions using control variates or dual variables, respectively. FedProx addresses local inconsistency through the addition of a proximal term. To ensure a fair evaluation under practical federated settings, we report the test accuracy after 2000 communication rounds across all experiments, following the baseline settings in Acar et al. (2021); Zhang et al. (2025). Further implementation details and hyperparameter configurations are provided in the Appendix A.

**Evaluation metrics.** Different algorithms are evaluated with two metrics: the convergence rate (i.e., number of rounds required to reach a target test accuracy) and the generalization performance (i.e., final global model performance on a validation set throughout training). These two metrics are widely adopted by many prior works such as Acar et al. (2021); Kim et al. (2024); Zhang et al. (2025).

## 5.2 MAIN RESULTS

We evaluate the aforementioned FL algorithms under a variety of challenging settings, including scenarios characterized by extreme data heterogeneity and very low client participation rates. The peak test accuracies reported in Table 1 are obtained by averaging results over five independent runs with different random seeds to ensure statistical reliability. We further investigate the sensitivity of performance to the hyperparameter $\beta$, $\mu$ and $\lambda$ under varying configurations. In addition, Appendix A.8 presents the variance of test accuracy across different seeds to complement the robustness analysis. Table 2 reports the test accuracies of FedSAGD and baseline methods on CIFAR-10, CIFAR-100, EMNIST, and Shakespeare under different settings, and the corresponding convergence curves are shown in Appendix A.9. We also verified the stability and empirical generalization error of most algorithms as shown in Appendix A.4. And we conducted experiments under feature-skewed settings, details can be found in Appendix A.7.

Table 2: Number of communication rounds required to achieve the preset target accuracy (presented in the first two lines on four datasets), and the top validation accuracy averaged over five random seeds for recent state-of-the-art methods (presented in the third line on four datasets), with the bold number representing the best result. Using CIFAR-10 as an example, the first two lines present the required communication rounds to reach 81% and 75% test accuracy, respectively. The third line shows the highest accuracy achieved by each method within 2000 rounds. '2000+' indicates that 81% accuracy was not reached even after 2000 rounds.

| Method | CIFAR-10 | | | CIFAR-100 | | | EMNIST | | | Shakespeare | | |
|---|---|---|---|---|---|---|---|---|---|---|---|---|
| | 75 | 81 | top(%) | 40 | 45 | top(%) | 78 | 80 | top(%) | 45 | 50 | top(%) |
| FedAvg | 572 | 2000+ | 78.64 | 2000+ | 2000+ | 40.79 | 102 | 271 | 81.01 | 176 | 516 | 51.61 |
| FedProx | 572 | 2000+ | 78.88 | 1207 | 2000+ | 41.74 | 46 | 154 | 81.83 | 194 | 522 | 51.54 |
| FedDyn | 354 | 2000+ | 80.14 | 1512 | 2000+ | 41.89 | 65 | 146 | 81.02 | 231 | 489 | 51.77 |
| SCAFFOLD | 364 | 1535 | 81.34 | 1308 | 2000+ | 42.24 | 60 | 141 | 81.55 | 180 | 465 | 52.04 |
| MimeLite | 439 | 1959 | 81.02 | 578 | 1050 | 47.59 | 57 | 129 | 81.91 | 177 | 477 | 51.71 |
| FedAvgM | 489 | 1522 | 81.86 | 714 | 1197 | 48.31 | 50 | 151 | 81.92 | 84 | 313 | 52.32 |
| FedAdam | 1295 | 2000+ | 78.29 | 1458 | 1877 | 44.51 | 61 | 162 | 81.58 | 99 | 359 | 52.30 |
| FedCM | 530 | 1891 | 81.09 | 1265 | 1718 | 45.02 | 277 | 1101 | 80.16 | 2000+ | 2000+ | 39.14 |
| FedLNAG | 497 | 2000+ | 79.60 | 1031 | 1602 | 42.31 | 41 | 196 | 81.92 | 93 | 296 | 52.23 |
| FedANAG | 338 | 882 | 83.51 | 673 | 1701 | 45.32 | 38 | 181 | 82.01 | 68 | 183 | 52.46 |
| FedACG | 338 | 891 | 83.52 | 582 | 1139 | 50.59 | 36 | 155 | 82.02 | 72 | 214 | 52.52 |
| FedSAGD(ours) | **287** | **851** | **84.05** | **536** | **852** | **54.53** | **30** | **131** | **82.24** | **56** | **147** | **53.03** |

Table 3: Accuracy (%) vs. Non-iid-ness, With PR = 1%.

| **Dirichlet** | 0.3 | 0.6 | 0.8 | iid |
|---|---|---|---|---|
| FedSAGD | **82.86** | **84.08** | **84.46** | **85.36** |
| FedACG | 81.52 | 82.34 | 83.12 | 85.03 |
| FedANAG | 82.37 | 83.11 | 84.27 | 85.13 |
| SCAFFOLD | 80.88 | 81.88 | 82.31 | 83.17 |
| FedDyn | 78.45 | 81.95 | 83.01 | 83.10 |
| FedLNAG | 79.05 | 80.53 | 80.93 | 81.99 |
| FedAvgM | 81.34 | 82.15 | 83.12 | 84.70 |
| FedAdam | 78.81 | 81.10 | 82.52 | 84.5 |
| FedCM | 79.44 | 80.46 | 81.62 | 82.07 |

Table 4: Accuracy (%) vs. PR, With Dirichlet = 0.3.

| **PR(%)** | 1 | 2 | 5 | 10 |
|---|---|---|---|---|
| FedSAGD | **82.86** | **84.05** | **83.49** | **83.01** |
| FedACG | 81.52 | 83.52 | 83.14 | 82.63 |
| FedANAG | 82.37 | 83.51 | 83.04 | 82.50 |
| SCAFFOLD | 80.88 | 81.34 | 81.97 | 82.12 |
| FedDyn | 78.45 | 80.14 | 80.45 | 82.46 |
| FedLNAG | 79.05 | 79.60 | 80.10 | 79.54 |
| FedAvgM | 81.34 | 81.86 | 82.71 | 82.08 |
| FedAdam | 78.81 | 78.29 | 81.77 | 81.73 |
| FedCM | 79.44 | 81.09 | 82.00 | 81.89 |

As shown in Table 2, FedSAGD consistently and significantly improves both convergence speed and accuracy across almost all scenarios. This remarkable performance is attributed to the global momentum at the client side and the hybrid proximal term. This momentum-based acceleration mechanism not only leverages global momentum to accelerate both local and global training, but also provides a globally consistent update direction, thereby enhancing global convergence. Furthermore, aligning local updates with the trajectory of global gradients improves inter-client model consistency, resulting in increased stability. In contrast, other momentum-based methods such as FedLNAG can accelerate local convergence toward local optima, but due to the misalignment between local and global optima, the performance of the aggregated global model remains suboptimal. On the other

hand, approaches that rely solely on global momentum such as FedCM, FedACG, and FedANAG may improve global convergence, but have limited ability to improve inter-client consistency, which in turn leads to degraded generalization performance. This observation is also theoretically supported by our stability analysis. In addition, compared with methods that rely solely on global momentum and the current gradient, FedSAGD leverages the hybrid proximal term to correct the gradient, thereby further improving the consistency of client update directions. This process reduces model discrepancies across clients and further enhances training efficiency.

Tables 3 and 4 demonstrate that FedSAGD exhibits the highest robustness under conditions of severe data heterogeneity and low client participation rates. The results under milder heterogeneity settings are provided in Appendix A.6. To further evaluate performance, we examine the peak validation accuracy achieved by each method on the CIFAR-10 dataset under varying degrees of non-i.i.d. distributions and client participation rates (PR). Across both i.i.d. and non-i.i.d. scenarios, FedSAGD consistently outperforms all baselines under various participation settings in terms of generalization. Moreover, increases in data heterogeneity and decreases in client participation exert only a minimal adverse impact on the performance of FedSAGD.

## 6 CONCLUSION

We propose FedSAGD, a novel and practical federated learning algorithm designed to address the issue of unstable client updates in existing momentum-based and variance-reduction-based methods. To this end, we introduce a consistency-based stability metric and show that current acceleration schemes do not yield notable advantages in terms of stability. FedSAGD employs a global momentum acceleration mechanism to guide client updates and introduces a hybrid proximal term to further enhance the consistency among client models. We provide rigorous theoretical analysis demonstrating that FedSAGD achieves the optimal $\mathcal{O}(1 - (1 - \Gamma)^T)$ stability and $\mathcal{O}(\frac{1}{\sqrt{sKT}})$ convergence rates in non-convex settings. Extensive experiments conducted on multiple datasets validate the superiority of FedSAGD in both training efficiency and final performance, which is highly consistent with our theoretical findings.

## ETHICS STATEMENT

Our research conforms to the ICLR Code of Ethics in every respect. We have thoroughly reviewed the guidelines and ensured that our work adheres to the ethical standards set forth.

## REPRODUCIBILITY STATEMENT

We have made every effort to ensure the reproducibility of our work. The details of the model architecture, training process, and hyperparameters are provided in Section 5.1 and Appendix A.3. A complete description of the experimental setup, including datasets, models, and evaluation metrics, is included in Section 5.1 and Appendix A.1. Algorithmic details and proofs of theoretical claims are presented in Appendix B, Appendix C and Appendix D.

This study adheres to the principles of open science, emphasizing transparency and accessibility in research. The source code accompanying this work is publicly available on Anonymous GitHub at `https://anonymous.4open.science/r/Fed-SAGD-808E`. The repository provides artifact instructions, dependencies, core codes (*e.g.*, data, models, evaluation), in compliance with ICLR's reproducibility policy.

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

# A   EXPERIMENT DETAILS

## A.1   REAL DATA

We evaluate our models on four benchmark datasets: EMNIST-L, CIFAR-10, CIFAR-100, and Shakespeare. EMNIST-L is a subset of the EMNIST dataset, comprising the first 10 characters from the alphabet split, following prior work such as FedProx (Li et al., 2020) and FedDyn (Acar et al., 2021). For image classification tasks, EMNIST-L, CIFAR-10, and CIFAR-100 are used, with image dimensions of $1 \times 28 \times 28$ for EMNIST-L and $3 \times 32 \times 32$ for both CIFAR-10 and CIFAR-100. These datasets contain 10, 10, and 100 classes, respectively. The Shakespeare dataset is used for next-character prediction, with 80 distinct characters as inputs and the subsequent character as the label, resulting in 80 output classes. We adopt the standard training/testing splits for all datasets. A summary of the number of training and test samples for each benchmark is provided in Table 7.

To generate Non-IID data partitions for EMNIST, CIFAR-10, and CIFAR-100, we allocate training samples to clients based on class labels. Specifically, following prior works (Yurochkin et al., 2019), we use the Dirichlet distribution to create federated heterogeneous datasets by sampling a class-probability vector for each client, where each vector is drawn from a Dirichlet distribution with a concentration parameter that controls the degree of data heterogeneity. For each client, labels are sampled according to this probability vector, and corresponding images are drawn without replacement. This process is repeated until all data points are allocated. As a result, each client's label distribution follows the Dirichlet distribution, with the concentration parameter governing the level of statistical heterogeneity across devices. For instance, when the Dirichlet factor is set to 0.3, approximately 80% of each client's data is concentrated in 3–4 dominant classes. In the IID setting, data is randomly shuffled and evenly distributed across clients.

For the Shakespeare dataset, we adopt the LEAF framework (Caldas et al., 2018) to generate Non-IID data, capping each client's dataset to 2000 samples, consistent with FedDyn. In this setting, data is naturally partitioned by role, where each device corresponds to a character in the script and holds all lines spoken by that character. In the IID setting, all lines are merged and then randomly distributed across clients.

## A.2   MODELS

For CIFAR-10 dataset, we adopt a CNN architecture similar to those used in FedAvg and FedDyn, which consists of two convolutional layers with 64 filters of size $5 \times 5$, followed by two $2 \times 2$ max-pooling layers, two fully connected layers with 384 and 192 neurons respectively, and a final softmax output layer (see Table 5 for complete model details). This CNN model is identical to the one used in FedAvg, except that batch normalization layers are excluded. For the CIFAR-100 dataset, we use standard ResNet-18-GN model (He et al., 2016) backbone with the 7×7 filter size in the first convolution layer as implemented in the previous works, e.g. for (Karimireddy et al., 2020; Acar et al., 2021). We follow the (Hsieh et al., 2020; Sun et al., 2023a) to replace the batch normalization layer with group normalization layer (Wu & He, 2018), which can be aggregated directly by averaging. These are all common setups in many previous works. For the EMNIST-L dataset, we use a multi-class logistic regression model with a cross-entropy loss function. For the next-character prediction task on the Shakespeare dataset, we follow the configurations used in FedProx and FedDyn, employing an RNN model that first embeds the input character sequence into an 8-dimensional space, then passes it through a two-layer LSTM with 100 units, and finally outputs predictions via a softmax layer (see Table 6 for full specifications).

## A.3   HYPER-PARAMETERS

In our experiments, all algorithms are implemented using PyTorch 2.0.0 with CUDA 11.8 on a GEFORCE RTX 4090 GPU. We consider different hyperparameter configurations for various setups and datasets. For all experiments, we fix the batch size as 48 for EMNIST-L, 50 for CIFAR-10 and CIFAR-100, and 100 for Shakespeare. For each dataset, we compare the performance of different methods under various hyperparameter configurations. The client learning rate $\eta_l$ and the learning rate decay factor are individually tuned via grid search.

Table 5: CNN Architecture for CIFAR-10

| Layer Type | Size |
|---|---|
| Convolution + ReLU | $5 \times 5 \times 64$ |
| Max Pooling | $2 \times 2$ |
| Convolution + ReLU | $5 \times 5 \times 64$ |
| Max Pooling | $2 \times 2$ |
| Fully Connected + ReLU | $1600 \times 384$ |
| Fully Connected + ReLU | $384 \times 192$ |
| Fully Connected | $192 \times 10$ & $192 \times 100$ |

Table 6: Shakespeare model architecture

| Layer Type | Size |
|---|---|
| Embedding | $(80, 8)$ |
| LSTM | $(80, 100)$ |
| LSTM | $(80, 100)$ |
| Fully Connected | $(100, 80)$ |

Table 7: Train and test splits

| Dataset | No. Train | No. Test | No. clients | sampling rate | No. Train per client | Batch size | Rounds |
|---|---|---|---|---|---|---|---|
| EMNIST-L | 48000 | 8000 | 500 | 1% | 96 | 48 | 2000 |
| CIFAR-10 | 50000 | 10000 | 200 | 2% | 250 | 50 | 2000 |
| CIFAR-100 | 50000 | 10000 | 200 | 2% | 250 | 50 | 2000 |
| Shakespeare | 200000 | 40000 | 100 | 3% | 2000 | 100 | 1000 |

**EMNIST-L.** As for the 500 devices, balanced data, full participation setup, hyperparameters are searched for all algorithms in all IID and Dirichlet settings for a fixed 100 communication rounds. The search space consists of learning rates in $[0.1, 0.01, 0.001]$, $\lambda$s in $[1, 0.01, 0.001]$ and $\alpha$s in $[0.001, 0.01, 0.02, 0.1]$, epochs is 20. The learning rate decay is selected from the range of $[0.995, 0.998, 0.999, 1.0]$. The selected configuration for FedAvg is 0.1 learning rate; for FedProx is 0.001 learning rate and 0.001 $\mu$; for FedDyn is 0.001 learning rate and $0.01\alpha$; and for SCAFFOLD is 0.001 learning rate; and for FedSAGD, FedANAG, FedACG, FedLNAG, FedAvgM are 0.001 learning rate and $0.9\beta$; and for FedCM is 0.001 learning rate and $0.1\beta$; and for MimeLite is 0.001 learning rate and $0.2\beta$; and for Fedadam is 0.001 learning rate and 0.1 glabal learning rate for all IID and Dirichlet settings. These configurations are fixed and their performances are obtained for 500 communication rounds.

**CIFAR-10**. We used similar hyperparameters as in EMNIST-L dataset. The configuration includes: 0.1 for learning rate, 5 for epochs. The learning rate decay is selected from the range of $[0.995, 0.998, 0.999, 1.0]$. The $\alpha$ value is selected from the range of $[0.001, 0.01, 0.1]$ for FedDyn. The $\mu$ value is selected from the range of $[0.1, 0.01, 0.001, 0.0001]$.

**CIFAR-100.** The same hyperparameters are applied to the CIFAR-100 experiments with 200 devices including: 0.1 for learning rate, 5 for epochs. The learning rate decay is selected from the range of $[0.995, 0.998, 0.999, 1.0]$. The $\alpha$ value is selected from the range of $[0.001, 0.01, 0.1]$ for FedDyn. The $\mu$ value is selected from the range of $[0.1, 0.01, 0.001, 0.0001]$.

**Shakespeare.** As for 100 devices, balanced data, full participation setup, the hyperparameters are searched with all combinations of learning rate in 1, epochs in $[1, 5]$, $\lambda$s in $[0.01, 0.001]$ and $\alpha$s in $[0.001, 0.009, 0.01, 0.015]$. The learning rate decay is selected from the range of $[0.995, 0.998, 0.999, 1.0]$. The learning rate decay is selected from the range of $[0.995, 0.998, 0.999, 1.0]$. The selected configuration for FedSAGD, FedANAG, FedACG, FedL-NAG, FedAvgM are $0.9\beta$; and for FedCM is $0.1\beta$; and for MimeLite is $0.2\beta$; The $\alpha$ value is selected from the range of $[0.001, 0.01, 0.1]$ for FedDyn. The $\mu$ value is selected from the range of $[0.1, 0.01, 0.001, 0.0001]$.

### A.4 VALIDATION OF STABILITY AND EMPIRICAL GENERALIZATION ERROR

As stated in Section 3.2, since we can establish a connection between the difference in loss functions and the difference in models, we measure stability by computing the maximum discrepancy among client models after two times of FL training. We evaluate the global update stability by measuring the model discrepancy for models including multiclass logistic classification and CNN. We use EMNIST on multi-class logistic classification model with 500 clients and 1% client sampling, and CIFAR10/CIFAR100 on CNN with 200 clients and 2% client sampling. Empirical evaluation on

logistic and CNN models in Fig 1 and Fig 2. The model discrepancy here is measured using the $L_2$ norm. For the empirical generalization error, we measure it using the absolute difference between the training and test losses, as well as the absolute difference between the training and test accuracies. We evaluate this metric on CIFAR-10 using a CNN model with 200 clients and a 2% client sampling rate. The results are shown in Fig. 3.

As show in Fig 1, the experimental results have verified local-update stability of our method. Existing momentum-based algorithms, due to data heterogeneity, may actually impair local-update stability when local momentum acceleration is used (e.g., FedLNAG); using global momentum (e.g., FedAdam) performs slightly better, but it still does not improve compared to FedAvg. Besides, while FedDyn's stability is enhanced compared to FedAvg, its local-update stability is inferior to that of FedSAGD and FedProx due to low participation rates.

As show in Fig 2, the experimental results have verified global-update stability of our method. The FedLNAG algorithm exhibits significantly poorer stability than other methods across all tasks. The stability of other momentum-based methods is largely comparable to that of FedAvg. In terms of global update stability, the improvements achieved by FedDyn and FedProx over FedAvg and related methods are rather limited, and their performance remains far inferior to that of FedSAGD. In contrast, FedSAGD demonstrates substantially better global stability than all other methods, further validating the effectiveness of our approach. Therefore, our algorithm can still achieve optimal stability while accelerating training.

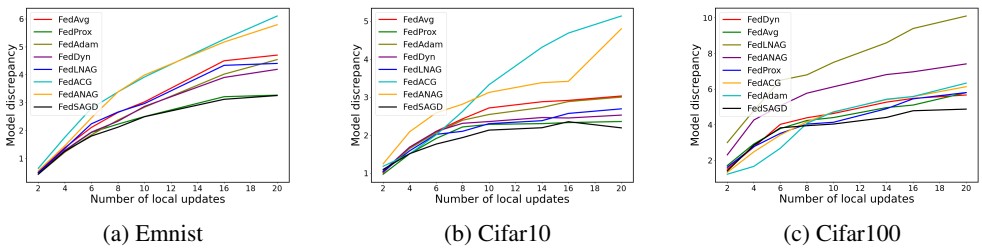

|         |         |         |
|:-------:|:-------:|:-------:|
| (a) Emnist | (b) Cifar10 | (c) Cifar100 |

Figure 1: Local model discrepancy (local-update stability) vs. index of local updates $K$.

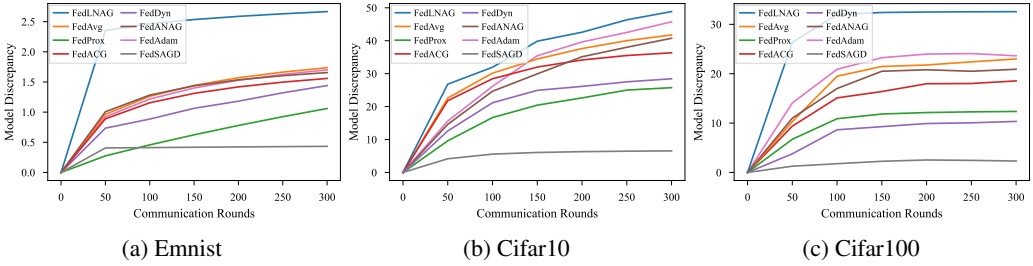

|         |         |         |
|:-------:|:-------:|:-------:|
| (a) Emnist | (b) Cifar10 | (c) Cifar100 |

Figure 2: Global model discrepancy (global-update stability) vs. index of communication rounds $T$.

As show in Fig. 3, the results are highly consistent with the stability results, further confirming the reliability of our theoretical findings.

## A.5 HYPERPARAMETER SENSITIVITY

We evaluate the performance of our algorithm under different settings of $\beta$, $\beta_0$ and $\lambda$ on the CIFAR-10 dataset, using 500 communication rounds and 10% client participation among 100 clients.

**Momentum coefficent $\beta$ for global momentum.** In the simple analysis, $\beta$ can be selected as a proper value which has no impact on the convergence complexity. Table 8 shows that performance of FedSAGD remains stable across $\beta$ values ranging from 0.1 to 0.99. Despite slight fluctuations, the accuracy stays consistently high, peaking at $\beta = 0.9$.

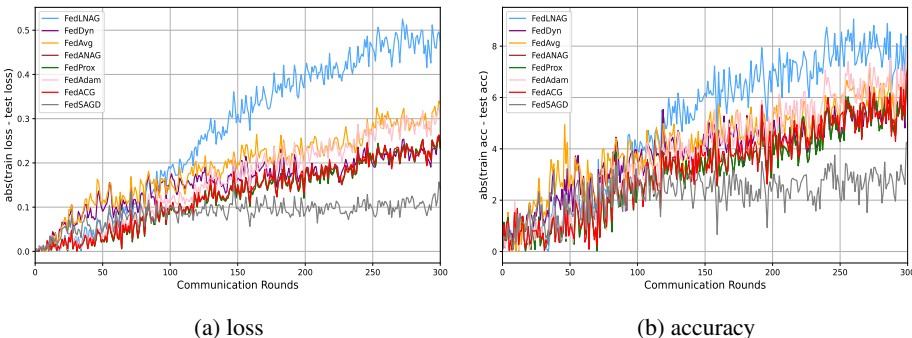

(a) loss       (b) accuracy

Figure 3: Empirical generalization error vs. communication rounds $T$.

Table 8: Performance of different $\beta$ with $\mu = 0.001$ and $\lambda = 0.01$.

| $\beta$ | 0.99 | 0.95 | 0.9 | 0.8 | 0.7 | 0.6 | 0.1 |
|---|---|---|---|---|---|---|---|
| Acc. | 80.27 | 81.14 | 81.34 | 81.02 | 80.81 | 80.63 | 78.57 |

**Coefficent $\mu$ for the weight of the proximal term.** As shown in Table 9, the performance of FedSAGD remains stable across $\mu$ values ranging from 0.00001 to 0.1. Despite slight fluctuations, the accuracy stays consistently high, peaking at $\mu = 0.001$.

Table 9: Performance of different $\mu$ with $\beta = 0.9$ and $\lambda = 0.01$.

| $\mu$ | 0.1 | 0.01 | 0.005 | 0.001 | 0.0005 | 0.0001 | 0.00001 |
|---|---|---|---|---|---|---|---|
| Acc. | 77.22 | 80.09 | 80.93 | 81.34 | 81.14 | 80.08 | 78.72 |

**Coefficent $\lambda$ for the weight of the proximal term.** As shown in Table 10, the performance of FedSAGD remains stable across $\lambda$ values ranging from 0.0001 to 1. Despite slight fluctuations, the accuracy stays consistently high, peaking at $\lambda = 0.01$.

Table 10: Performance of different $\lambda$ with $\beta = 0.9$ and $\mu = 0.001$.

| $\lambda$ | 1 | 0.1 | 0.05 | 0.01 | 0.005 | 0.001 | 0.0001 |
|---|---|---|---|---|---|---|---|
| Acc. | 76.07 | 80.68 | 80.96 | 81.34 | 81.26 | 81.19 | 81.06 |

The performance of FedSAGD remains stable across a wide range of values for the parameters $\beta$, $\mu$, and $\lambda$. Specifically, for $\beta$ values ranging from 0.6 to 0.95, FedSAGD maintains consistently high accuracy with slight fluctuations, peaking at $\beta = 0.9$. For $\mu$ values ranging from 0.1 to 0.0001, the accuracy remains high and stable, with the highest performance observed at $\mu = 0.001$. Also, for $\lambda$ values ranging from 0.1 to 0.001, FedSAGD shows stable performance with consistent accuracy, reaching its peak at $\lambda = 0.01$. These results demonstrate that FedSAGD is robust to variations in these hyperparameters, even under the condition that of high heterogeneity and low participation rates. Furthermore, we also conducted a hyperparameter sensitivity analysis under a more extreme heterogeneity setting with $\alpha = 0.1$. The conclusions are consistent with the above.

Table 11: Performance of different $\beta$ with $\mu = 0.001, \lambda = 0.01$ and $\alpha = 0.1$.

| $\beta$ | 0.99 | 0.95 | 0.9 | 0.8 | 0.7 | 0.6 | 0.1 |
|---------|------|------|-----|-----|-----|-----|-----|
| Acc. | 80.27 | 81.14 | 81.34 | 81.02 | 80.81 | 80.63 | 78.57 |

Table 12: Performance of different $\mu$ with $\beta = 0.9$, $\lambda = 0.01$ and $\alpha = 0.1$.

| $\mu$ | 0.1 | 0.01 | 0.005 | 0.001 | 0.0005 | 0.0001 | 0.00001 |
|-------|-----|------|-------|-------|--------|--------|---------|
| Acc. | 77.22 | 80.09 | 80.93 | 81.34 | 81.14 | 80.08 | 78.72 |

Table 13: Performance of different $\lambda$ with $\beta = 0.9$, $\mu = 0.001$ and $\alpha = 0.1$.

| $\lambda$ | 1 | 0.1 | 0.05 | 0.01 | 0.005 | 0.001 | 0.0001 |
|-----------|---|-----|------|------|-------|-------|--------|
| Acc. | 76.07 | 80.68 | 80.96 | 81.34 | 81.26 | 81.19 | 81.06 |

## A.6 DIFFERENT HETEROGENEITY

To further examine the behavior of all algorithms under more extreme levels of data heterogeneity, we additionally evaluate the setting with smaller heterogeneity parameters, specifically $\alpha = 0.1$ and $\alpha = 0.2$, while keeping all other experimental configurations identical to those in Table 3. The results in Table 14 show that FedSAGD exhibits the highest robustness under conditions of severe data heterogeneity.

Table 14: Performance under more extreme heterogeneity.

| $\alpha$ | FedSAGD | FedACG | FedANAG | SCAFFOLD | FedDyn | FedLNAG | FedAvgM | FedAdam | FedCM |
|----------|---------|--------|---------|----------|--------|---------|---------|---------|-------|
| 0.1 | **81.01** | 76.64 | 80.56 | 79.18 | 76.65 | 76.52 | 78.7 | 74.3 | 73.39 |
| 0.2 | **82.37** | 80.97 | 81.86 | 80.35 | 77.51 | 78.05 | 80.09 | 78.31 | 78.82 |

Table 15: Performance comparison under feature skew

| Method | Acc. (%, ↑) 250R | Acc. (%, ↑) 500R | Rounds (↓) 75% | Rounds (↓) 80% |
|--------|------------------|------------------|----------------|----------------|
| FedSAGD(ours) | **79.82** | **81.14** | **80** | **267** |
| FedACG | 79.5 | 80.62 | 82 | 285 |
| FedANAG | 79.37 | 80.54 | 86 | 288 |
| FedAdam | 76.79 | 79.01 | 180 | 500+ |
| SCAFFOLD | 78.58 | 79.86 | 175 | 500+ |
| FedLNAG | 74.46 | 78.31 | 346 | 500+ |
| FedAvg | 77.29 | 78.74 | 192 | 500+ |
| MimeLite | 78.78 | 80.18 | 179 | 372 |
| FedAvgM | 78.73 | 80.07 | 166 | 467 |
| FedCM | 75.58 | 79.05 | 226 | 500+ |
| FedProx | 77.79 | 79.33 | 187 | 500+ |
| Alg12 | 77.31 | 79.7 | 172 | 487 |

## A.7  FEATURE SKEW

We additionally evaluate our method under feature-skew settings, following the experimental setup in Kim et al. (2024). We conduct the evaluation on the real-world LEAF dataset FEMNIST, which naturally contains feature-skewed client distributions. In this experiment, we set the number of clients to 2000, use user-level partitioning, randomly sample 5 clients per communication round, and adopt a two-layer CNN architecture as in Caldas et al. (2018). The experimental results in Table 15 demonstrate that our method also outperforms the baselines under feature-skew settings, highlighting its robustness to feature-skew heterogeneity.

## A.8  VARIANCE MEASUREMENTS OF TOP ACCURACY WITH DIFFERENT SEEDS

Table 16: The variance measurement of top validation accuracy that can be achieved, with 5 random seeds.

| Dataset | FedSAGD | FedANAG | FedACG | FedLNAG | FedCM | FedAdam |
|---|---|---|---|---|---|---|
| EMNIST | **82.24** $\pm$ 0.07 | 82.01 $\pm$ 0.18 | 82.02 $\pm$ 0.16 | 81.92 $\pm$ 0.35 | 80.16 $\pm$ 0.82 | 81.58 $\pm$ 0.60 |
| CIFAR-10 | **84.05** $\pm$ 0.13 | 83.51 $\pm$ 0.17 | 83.52 $\pm$ 0.16 | 79.60 $\pm$ 0.47 | 81.09 $\pm$ 0.81 | 78.29 $\pm$ 0.34 |
| CIFAR-100 | **54.53** $\pm$ 0.71 | 45.32 $\pm$ 0.85 | 50.59 $\pm$ 0.92 | 42.31 $\pm$ 1.64 | 45.02 $\pm$ 1.92 | 44.51 $\pm$ 1.34 |
| Shakespeare | **53.03** $\pm$ 0.11 | 52.46 $\pm$ 0.14 | 52.52 $\pm$ 0.13 | 52.23 $\pm$ 0.11 | 39.24 $\pm$ 1.89 | 52.30 $\pm$ 0.18 |
| Dataset | FedAvgM | MimeLite | SCAFFOLD | FedDyn | FedProx | FedAvg |
| EMNIST | 81.92 $\pm$ 0.19 | 81.91 $\pm$ 0.14 | 81.55 $\pm$ 0.15 | 81.02 $\pm$ 0.65 | 81.83 $\pm$ 0.56 | 81.01 $\pm$ 0.53 |
| CIFAR-10 | 81.86 $\pm$ 0.37 | 81.02 $\pm$ 0.23 | 81.34 $\pm$ 0.34 | 80.14 $\pm$ 0.61 | 78.88 $\pm$ 0.51 | 78.64 $\pm$ 0.39 |
| CIFAR-100 | 48.31 $\pm$ 1.67 | 47.59 $\pm$ 0.80 | 42.24 $\pm$ 1.18 | 41.89 $\pm$ 1.19 | 41.74 $\pm$ 1.12 | 40.79 $\pm$ 1.34 |
| Shakespeare | 52.32 $\pm$ 0.08 | 51.71 $\pm$ 0.31 | 52.04 $\pm$ 0.25 | 51.77 $\pm$ 0.23 | 51.54 $\pm$ 0.21 | 51.61 $\pm$ 0.19 |

## A.9 CONVERGENCE AND TOP VALIDATION ACCURACY

The experimental results demonstrate that our algorithm consistently outperforms the majority of existing optimization algorithms proposed to address data heterogeneity as shown in Fig 4 and Fig 5.

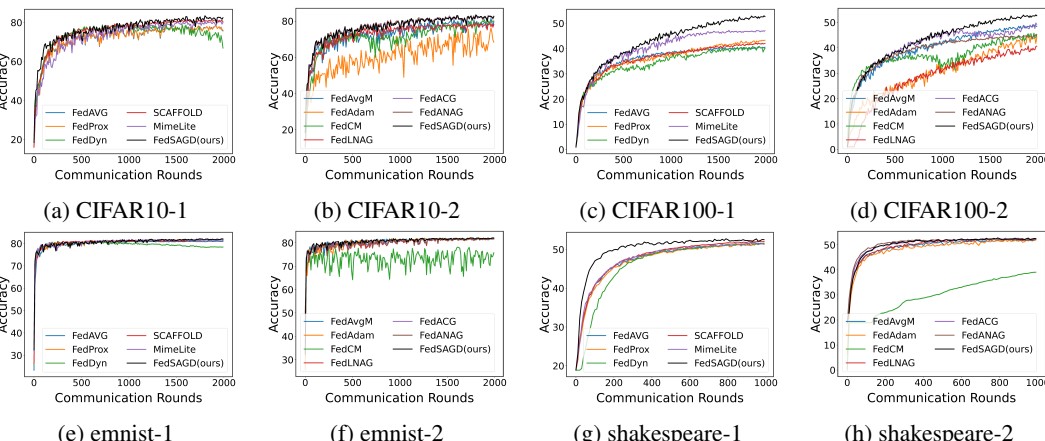

Figure 4: Convergence, top validation accuracy for various FL methods in the different datasets.

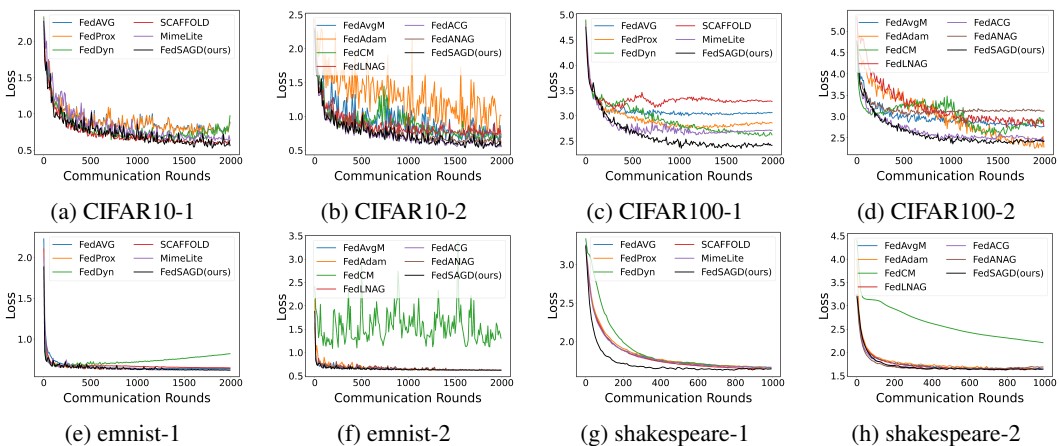

Figure 5: Convergence, loss for various FL methods in the different datasets.

## A.10 ABLATION STUDIES

Compared with the FedAvg, FedSpeed adds two main modules: (1) global momentum, (2) hybrid proximal term. We test the performance of 2000 communication rounds of the different combination of the modules above on the CIFAR-10 with the settings of 2% participating ratio of total 200 clients. The Table 17 shows their performance.

Table 17: Ablation studies on different modules.

| Momentum | Hybrid proximal term | Accuracy |
|:---:|:---:|:---:|
| ✓ | ✗ | 82.84 |
| ✗ | ✓ | 80.54 |
| ✓ | ✓ | 84.05 |

## A.11 CONCEPTUAL ILLUSTRATION OF STABILITY

Fig. 6 provides a conceptual illustration of stability. In Fig. 6a, each curve represents the local update trajectory of a client. Stronger local-update stability implies that variations in the set of clients selected for local training have a smaller impact on the aggregated model. In Fig. 6b, each curve denotes the global update trajectory of an FL task. Stronger global-update stability indicates that changes in the participating client set exert less influence on the final global model.

For local-update stability, under the same learning rate, FedSAGD achieves a smaller maximum local model discrepancy than FedAvg and remains comparable to FedProx. FedSAGD (blue solid line) consistently updates with a larger step size than FedProx (green solid line), thereby moving closer to the global optimum of the participating clients. This enables superior performance while maintaining the same maximum model discrepancy. As illustrated, the updates of FedSAGD are closest to $X^*$ (black solid line), followed by FedProx, while FedAvg remains the farthest. When the learning rate is set to $\eta_l = \mathcal{O}\left(\frac{1}{LK\sqrt{T}}\right)$, FedProx, FedAvg, and FedSAGD achieve the same maximum model discrepancy; however, due to momentum acceleration, FedSAGD still outperforms FedProx.

For global-update stability, regardless of whether the learning rate is fixed or set as $\eta_l = \mathcal{O}\left(\frac{1}{LK\sqrt{T}}\right)$, FedSAGD consistently yields a smaller global model discrepancy compared to both FedAvg and FedProx. FedSAGD (blue dashed line) updates with a larger step size than FedProx (green dashed line), and thus, while maintaining a smaller maximum model discrepancy, it approaches the global optimum of all clients more closely.

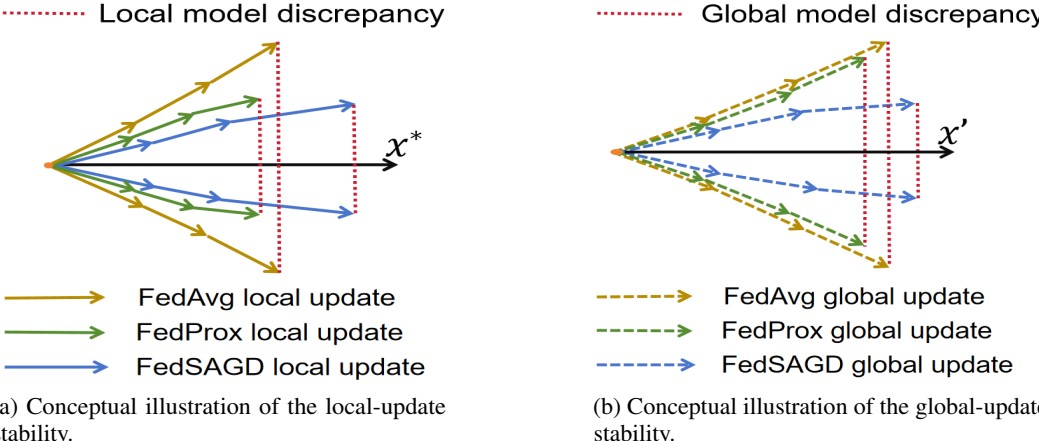

(a) Conceptual illustration of the local-update stability.

(b) Conceptual illustration of the global-update stability.

Figure 6: Conceptual illustration of stability. $X^*$ is the the global optimum of the participating clients; $X'$ is the global optimum of all clients.

## B    PROOF OF THEOREM 1

In this section, we provide the proof of Theorem 3.6.

Given that $\mathcal{S}$ and $\mathcal{S}'$ only differ by one client. According to the definition of $R_e$ and $R_e$, we have

$$\mathbb{E}_{\mathcal{S}}\mathbb{E}_{\mathcal{A}}\left[\frac{1}{|\mathcal{S}|}\sum_{i\in\mathcal{S}}\mathbb{E}_{\xi_i\sim\mathcal{P}_i}[F_i(A(\mathcal{S}),\xi_i)]\right]$$

$$= \mathbb{E}_{\mathcal{S}'}\mathbb{E}_{\mathcal{S}}\mathbb{E}_{\mathcal{A}}\left[\frac{1}{|\mathcal{S}|}\sum_{i\in\mathcal{S}}\mathbb{E}_{\xi_i\sim\mathcal{P}_i}[F_i(A(\mathcal{S}),\xi_i)]\right]$$

$$= \mathbb{E}_{\mathcal{S}'}\mathbb{E}_{\mathcal{S}}\mathbb{E}_{\mathcal{A}}\left[\frac{1}{|\mathcal{S}|}\sum_{i\in\mathcal{S}}(\mathbb{E}_{\xi_i\sim\mathcal{P}_i}[F_i(A(\mathcal{S}),\xi_i)] - \mathbb{E}_{\xi_i\sim\mathcal{P}_i}[F_i(A(\mathcal{S}'),\xi_i)])\right]$$

$$+ \mathbb{E}_{\mathcal{S}'}\mathbb{E}_{\mathcal{S}}\mathbb{E}_{\mathcal{A}}\left[\frac{1}{|\mathcal{S}|}\sum_{i\in\mathcal{S}}\mathbb{E}_{\xi_i\sim\mathcal{P}_i}[F_i(A(\mathcal{S}'),\xi_i)]\right]$$

$$= \mathbb{E}_{\mathcal{S}'}\mathbb{E}_{\mathcal{S}}\mathbb{E}_{\mathcal{A}}\left[\frac{1}{|\mathcal{S}|}\sum_{i\in\mathcal{S}}(\mathbb{E}_{\xi_i\sim\mathcal{P}_i}[F_i(A(\mathcal{S}),\xi_i)] - \mathbb{E}_{\xi_i\sim\mathcal{P}_i}[F_i(A(\mathcal{S}'),\xi_i)])\right]$$

$$+ \mathbb{E}_{\mathcal{S}'}\mathbb{E}_{\mathcal{A}}\left[\mathbb{E}_{i\sim\mathcal{C}}[\mathbb{E}_{\xi_i\sim\mathcal{P}_i}[F_i(A(\mathcal{S}'),\xi_i)]]\right].$$

Since $\mathcal{A}$ satisfies $\epsilon$-global update stability, rearranging the above equality we can get

$$\mathbb{E}_{\mathcal{S}}\mathbb{E}_{\mathcal{A}}[|R_u(A(\mathcal{S})) - R_e(A(\mathcal{S}))|] \leq \epsilon.$$

## C    STABILITY UPPER BOUND FOR NON-CONVEX LOSSES

**Lemma C.1.** *Suppose Assumption 3.1-3.4 hold. Then*

$$\mathbb{E}\|\nabla F_i(x_{t,k}^i;\xi_i)\| \leq (\sigma_l^2 + 2\sigma_g^2 + 2Z^2)^{\frac{1}{2}}, \forall i\in[M], \forall 0\leq k\leq K-1.$$

*Proof.*

$$\mathbb{E}\|\nabla F_i(x_{t,k}^i)\|^2 = \mathbb{E}\|\nabla F_i(x_{t,k}^i) - \nabla f(x_{t,k}^i) + \nabla f(x_{t,k}^i)\|^2$$
$$\leq 2\mathbb{E}\|\nabla F_i(x_{t,k}^i) - \nabla f(x_{t,k}^i)\|^2 + 2\mathbb{E}\|\nabla f(x_{t,k}^i)\|^2.$$

Since

$$\mathbb{E}\|\nabla F_i(x_{t,k}^i,\xi_i) - \nabla F_i(x_{t,k}^i)\|^2$$
$$= \mathbb{E}\|\nabla F_i(x_{t,k}^i,\xi_i)\|^2 + \mathbb{E}\|\nabla F_i(x_{t,k}^i)\|^2 - 2\langle\mathbb{E}\nabla F_i(x_{t,k}^i,\xi_i),\nabla F_i(x_{t,k}^i)\rangle$$
$$= \mathbb{E}\|\nabla F_i(x_{t,k}^i,\xi_i)\|^2 - \mathbb{E}\|\nabla F_i(x_{t,k}^i)\|^2,$$

we have

$$\mathbb{E}\|\nabla F_i(x_{t,k}^i,\xi_i)\|^2$$
$$\leq \mathbb{E}\|\nabla F_i(x_{t,k}^i,\xi_i) - \nabla F_i(x_{t,k}^i)\|^2 + 2\mathbb{E}\|\nabla F_i(x_{t,k}^i) - \nabla f(x_{t,k}^i)\|^2 + 2\mathbb{E}\|\nabla f(x_{t,k}^i)\|^2$$
$$\leq \sigma_l^2 + 2\sigma_g^2 + 2Z^2.$$

The last inequality follows from Assumption 3.4, by the Lipschitz continuity, we can obtain that $\nabla f \leq Z$. □

### C.1    ANALYSIS FOR FEDAVG UNDER NON-CONVEX LOSSES

The local update at iteration $k$ is described as follows:

$$x_{t,k+1}^i = x_{t,k}^i - \eta_l\nabla F_i(x_{t,k}^i,\xi_i).$$

**Theorem C.2.** *(global update stability of FedAvg). Suppose Assumption 3.1-3.4 hold and consider FedAvg. Let $x_T$ and $x'_T$ be two model results obtained by neighboring active client-sets $\mathcal{S}$ and $\mathcal{S}'$, respectively. Under the assumption that $F_i$ is a non-convex and L-smooth function, then*

$$\mathbb{E}\|x_T - x'_T\| \le 2\eta_l K(\sigma_l^2 + 2\sigma_g^2 + 2Z^2)^{\frac{1}{2}}T.$$

*Proof.* Consider two clients $i \in \mathcal{S}$ and $j \in \mathcal{S}'$. According to the local update formula, we have

$$\mathbb{E}\|x_{t,k}^i - x_{t,k}^j\|$$
$$= \mathbb{E}\|x_{t,k-1}^i - \eta_l \nabla F_i(x_{t,k-1}^i, \xi_i) - (x_{t,k-1}^j - \eta_l \nabla F_j(x_{t,k-1}^j, \xi_j))\|$$
$$\le \mathbb{E}\|x_{t,k-1}^i - x_{t,k-1}^j\| + \eta_l \mathbb{E}\|\nabla F_i(x_{t,k-1}^i, \xi_i)\| + \eta_l \mathbb{E}\|\nabla F_j(x_{t,k-1}^j, \xi_j)\|$$
$$\le \mathbb{E}\|x_{t,k-1}^i - x_{t,k-1}^j\| + 2(\sigma_l^2 + 2\sigma_g^2 + 2Z^2)^{\frac{1}{2}}\eta_l,$$

where the last inequality we use Lemma C.1. Unrolling it gives:

$$\mathbb{E}\|x_{t,K}^i - x_{t,K}^j\| \le \mathbb{E}\|x_t - x'_t\| + 2(\sigma_l^2 + 2\sigma_g^2 + 2Z^2)^{\frac{1}{2}}\eta_l K.$$

Note: In the local stability analysis of Zhang et al. (2025), it holds that $\mathbb{E}\|x_t - x'_t\| = 0$. Therefore, the local-update stability of the FedAvg algorithm is given by $\mathcal{O}\left(2(\sigma_l^2 + 2\sigma_g^2 + 2Z^2)^{\frac{1}{2}}\eta_l K\right)$.

Similarly, the local-update stability of subsequent algorithms can be derived from $\mathbb{E}\|x_{t,K}^i - x_{t,K}^j\|$. Since $x_{t+1} = x_t + \frac{1}{s}\sum_{i \in \mathcal{S}}(x_{t,K}^i - x_t^i)$, we have

$$\mathbb{E}\|x_{t+1} - x'_{t+1}\| = \frac{1}{s}\mathbb{E}\|\sum_{i \in \mathcal{S}} x_{t,K}^i - \sum_{j \in \mathcal{S}'} x_{t,K}^j\|$$
$$\le \mathbb{E}\|x_t - x'_t\| + 2\eta_l K(\sigma_l^2 + 2\sigma_g^2 + 2Z^2)^{\frac{1}{2}}.$$

Note that $x_0 = x'_0$, then unrolling it gives

$$\mathbb{E}\|x_T - x'_T\| \le 2\eta_l K(\sigma_l^2 + 2\sigma_g^2 + 2Z^2)^{\frac{1}{2}}T.$$

$\square$

### C.2 ANALYSIS FOR FEDPROX UNDER NON-CONVEX LOSSES

The local update at iteration $k$ is described as follows:

$$x_{t,k+1}^i = x_{t,k}^i - \eta_l(\nabla F_i(x_{t,k}^i, \xi_i) + \lambda(x_{t,k}^i - x_t)).$$

**Theorem C.3.** *(global update stability of of FedProx). Suppose Assumption 3.1-3.4 hold and consider FedProx. Let $x_T$ and $x'_T$ be two model results obtained by neighboring active client-sets $\mathcal{S}$ and $\mathcal{S}'$, respectively. Under the assumption that $F_i$ is a non-convex and L-smooth function, then*

$$\mathbb{E}\|x_T - x'_T\| \le \frac{2}{\lambda}(\sigma_l^2 + 2\sigma_g^2 + 2Z^2)^{\frac{1}{2}}[1 - (1 - \lambda\eta_l)^K]T.$$

*Proof.* Consider two clients $i \in \mathcal{S}$ and $j \in \mathcal{S}'$. According to the local update rule, we have

$$\mathbb{E}\|x_{t,k}^i - x_{t,k}^j\|$$
$$= \mathbb{E}\|(1 - \eta_l\lambda)(x_{t,k-1}^i - x_{t,k-1}^j) - \eta_l \nabla F_i(x_{t,k-1}^i, \xi_i) + \eta_l(\nabla F_j(x_{t,k-1}^j, \xi_j) + \lambda\eta_l(x_t - x'_t)\|$$
$$\le \mathbb{E}\|(1 - \eta_l\lambda)(x_{t,k-1}^i - x_{t,k-1}^j)\| + \eta_l\mathbb{E}\|\nabla F_i(x_{t,k-1}^i, \xi_i) - \nabla F_j(x_{t,k-1}^j, \xi_j)\| + \lambda\eta_l\mathbb{E}\|x_t - x'_t\|$$
$$\le (1 - \lambda\eta_l)\mathbb{E}\|x_{t,k-1}^i - x_{t,k-1}^j\| + 2(\sigma_l^2 + 2\sigma_g^2 + 2Z^2)^{\frac{1}{2}}\eta_l + \lambda\eta_l\mathbb{E}\|x_t - x'_t\|,$$

where the last inequality follows Lemma C.1. Unrolling it gives:

$$\mathbb{E}\|x_{t,K}^i - x_{t,K}^j\| \le (1-\lambda\eta_l)^K \mathbb{E}\|x_t - x_t'\| + \sum_{k=0}^{K-1}(1-\lambda\eta_l)^k[2(\sigma_l^2 + 2\sigma_g^2 + 2Z^2)^{\frac{1}{2}}\eta_l$$

$$+ \lambda\eta_l\mathbb{E}\|x_t - x_t'\|]$$

$$= (1-\lambda\eta_l)^K \mathbb{E}\|x_t - x_t'\| + \frac{1-(1-\lambda\eta_l)^K}{\lambda\eta_l}[2(\sigma_l^2 + 2\sigma_g^2 + 2Z^2)^{\frac{1}{2}}\eta_l$$

$$+ \lambda\eta_l\mathbb{E}\|x_t - x_t'\|]$$

$$= \mathbb{E}\|x_t - x_t'\| + \frac{2}{\lambda}(\sigma_l^2 + 2\sigma_g^2 + 2Z^2)^{\frac{1}{2}}[1-(1-\lambda\eta_l)^K].$$

Since $x_{t+1} = \frac{1}{s}\sum_{i\in\mathcal{S}} x_{t,K}^i$, we obtain

$$\mathbb{E}\|x_{t+1} - x_{t+1}'\| = \frac{1}{s}\mathbb{E}\|\sum_{i\in\mathcal{S}} x_{t,K}^i - \sum_{j\in\mathcal{S}'} x_{t,K}^j\|$$

$$\le \mathbb{E}\|x_t - x_t'\| + \frac{2}{\lambda}(\sigma_l^2 + 2\sigma_g^2 + 2Z^2)^{\frac{1}{2}}[1-(1-\lambda\eta_l)^K].$$

Note that $x_0 = x_0'$, then unrolling it gives

$$\mathbb{E}\|x_T - x_T'\| \le \frac{2}{\lambda}(\sigma_l^2 + 2\sigma_g^2 + 2Z^2)^{\frac{1}{2}}[1-(1-\lambda\eta_l)^K]T.$$

$\square$

## C.3   ANALYSIS FOR SCAFFOLD UNDER NON-CONVEX LOSSES

The local update at iteration $k$ is described as follows:

$$x_{t,k+1}^i = x_{t,k}^i - \eta_l(\nabla F_i(x_{t,k}^i, \xi_i) - c_i + c),$$

where $c_i = \frac{1}{K}\sum_{k=0}^{K-1} \nabla F(x_{t-\tau,k}^i, \xi_i)$ is the control variate of the client $i$ whose last participation in training was in the $t-\tau$ round. Note that, unlike other algorithms, the local update here involves gradient information from $t-\tau$ rounds.

**Theorem C.4.** *(global update stability of SCAFFOLD). Suppose Assumption 3.1-3.4 hold and consider SCAFFOLD. Let $x_T$ and $x_T'$ be two model results obtained by neighboring active client-sets $\mathcal{S}$ and $\mathcal{S}'$, respectively. Under the assumption that $F_i$ is a non-convex and $L$-smooth function, then*

$$\mathbb{E}\|x_T - x_T'\| \le 6\eta_l K(\sigma_l^2 + 2\sigma_g^2 + 2Z^2)^{\frac{1}{2}}T.$$

*Proof.* Consider two clients $i \in \mathcal{S}$ and $j \in \mathcal{S}'$. According to the local update rule, we have

$$\mathbb{E}\|x_{t,k}^i - x_{t,k}^j\|$$

$$= \mathbb{E}\|x_{t,k-1}^i - \eta_l(\nabla F_i(x_{t,k-1}^i, \xi_i) - c_i + c) - x_{t,k-1}^j + \eta_l(\nabla F_j(x_{t,k-1}^j, \xi_j) - c_j + c')\|$$

$$\le \mathbb{E}\|x_{t,k-1}^i - x_{t,k-1}^j\| + \eta_l\mathbb{E}\|c_i - c_j\| + \eta_l E\|\nabla F_i(x_{t,k-1}^i, \xi_i) - \nabla F_j(x_{t,k-1}^j, \xi_j)\|$$

$$+ \eta_l\mathbb{E}\|c - c'\|$$

$$\le \mathbb{E}\|x_{t,k-1}^i - x_{t,k-1}^j\| + 6\eta_l(\sigma_l^2 + 2\sigma_g^2 + 2Z^2)^{\frac{1}{2}},$$

where the last inequality follows Lemma C.1. Unrolling it gives:

$$\mathbb{E}\|x_{t,K}^i - x_{t,K}^j\| \le \mathbb{E}\|x_t - x_t'\| + 6(\sigma_l^2 + 2\sigma_g^2 + 2Z^2)^{\frac{1}{2}}\frac{1}{2}\eta_l K.$$

Since $x_{t+1} = x_t + \frac{1}{s}\sum_{i\in\mathcal{S}}(x_{t,K}^i - x_t^i)$, we have

$$\mathbb{E}\|x_{t+1} - x_{t+1}'\| = \frac{1}{s}\mathbb{E}\|\sum_{i\in\mathcal{S}} x_{t,K}^i - \sum_{j\in\mathcal{S}'} x_{t,K}^j\|$$

$$\le \mathbb{E}\|x_t - x_t'\| + 6\eta_l K(\sigma_l^2 + 2\sigma_g^2 + 2Z^2)^{\frac{1}{2}}.$$

Note that $x_0 = x_0'$, then unrolling it gives

$$\mathbb{E}\|x_T - x_T'\| \leq 6\eta_l K(\sigma_l^2 + 2\sigma_g^2 + 2Z^2)^{\frac{1}{2}}T.$$

$\square$

## C.4 ANALYSIS FOR FEDDYN UNDER CONVEX LOSSES

The local update at iteration $k$ is described as follows:

$$x_{t,k+1}^i = x_{t,k}^i - \eta_l(\nabla F_i(x_{t,k}^i, \xi_i) - \nabla F_i(x_{t-\tau,K}^i, \xi_i) + \lambda(x_{t,k}^i - x_t)),$$

where the $t - \tau$ is because clients do not participate in training every round and recently participated in round $t - \tau$.

**Theorem C.5.** *(global update stability of FedDyn). Suppose Assumption 3.1-3.4 hold and consider FedDyn. Let $x_T$ and $x_T'$ be two model results obtained by neighboring active client-sets $\mathcal{S}$ and $\mathcal{S}'$, respectively. Under the assumption that $F_i$ is a non-convex and $L$-smooth function, then*

$$\mathbb{E}\|x_T - x_T'\| \leq \frac{4}{\lambda}(\sigma_l^2 + 2\sigma_g^2 + 2Z^2)^{\frac{1}{2}}[1 - (1 - \lambda\eta)^K]T.$$

*Proof.* Consider two clients $i \in \mathcal{S}$ and $j \in \mathcal{S}'$. According to the local update rule, we have

$$\mathbb{E}\|x_{t,k}^i - x_{t,k}^j\|$$
$$= \mathbb{E}\|x_{t,k-1}^i - \eta_l(\nabla F_i(x_{t,k-1}^i, \xi_i) - \nabla F_i(x_{t-\tau,K}^i, \xi_i) + \lambda(x_{t,k-1}^i - x_t)) - x_{t,k-1}^j$$
$$\quad + \eta_l(\nabla F_j(x_{t,k-1}^j, \xi_j) - \nabla F_j(x_{t-\tau,K}^i, \xi_j) + \lambda(x_{t,k-1}^j - x_t'))\|$$
$$\leq \mathbb{E}\|x_{t,k-1}^i - x_{t,k-1}^j - \lambda\eta_l(x_{t,k-1}^i - x_{t,k-1}^j)\| + \eta_l\mathbb{E}\|\nabla F_i(x_{t,k-1}^i, \xi_i) - \nabla F_j(x_{t,k-1}^j, \xi_j)\|$$
$$\quad + \eta_l\mathbb{E}\|\nabla F_i(x_{t-\tau,K}^i, \xi_i) - \nabla F_j(x_{t-\tau,K}^j, \xi_j)\| + \lambda\eta_l\mathbb{E}\|x_t - x_t'\|$$
$$\leq (1 - \lambda\eta_l)\mathbb{E}\|x_{t,k-1}^i - x_{t,k-1}^j\| + 4\eta_l(\sigma_l^2 + 2\sigma_g^2 + 2Z^2)^{\frac{1}{2}} + \lambda\eta_l\mathbb{E}\|x_t - x_t'\|,$$

where the last inequality follows Lemma C.1. Unrolling it gives:

$$\mathbb{E}\|x_{t,K}^i - x_{t,K}^j\| \leq \mathbb{E}\|x_t - x_t'\| + \frac{4}{\lambda}(\sigma_l^2 + 2\sigma_g^2 + 2Z^2)^{\frac{1}{2}}[1 - (1 - \lambda\eta_l)^K].$$

Since $x_{t+1} = x_t + \frac{1}{s}\sum_{i\in\mathcal{S}}(x_{t,K}^i - x_{t,0}^i)$, we obtain

$$\mathbb{E}\|x_{t+1} - x_{t+1}'\| = \frac{1}{s}\mathbb{E}\|\sum_{i\in\mathcal{S}} x_{t,K}^i - \sum_{j\in\mathcal{S}'} x_{t,K}^j\|$$
$$\leq \mathbb{E}\|x_t - x_t'\| + \frac{4}{\lambda}(\sigma_l^2 + 2\sigma_g^2 + 2Z^2)^{\frac{1}{2}}[1 - (1 - \lambda\eta_l)^K].$$

Note that $x_0 = x_0'$, then unrolling it gives

$$\mathbb{E}\|x_T - x_T'\| \leq \frac{4}{\lambda}(\sigma_l^2 + 2\sigma_g^2 + 2Z^2)^{\frac{1}{2}}[1 - (1 - \lambda\eta_l)^K]T.$$

$\square$

## C.5 ANALYSIS FOR FEDLNAG UNDER NON-CONVEX LOSS

Consider the orignial update rule of NAG:

$$y_{t,k}^i = x_{t,k}^i - \beta m_{t,k}^i$$
$$m_{t,k+1}^i = \beta m_{t,k}^i + (1 - \beta)\nabla F_i(y_{t,k}^i, \xi_i)$$
$$x_{t,k+1}^i = x_{t,k}^i - \eta_l m_{t,k+1}^i.$$

**Theorem C.6.** *(global update stability of FedLNAG). Suppose Assumption 3.1-3.4 hold and consider FedLNAG. Let $x_T$ and $x'_T$ be two model results obtained by neighboring active client-sets $\mathcal{S}$ and $\mathcal{S}'$, respectively. Under the assumption that $F_i$ is a non-convex and L-smooth function, then*

$$\mathbb{E}\|x_T - x'_T\| \le 2\eta_l K(\sigma_l^2 + 2\sigma_g^2 + 2Z^2)^{\frac{1}{2}}T.$$

*Proof.* For the momentum term $m_{t,k}$, by recursion we have

$$m_{t,k} = (1-\beta)\sum_{\tau=0}^{k-1}\beta^\tau \nabla F(y_{t,k-\tau-1}, \xi).$$

Consider two clients $i \in \mathcal{S}$ and $j \in \mathcal{S}'$. According to the local update rule, we have

$$\mathbb{E}\|x_{t,k+1}^i - x_{t,k+1}^j\|$$
$$= \mathbb{E}\|x_{t,k}^i - \eta_l m_{t,k+1}^i - (x_{t,k}^j - \eta_l m_{t,k+1}^j)\|$$
$$\le \mathbb{E}\|x_{t,k}^i - x_{t,k}^j\| + \eta_l\|m_{t,k+1}^i\| + \eta_l\|m_{t,k+1}^j\|$$
$$\le \mathbb{E}\|x_{t,k}^i - x_{t,k}^j\| + \eta_l(1-\beta)\sum_{\tau=0}^{k}\beta^\tau\|\nabla F_i(y_{t,k-\tau}^i, \xi_i)\| + \eta_l(1-\beta)\sum_{\tau=0}^{k}\beta^\tau\|\nabla F_j(y_{t,k-\tau}^j, \xi_j)\|$$
$$\le \mathbb{E}\|x_{t,k}^i - x_{t,k}^j\| + 2\eta_l(1-\beta^{k+1})(\sigma_l^2 + 2\sigma_g^2 + 2Z^2)^{\frac{1}{2}}.$$

where the last inequality follows Lemma C.1. Unrolling it gives

$$\mathbb{E}\|x_{t,K}^i - x_{t,K}^j\| \le \mathbb{E}\|x_t - x'_t\| + \|2\eta_l(\sigma_l^2 + 2\sigma_g^2 + 2Z^2)^{\frac{1}{2}}K.$$

Since $x_{t+1} = x_t + \frac{1}{s}\sum_{i\in\mathcal{S}}(x_{t,K}^i - x_{t,0}^i)$, we obtain

$$\mathbb{E}\|x_{t+1} - x'_{t+1}\| = \frac{1}{s}\mathbb{E}\|\sum_{i\in\mathcal{S}}x_{t,K}^i - \sum_{j\in\mathcal{S}'}x_{t,K}^j\|$$
$$\le \mathbb{E}\|x_t - x'_t\| + 2\eta_l(\sigma_l^2 + 2\sigma_g^2 + 2Z^2)^{\frac{1}{2}}K.$$

Note that $x_0 = x'_0$, then unrolling it gives

$$\mathbb{E}\|x_T - x'_T\| \le 2\eta_l K(\sigma_l^2 + 2\sigma_g^2 + 2Z^2)^{\frac{1}{2}}T.$$

$\square$

## C.6 ANALYSIS FOR FEDADAM UNDER NON-CONVEX LOSSES

The local update at iteration $k$ is described as follows:

$$x_{t,k+1}^i = x_{t,k}^i - \eta_l \nabla F_i(x_{t,k}^i, \xi_i).$$

**Theorem C.7.** *(global update stability of FedAdam). Suppose Assumption 3.1-3.4 hold and consider FedAdam. Let $x_T$ and $x'_T$ be two model results obtained by neighboring active client-sets $\mathcal{S}$ and $\mathcal{S}'$, respectively. Under the assumption that $F_i$ is a non-convex and L-smooth function, then*

$$\mathbb{E}\|x_T - x'_T\| \le \frac{2}{\tau}\eta_l K(\sigma_l^2 + 2\sigma_g^2 + 2Z^2)^{\frac{1}{2}}T.$$

*Proof.* Firstly,

$$\Delta_t^i = x_{t,K}^i - x_t$$
$$= x_{t,K-1}^i - \eta_l \nabla F_i(x_{t,K-1}^i, \xi_i) - x_t$$
$$= \sum_{k=0}^{K-1} -\eta_l \nabla F_i(x_{t,k}^i, \xi_i),$$

By using Lemma C.1 we get

$$\mathbb{E}\|\Delta_t^i\| \leq \eta_l K(\sigma_l^2 + 2\sigma_g^2 + 2Z^2)^{\frac{1}{2}}.$$

Note that $\Delta_t = \beta_1\Delta_{t-1} + (1-\beta_1)(\frac{1}{s}\sum_{i\in\mathcal{S}}\Delta_t^i)$, so we can deduce that

$$\mathbb{E}\|\Delta_t\| \leq \eta_l K(\sigma_l^2 + 2\sigma_g^2 + 2Z^2)^{\frac{1}{2}}.$$

since $x_{t+1} = x_t + \frac{\Delta_t}{\sqrt{v_t}+\tau}$, we have

$$\mathbb{E}\|x_{t+1} - x'_{t+1}\| = \mathbb{E}\|x_t + \frac{\Delta_t}{\sqrt{v_t}+\tau} - (x'_t + \frac{\Delta'_t}{\sqrt{v'_t}+\tau})\|$$

$$\leq \mathbb{E}\|x_t - x'_t\| + \mathbb{E}\|\frac{\Delta_t}{\sqrt{v_t}+\tau}\| + \mathbb{E}\|\frac{\Delta'_t}{\sqrt{v'_t}+\tau}\|$$

$$\leq \mathbb{E}\|x_t - x'_t\| + \mathbb{E}\|\frac{\Delta_t}{\tau}\| + \mathbb{E}\|\frac{\Delta'_t}{\tau}\|$$

$$\leq \mathbb{E}\|x_t - x'_t\| + \frac{2}{\tau}\eta_l K(\sigma_l^2 + 2\sigma_g^2 + 2Z^2)^{\frac{1}{2}}.$$

Note that $x_0 = x'_0$, then unrolling it gives

$$\mathbb{E}\|x_T - x'_T\| \leq \frac{2}{\tau}\eta_l K(\sigma_l^2 + 2\sigma_g^2 + 2Z^2)^{\frac{1}{2}}T.$$

$\square$

## C.7 ANALYSIS FOR FEDCM UNDER NON-CONVEX LOSSES

The local update at iteration $k$ is described as follows:

$$x_{t,k+1}^i = x_{t,k}^i - \eta_l(\alpha\nabla F_i(x_{t,k}^i, \xi_i) + (1-\alpha)m_t),$$

where $\alpha \in [0,1]$ is momentum parameter.

**Theorem C.8.** *(global update stability of FedCM). Suppose Assumption 3.1-3.4 hold and consider FedCM. Let $x_T$ and $x'_T$ be two model results obtained by neighboring active client-sets $\mathcal{S}$ and $\mathcal{S}'$, respectively. Under the assumption that $F_i$ is a non-convex and $L$-smooth function, then*

$$\mathbb{E}\|x_T - x'_T\| \leq 2(\sigma_l^2 + 2\sigma_g^2 + 2Z^2)^{\frac{1}{2}}T.$$

*Proof.* Consider two clients $i \in \mathcal{S}$ and $j \in \mathcal{S}'$. Accoring to the local update rule, we have

$$\mathbb{E}\|x_{t,k}^i - x_{t,k}^j\|$$

$$= \mathbb{E}\|x_{t,k-1}^i - x_{t,k-1}^j - \eta_l\alpha(\nabla F_i(x_{t,k-1}^i, \xi_i) - \nabla F_j(x_{t,k-1}^j, \xi_j)) - \eta_l(1-\alpha)(m_t - m'_t)\|$$

$$\leq \mathbb{E}\|x_{t,k-1}^i - x_{t,k-1}^j\| + \eta_l\alpha\mathbb{E}\|\nabla F_i(x_{t,k-1}^i, \xi_i) - \nabla F_j(x_{t,k-1}^j, \xi_j)\| + \eta_l(1-\alpha)\mathbb{E}\|m_t - m'_t\|$$

$$\leq \mathbb{E}\|x_{t,k-1}^i - x_{t,k-1}^j\| + 2\eta_l(\sigma_l^2 + 2\sigma_g^2 + 2Z^2)^{\frac{1}{2}},$$

where the last inequality follows Lemma C.1. Unrolling it gives:

$$\mathbb{E}\|x_{t,K}^i - x_{t,K}^j\| \leq \mathbb{E}\|x_t - x'_t\| + 2\eta_l K(\sigma_l^2 + 2\sigma_g^2 + 2Z^2)^{\frac{1}{2}}.$$

Since $x_{t+1} = x_t + \frac{1}{\eta_l Ks}\sum_{i\in\mathcal{S}}(x_{t,K}^i - x_t)$, we obtain

$$\mathbb{E}\|x_{t+1} - x'_{t+1}\| \leq \mathbb{E}\|(1 - \frac{1}{\eta_l K})(x_t - x'_t)\| + \frac{1}{\eta_l Ks}\mathbb{E}\|\sum_{i\in\mathcal{S}}x_{t,K}^i - \sum_{j\in\mathcal{S}'}x_{t,K}^j\|$$

$$\leq (1 - \frac{1}{\eta_l K})\mathbb{E}\|x_t - x'_t\| + \frac{1}{\eta_l K}\mathbb{E}\|x_t - x'_t\| + 2(\sigma_l^2 + 2\sigma_g^2 + 2Z^2)^{\frac{1}{2}}$$

$$= \mathbb{E}\|x_t - x'_t\| + 2(\sigma_l^2 + 2\sigma_g^2 + 2Z^2)^{\frac{1}{2}}.$$

Note that $x_0 = x'_0$, unrolling it gives

$$\mathbb{E}\|x_T - x'_T\| \leq 2(\sigma_l^2 + 2\sigma_g^2 + 2Z^2)^{\frac{1}{2}}T.$$

$\square$

## C.8 ANALYSIS FOR MIMELITE UNDER NON-CONVEX LOSSES

The local update at iteration $k$ is described as follows:

$$x^i_{t,k+1} = x^i_{t,k} - \eta_l((1-\beta)\nabla F_i(x^i_{t,k}, \xi_i) + \beta m),$$

where $\beta \in [0,1]$ is momentum parameter.

**Theorem C.9.** *(global update stability of MimeLite). Suppose Assumption 3.1-3.4 hold and consider MimeLite. Let $x_T$ and $x'_T$ be two model results obtained by neighboring active client-sets $\mathcal{S}$ and $\mathcal{S}'$, respectively. Under the assumption that $F_i$ is a non-convex and L-smooth function, then*

$$\mathbb{E}\|x_T - x'_T\| \le 2\eta_l K(\sigma_l^2 + 2\sigma_g^2 + 2Z^2)^{\frac{1}{2}} T.$$

*Proof.* Consider two clients $i \in \mathcal{S}$ and $j \in \mathcal{S}'$. According to the local update rule, we have

$$\mathbb{E}\|x^i_{t,k} - x^j_{t,k}\|$$
$$= \mathbb{E}\|x^i_{t,k-1} - \eta_l(1-\beta)\nabla F_i(x^i_{t,k-1}, \xi_i) - (x^j_{t,k-1} - \eta_l(1-\beta)\nabla F_j(x^j_{t,k-1}, \xi_j))\|$$
$$\quad + \beta\eta_l\mathbb{E}\|m - m'\|$$
$$\le \mathbb{E}\|x^i_{t,k-1} - x^j_{t,k-1}\| + \eta_l(1-\beta)\mathbb{E}\|\nabla F_i(x^i_{t,k-1}, \xi_i) - \nabla F_j(x^j_{t,k-1}, \xi_j)\| + \eta_l\beta\mathbb{E}\|m - m'\|$$
$$\le \mathbb{E}\|x^i_{t,k-1} - x^j_{t,k-1}\| + 2\eta_l(\sigma_l^2 + 2\sigma_g^2 + 2Z^2)^{\frac{1}{2}},$$

where the last inequality follows Lemma C.1. Unrolling it gives:

$$\mathbb{E}\|x^i_{t,K} - x^j_{t,K}\| \le \mathbb{E}\|x_t - x'_t\| + 2\eta_l K(\sigma_l^2 + 2\sigma_g^2 + 2Z^2)^{\frac{1}{2}}.$$

Since $x_{t+1} = \frac{1}{s}\sum_{i \in \mathcal{S}} x^i_{t,K}$, we have

$$\mathbb{E}\|x_{t+1} - x'_{t+1}\| = \frac{1}{s}\mathbb{E}\|\sum_{i \in \mathcal{S}} x^i_{t,K} - \sum_{j \in \mathcal{S}'} x^j_{t,K}\|$$
$$\le \mathbb{E}\|x_t - x'_t\| + 2\eta_l K(\sigma_l^2 + 2\sigma_g^2 + 2Z^2)^{\frac{1}{2}}.$$

Note that $x_0 = x'_0$, unrolling it gives

$$\mathbb{E}\|x_T - x'_T\| \le 2\eta_l K(\sigma_l^2 + 2\sigma_g^2 + 2Z^2)^{\frac{1}{2}} T.$$

$\square$

## C.9 ANALYSIS FOR FEDACG UNDER NON-CONVEX LOSSES

The local update at iteration $k$ is described as follows:

$$x^i_{t,k+1} = x^i_{t,k} - \eta_l\nabla F_i(x^i_{t,k}, \xi_i).$$

**Theorem C.10.** *(global update stability of FedACG). Suppose Assumption 3.1-3.4 hold and consider FedACG. Let $x_T$ and $x'_T$ be two model results obtained by neighboring active client-sets $\mathcal{S}$ and $\mathcal{S}'$, respectively. Under the assumption that $F_i$ is a non-convex and L-smooth function, then*

$$\mathbb{E}\|x_T - x'_T\| \le \frac{4-2\lambda}{1-\lambda}\eta_l K(\sigma_l^2 + 2\sigma_g^2 + 2Z^2)^{\frac{1}{2}} T.$$

*Proof.* Consider two clients $i \in \mathcal{S}$ and $j \in \mathcal{S}'$. According to the local update rule, we have

$$\mathbb{E}\|x^i_{t,k} - x^j_{t,k}\|$$
$$= \mathbb{E}\|x^i_{t,k-1} - \eta_l\nabla F_i(x^i_{t,k-1}, \xi_i) - (x^j_{t,k-1} - \eta_l\nabla F_j(x^j_{t,k-1}, \xi_j))\|$$
$$\le \mathbb{E}\|x^i_{t,k-1} - x^j_{t,k-1}\| + \eta_l\mathbb{E}\|\nabla F_i(x^i_{t,k-1}, \xi_i) - \nabla F_j(x^j_{t,k-1}, \xi_j)\|$$
$$\le \mathbb{E}\|x^i_{t,k-1} - x^j_{t,k-1}\| + 2\eta_l(\sigma_l^2 + 2\sigma_g^2 + 2Z^2)^{\frac{1}{2}},$$

where the last inequality follows Lemma C.1. Unrolling it gives:

$$\mathbb{E}\|x_{t,K}^i - x_{t,K}^j\| \le \mathbb{E}\|x_{t,0}^i - x_{t,0}^j\| + 2\eta_l K(\sigma_l^2 + 2\sigma_g^2 + 2Z^2)^{\frac{1}{2}}.$$

Since $x_{t,0}^i = x_{t-1} + \lambda m_{t-1}$ and $m_t = \lambda m_{t-1} + \frac{1}{s}\sum_{i\in\mathcal{S}}(x_{t,K}^i - x_{t,0}^i)$, then

$$x_t = x_{t-1} + m_t$$

$$= x_{t-1} + \lambda m_{t-1} + \frac{1}{s}\sum_{i\in\mathcal{S}}(x_{t,K}^i - x_{t,0}^i)$$

$$= \frac{1}{s}\sum_{i\in\mathcal{S}} x_{t,K}^i,$$

so we have

$$\mathbb{E}\|x_{t+1} - x'_{t+1}\| = \frac{1}{s}\mathbb{E}\|\sum_{i\in\mathcal{S}} x_{t+1,K}^i - \sum_{j\in\mathcal{S}'} x_{t+1,K}^j\|$$

$$\le \mathbb{E}\|x_t + \lambda m_t - (x'_t + \lambda m'_t)\| + 2\eta_l K(\sigma_l^2 + 2\sigma_g^2 + 2Z^2)^{\frac{1}{2}}$$

$$\le \mathbb{E}\|x_t - x'_t\| + \lambda\mathbb{E}\|m_t - m'_t\| + 2\eta_l K(\sigma_l^2 + 2\sigma_g^2 + 2Z^2)^{\frac{1}{2}}$$

$$\le \mathbb{E}\|x_t - x'_t\| + \left(\frac{1-\lambda^t}{1-\lambda} + 1\right) 2\eta_l K(\sigma_l^2 + 2\sigma_g^2 + 2Z^2)^{\frac{1}{2}}$$

$$\le \mathbb{E}\|x_t - x'_t\| + \frac{4-2\lambda}{1-\lambda}\eta_l K(\sigma_l^2 + 2\sigma_g^2 + 2Z^2)^{\frac{1}{2}}.$$

Note that $x_0 = x'_0$, then unrolling it gives

$$\mathbb{E}\|x_T - x'_T\| \le \frac{4-2\lambda}{1-\lambda}\eta_l K(\sigma_l^2 + 2\sigma_g^2 + 2Z^2)^{\frac{1}{2}}T.$$

$\square$

### C.10    ANALYSIS FOR FEDANAG UNDER NON-CONVEX LOSSES

The local update at iteration $k$ is described as follows:

$$x_{t,k+1}^i = x_{t,k}^i - \eta_l((1+\beta)\nabla F_i(x_{t,k}^i, \xi_i) + \beta^2 m_t).$$

**Theorem C.11.** *(global update stability of FedANAG). Suppose Assumption 3.1-3.4 hold and consider FedANAG. Let $x_T$ and $x'_T$ be two model results obtained by neighboring active client-sets $\mathcal{S}$ and $\mathcal{S}'$, respectively. Under the assumption that $F_i$ is a non-convex and L-smooth function, then*

$$\mathbb{E}\|x_T - x'_T\| \le \frac{2}{1-\beta}\eta_l K(\sigma_l^2 + 2\sigma_g^2 + 2Z^2)^{\frac{1}{2}}T.$$

*Proof.* Consider two clients $i \in \mathcal{S}$ and $j \in \mathcal{S}'$. According to the local update rule, we have

$$\mathbb{E}\|x_{t,k}^i - x_{t,k}^j\|$$

$$= \mathbb{E}\|x_{t,k-1}^i - \eta_l(1+\beta)\nabla F_i(x_{t,k-1}^i, \xi_i) - (x_{t,k-1}^j - \eta_l(1+\beta)\nabla F_j(x_{t,k-1}^j, \xi_j))$$

$$\quad - \eta_l\beta^2(m_t - m'_t)\|$$

$$\le \mathbb{E}\|x_{t,k-1}^i - x_{t,k-1}^j\| + \eta_l(1+\beta)\mathbb{E}\|\nabla F_i(x_{t,k-1}^i, \xi_i) - \nabla F_j(x_{t,k-1}^j, \xi_j)\|$$

$$\quad + \eta_l\beta^2\mathbb{E}\|m_t - m'_t\|$$

$$\le \mathbb{E}\|x_{t,k-1}^i - x_{t,k-1}^j\| + \left(1 + \beta + \frac{\beta^2(1-\beta^t)}{1-\beta}\right) 2\eta_l(\sigma_l^2 + 2\sigma_g^2 + 2Z^2)^{\frac{1}{2}}$$

$$\le \mathbb{E}\|x_{t,k-1}^i - x_{t,k-1}^j\| + \frac{2}{1-\beta}\eta_l(\sigma_l^2 + 2\sigma_g^2 + 2Z^2)^{\frac{1}{2}}.$$

where the second inequality follows Lemma C.1. Unrolling it gives:

$$\mathbb{E}\|x_{t,K}^i - x_{t,K}^j\| \le \mathbb{E}\|x_t - x_t'\| + \frac{2}{1-\beta}\eta_l K(\sigma_l^2 + 2\sigma_g^2 + 2Z^2)^{\frac{1}{2}}.$$

Since $x_{t+1} = x_t + \frac{1}{s}\sum_{i\in\mathcal{S}}(x_{t,K}^i - x_{t,0}^i)$, we have

$$\mathbb{E}\|x_{t+1} - x_{t+1}'\| = \frac{1}{s}\mathbb{E}\|\sum_{i\in\mathcal{S}} x_{t,K}^i - \sum_{j\in\mathcal{S}'} x_{t,K}^j\|$$

$$\le \mathbb{E}\|x_t - x_t'\| + \frac{2}{1-\beta}\eta_l K(\sigma_l^2 + 2\sigma_g^2 + 2Z^2)^{\frac{1}{2}}.$$

Note that $x_0 = x_0'$, then unrolling it gives

$$\mathbb{E}\|x_T - x_T'\| \le \frac{2}{1-\beta}\eta_l K(\sigma_l^2 + 2\sigma_g^2 + 2Z^2)^{\frac{1}{2}}T.$$

$\square$

### C.11 ANALYSIS FOR FEDSAGD UNDER NON-CONVEX LOSSES

The local update at iteration $k$ is described as follows:

$$x_{t,k+1}^i = x_{t,k}^i - \eta_l(\nabla F_i(x_{t,k}^i, \xi_i) + \beta v_t + \lambda(x_{t,k}^i - x_{t,0}^i) - \mu x_{t,k}^i),$$

where $\beta \in (0,1)$ are the momentum coefficients.

**Theorem C.12.** *(global update stability of FedSAGD). Suppose Assumption 3.1-3.4 hold and consider FedSAGD. Let $x_T$ and $x_T'$ be two model results obtained by neighboring active client-sets $\mathcal{S}$ and $\mathcal{S}'$, respectively. Under the assumption that $F_i$ is a non-convex and $L$-smooth function, then*

$$\mathbb{E}\|x_T - x_T'\| \le \frac{1-(1-\Gamma)^T}{\Gamma}\zeta(2+2\beta)(\sigma_l^2 + 2\sigma_g^2 + 2Z^2)^{\frac{1}{2}},$$

*where $\Gamma = 1 - \left[\frac{\lambda}{\lambda+\mu} + \frac{\mu}{\lambda+\mu}(1-\eta_l(\lambda+\mu))^K\right] \in (0,1)$ and $\zeta = \frac{1-(1-\eta_l(\lambda+\mu))^K}{\lambda+\mu}$.*

*Proof.* Consider two clients $i \in \mathcal{S}$ and $j \in \mathcal{S}'$. According to the local update formula, we have

$$\mathbb{E}\|x_{t,k}^i - x_{t,k}^j\|$$
$$= \mathbb{E}\|x_{t,k-1}^i - \eta_l\nabla F_i(x_{t,k-1}^i, \xi_i) - \eta_l(\lambda+\mu)x_{t,k-1}^i - \beta\eta_l v_t - (x_{t,k}^j - \eta_l\nabla F_j(x_{t,k-1}^j, \xi_j)$$
$$\quad - \eta_l(\lambda+\mu)x_{t,k-1}^j - \beta\eta_l v_t') + \eta_l\lambda(x_t - x_t')\|$$
$$\le \mathbb{E}\|(1-\eta_l(\lambda+\mu))(x_{t,k-1}^i - x_{t,k-1}^j)\| + \eta_l\mathbb{E}\|\nabla F_i(x_{t,k-1}^i, \xi_i) - \nabla F_j(x_{t,k-1}^j, \xi_j)\|$$
$$\quad + \beta\eta_l\mathbb{E}\|v_t - v_t'\| + \eta_l\lambda\mathbb{E}\|x_t - x_t'\|$$
$$\le (1-\eta_l(\lambda+\mu))\mathbb{E}\|x_{t,k-1}^i - x_{t,k-1}^j\| + \eta_l\lambda\mathbb{E}\|x_t - x_t'\| + (2+2\beta)\eta_l(\sigma_l^2 + 2\sigma_g^2 + 2Z^2)^{\frac{1}{2}},$$

where the last inequality we use Lemma C.1. Unrolling it gives:

$$\mathbb{E}\|x_{t,K}^i - x_{t,K}^j\| \le (1-\eta_l(\lambda+\mu))^K\mathbb{E}\|x_t - x_t'\| + \frac{\lambda - \lambda(1-\eta_l(\lambda+\mu))^K}{\lambda+\mu}\mathbb{E}\|x_t - x_t'\|$$

$$\quad + \frac{1-(1-\eta_l(\lambda+\mu))^K}{\lambda+\mu}(2+2\beta)(\sigma_l^2 + 2\sigma_g^2 + 2Z^2)^{\frac{1}{2}}$$

$$= \left[\frac{\lambda}{\lambda+\mu} + \frac{\mu}{\lambda+\mu}(1-\eta_l(\lambda+\mu))^K\right]\mathbb{E}\|x_t - x_t'\|$$

$$\quad + \frac{1-(1-\eta_l(\lambda+\mu))^K}{\lambda+\mu}(2+2\beta)(\sigma_l^2 + 2\sigma_g^2 + 2Z^2)^{\frac{1}{2}}.$$

We define $\zeta = \frac{1-(1-\eta_l(\lambda+\mu))^K}{\lambda+\mu}$. Since $x_{t+1} = x_t + \frac{1}{s}\sum_{i\in\mathcal{S}}(x_{t,K}^i - x_{t,0}^i)$, we have

$$\mathbb{E}\|x_{t+1} - x'_{t+1}\| = \frac{1}{s}\mathbb{E}\|\sum_{i\in\mathcal{S}} x_{t,K}^i - \sum_{j\in\mathcal{S}'} x_{t,K}^j\|$$

$$\leq (1-\Gamma)\mathbb{E}\|x_t - x'_t\| + \frac{2+2\beta}{\lambda+\mu}(\sigma_l^2 + 2\sigma_g^2 + 2Z^2)^{\frac{1}{2}},$$

where $\Gamma = 1 - \left[\frac{\lambda}{\lambda+\mu} + \frac{\mu}{\lambda+\mu}(1 - \eta_l(\lambda+\mu))^K\right] \in (0,1)$. Note that $x_0 = x'_0$, then unrolling it gives

$$\mathbb{E}\|x_T - x'_T\| \leq \frac{1-(1-\Gamma)^T}{\Gamma}\frac{2+2\beta}{\lambda+\mu}(\sigma_l^2 + 2\sigma_g^2 + 2Z^2)^{\frac{1}{2}}.$$

$\square$

# D CONVERGENCE OF PROPOSED ALGORITHM

## D.1 PRELIMINARY LEMMAS

We will use the following foundational lemma for our proof.

**Lemma D.1.** *For $v_1, v_2 \in \mathbb{R}^d$, we have*

$$\|v_1 + v_2\|^2 \leq (1+a)\|v_1\|^2 + (1+\frac{1}{a})\|v_2\|^2.$$

**Lemma D.2.** *For $L$-smooth function $f$, and $x, y$ in its domain, the following is true:*

$$f(y) \leq f(x) + \langle\nabla f(x), y - x\rangle + \frac{L}{2}\|x - y\|^2.$$

**Lemma D.3.** *For independent, mean $0$ random variables $X_1, ..., X_n$, we have*

$$\mathbb{E}[\|X_1 + ... + X_n\|^2] = \mathbb{E}[\|X_1\|^2 + ... + \|X_n\|^2]. \tag{8}$$

**Lemma D.4.** *Given vector $X_1, \cdots, X_m \in \mathbb{R}^d$, if we sample $\mathcal{S} \subset \{1, \cdots, m\}$ uniformly randomly such that $|\mathcal{S}| = s$, then it holds that*

$$\mathbb{E}\left\|\frac{1}{s}\sum_{i\in\mathcal{S}} X_i\right\|^2 = \frac{m-s}{ms(m-1)}\sum_{i\in[M]}\|X_i\|^2 + \frac{s-1}{ms(m-1)}\left\|\sum_{i\in[M]} X_i\right\|^2$$

*Proof.* Let $\mathbb{I}\{i \in \mathcal{S}\}$ be the indicator for the event $i \in \mathcal{S}$, we prove this lemma by calculating as follows:

$$\mathbb{E}\left\|\frac{1}{s}\sum_{i\in\mathcal{S}} X_i\right\|^2 = \mathbb{E}\left\|\frac{1}{s}\sum_{i\in[M]} X_i\mathbb{I}\{i \in \mathcal{S}\}\right\|^2$$

$$= \frac{1}{s^2}\mathbb{E}\left(\sum_i \|X_i\|^2\mathbb{I}\{i \in \mathcal{S}\} + 2\sum_{i<j}\langle X_i, X_j\rangle\mathbb{I}\{i, j \in \mathcal{S}\}\right)$$

$$= \frac{1}{sm}\sum_{i\in[M]}\|X_i\|^2 + \frac{1}{s^2}\frac{s(s-1)}{m(m-1)}2\sum_{i<j}\langle X_i, X_j\rangle$$

$$= \frac{1}{sm}\sum_{i\in[M]}\|X_i\|^2 + \frac{1}{s^2}\frac{s(s-1)}{m(m-1)}\left(\left\|\sum_{i\in[M]} X_i\right\|^2 - \sum_{i\in[M]}\|X_i\|^2\right)$$

$$= \frac{m-s}{sm(m-1)}\sum_{i=1}^m\|X_i\|^2 + \frac{s-1}{sm(m-1)}\left\|\sum_{i=1}^m X_i\right\|^2.$$

$\square$

### D.2 Convergence of FedSAGD

In this subsection, we present the proofs for the FedSAGD algorithm. We denote $\nabla \hat{F}(x)$ as $\nabla F(x) + \mu x$, it can be derived that $\nabla \hat{F}$ also satisfies the three assumptions.

**Lemma D.5.** *For $x_{t,k}^i \in R^d$, $\forall i \in \mathcal{S}_t$, we denote $\delta_{t,k}^i = x_{t,k}^i - x_{t,k-1}^i$ with setting $\delta_{t,0}^i = 0$, and $\Delta_{t,k}^i = \sum_{\tau=0}^k \delta_{t,\tau}^i = x_{t,k}^i - x_{t,0}^i$. Under the local update rule in Algorithm 1, we have:*

$$\Delta_{t,K}^i = -\eta_l \sum_{k=0}^{K-1} (1 - \eta_l \lambda)^{K-1-k} \hat{g}_{t,k}^i - \beta \frac{1 - (1 - \eta_l \lambda)^K}{\lambda} v_t.$$

*Proof.* According to the update rule, we have:

$$
\begin{aligned}
\delta_{t,k}^i &= \Delta_{t,k}^i - \Delta_{t,k-1}^i \\
&= x_{t,k}^i - x_{t,k-1}^i \\
&= -\eta_l (\beta v_t + \hat{g}_{t,k}^i) - \eta_l \lambda (x_{t,k-1}^i - x_{t,0}^i) \\
&= -\eta_l (\beta v_t + \hat{g}_{t,k}^i) - \eta_l \lambda \Delta_{t,k-1}^i.
\end{aligned}
$$

Then we can obtain

$$
\begin{aligned}
\Delta_{t,k}^i &= (1 - \eta_l \lambda) \Delta_{t,k-1}^i - \eta_l (\beta v_t + \hat{g}_{t,k-1}^i) \\
&= -\eta_l \sum_{\tau=0}^{k-1} (1 - \eta_l \lambda)^{k-1-\tau} (\beta v_t + \hat{g}_{t,\tau-1}^i) \\
&= -\eta_l \sum_{\tau=0}^{k-1} (1 - \eta_l \lambda)^{k-1-\tau} \hat{g}_{t,\tau-1}^i - \beta \frac{1 - (1 - \eta_l \lambda)^k}{\lambda} v_t.
\end{aligned}
$$

Let $\alpha_k = (1 - \eta_l \lambda)^{K-1-k}$ and $\alpha = \frac{1 - (1 - \eta_l \lambda)^K}{\eta_l \lambda} = \sum_{k=0}^{K-1} (1 - \eta_l \lambda)^{K-1-k}$, so we have

$$\Delta_{t,K}^i = -\eta_l \sum_{k=0}^{K-1} \alpha_k \hat{g}_{t,k-1}^i - \alpha \beta \eta_l v_t,$$

$\square$

**Lemma D.6.** *Under the update rule of Algorithm 1, we have*

$$v_{t+1} = \frac{\gamma \beta}{(1 + \beta)} v_t + \frac{1}{(1 + \beta) s K} \sum_{i \in \mathcal{S}_t} G_t^i,$$

*where $\gamma = (1 + \frac{\alpha}{K})$ and $G_t^i = \sum_{k=0}^{K-1} \alpha_k \hat{g}_{t,k}^i$.*

*Proof.*

$$v_{t+1} = \frac{\beta}{1+\beta}v_t - \frac{\Delta x_t}{(1+\beta)\eta_l K}$$

$$= \frac{\beta}{1+\beta}v_t - \frac{1}{(1+\beta)\eta_l s K}\sum_{i\in\mathcal{S}_t}\Delta x_t^i$$

$$= \frac{\beta}{1+\beta}v_t - \frac{1}{(1+\beta)\eta_l s K}\sum_{i\in\mathcal{S}_t}\Delta_{t,K}^i$$

$$= \frac{\beta}{1+\beta}v_t + \frac{1}{(1+\beta)s K}\sum_{i\in\mathcal{S}_t}\left(\sum_{k=0}^{K-1}\alpha_k\hat{g}_{t,k}^i + \alpha\beta v_t\right)$$

$$= \frac{\beta}{(1+\beta)}\left(1+\frac{\alpha}{K}\right)v_t + \frac{1}{(1+\beta)s K}\sum_{i\in\mathcal{S}_t}\sum_{k=0}^{K-1}\alpha_k\hat{g}_{t,k}^i$$

$$= \frac{\gamma\beta}{(1+\beta)}v_t + \frac{1}{(1+\beta)s K}\sum_{i\in\mathcal{S}_t}G_t^i.$$

$\square$

Consider the auxiliary sequence $z_t$, given by

$$z_t = \frac{1+\beta}{1-\alpha'\beta}x_t - \frac{\gamma\beta}{1-\alpha'\beta}x_{t-1} + \frac{\beta\eta\eta_l}{(1-\alpha'\beta)s}\sum_{i\in\mathcal{S}_{t-1}}G_{t-1}^i,$$

where $\alpha' = \frac{\alpha}{K}$.

**Lemma D.7.** *Consider the sequence $z_t$, we have*

$$z_{t+1} - z_t = -\frac{\eta\eta_l}{(1-\alpha'\beta)s}\sum_{i\in\mathcal{S}_t}G_t^i.$$

*Proof.*

$$z_{t+1} - z_t$$

$$= \frac{1+\beta}{1-\alpha'\beta}(x_{t+1}-x_t) - \frac{\gamma\beta}{1-\alpha'\beta}(x_t-x_{t-1}) + \frac{\beta\eta\eta_l}{(1-\alpha'\beta)s}\left(\sum_{i\in\mathcal{S}_t}G_t^i - \sum_{i\in\mathcal{S}_{t-1}}G_{t-1}^i\right)$$

$$= -\frac{(1+\beta)\eta\eta_l}{(1-\alpha'\beta)s}\sum_{i\in\mathcal{S}_t}(G_t^i+\alpha\beta v_t) + \frac{\gamma\beta\eta\eta_l}{(1-\alpha'\beta)s}\sum_{i\in\mathcal{S}_{t-1}}(G_{t-1}^i+\alpha\beta v_{t-1})$$

$$\quad + \frac{\beta\eta\eta_l}{(1-\alpha'\beta)s}\left(\sum_{i\in\mathcal{S}_t}G_t^i - \sum_{i\in\mathcal{S}_{t-1}}G_{t-1}^i\right)$$

$$= -\frac{\eta\eta_l}{(1-\alpha'\beta)s}\sum_{i\in\mathcal{S}_t}G_t^i.$$

the last equality follow from Lemma D.6. Besides,

$$\mathbb{E}_t\|z_{t+1}-z_t\|^2 = \mathbb{E}_t\left\|-\frac{\eta\eta_l}{(1-\alpha'\beta)s}\sum_{i\in\mathcal{S}_t}G_t^i\right\|^2$$

$$= \frac{\eta^2\eta_l^2}{(1-\alpha'\beta)^2s^2}\mathbb{E}_t\left\|\sum_{i\in\mathcal{S}_t}\sum_{k=0}^{K-1}\alpha_k\hat{g}_{t,k}^i\right\|^2$$

$$\leq \frac{\eta^2\eta_l^2}{(1-\alpha'\beta)^2s^2}\mathbb{E}_t\left\|\sum_{i\in\mathcal{S}_t}\bar{G}_t^i\right\|^2 + \frac{\alpha\eta^2\eta_l^2}{(1-\alpha'\beta)^2s}\sigma_l^2.$$

where we denote $\bar{G}_t^i := \sum_{k=0}^{K-1}\alpha_k\nabla\hat{F}_i(x_{t,k}^i)$. The inequality is based on Assumption 3.2 and Lemma D.3. $\square$

**Lemma D.8.** *Consider the $z_t$ and $x_t$, then we have*

$$z_t - x_t = -\frac{\alpha(\beta + \beta^2)\eta\eta_l}{(1 - \alpha'\beta)}v_t.$$

*Proof.*

$$z_t - x_t = \frac{\gamma\beta}{1 - \alpha'\beta}(x_t - x_{t-1}) + \frac{\beta\eta\eta_l}{(1 - \alpha'\beta)s}\sum_{i\in\mathcal{S}_{t-1}}G_{t-1}^i$$

$$= -\frac{\gamma\beta\eta\eta_l}{(1 - \alpha'\beta)s}\Big(\sum_{i\in\mathcal{S}_{t-1}}G_{t-1}^i + \alpha\beta s v_{t-1}\Big) + \frac{\beta\eta\eta_l}{(1 - \alpha'\beta)s}\sum_{i\in\mathcal{S}_{t-1}}G_{t-1}^i$$

$$= -\frac{\alpha(\beta + \beta^2)\eta\eta_l}{(1 - \alpha'\beta)}v_t.$$

$\square$

**Lemma D.9.** *Consider the global momentum sequence $v_t$ and summing over $t$ from $0$ to $T-1$, then we have*

$$\sum_{k=0}^{T-1}\mathbb{E}_t\|v_t\|^2 \le \frac{1}{(1 - \alpha'\beta)^2}\sum_{t=0}^{T-1}\mathbb{E}_t\left\|\frac{1}{sK}\sum_{i\in\mathcal{S}_t}\bar{G}_t^i\right\|^2 + \frac{T\alpha^2}{(1 - \alpha'\beta)^2 sK^2}\sigma_l^2.$$

*Proof.*

$$\mathbb{E}_t\|v_t\|^2$$

$$= \mathbb{E}_t\left\|\frac{\gamma\beta}{1+\beta}v_{t-1} + \frac{1}{(1+\beta)sK}\sum_{i\in\mathcal{S}_{t-1}}\sum_{k=0}^{K-1}\alpha_k\hat{g}_{t-1,k}^i\right\|^2$$

$$= \mathbb{E}_t\left\|\frac{1}{(1+\beta)sK}\sum_{\tau=0}^{t-1}(\frac{\gamma\beta}{1+\beta})^{t-1-\tau}\sum_{i\in\mathcal{S}_\tau}G_\tau^i\right\|^2$$

$$\le \mathbb{E}_t\left\|\frac{1}{(1+\beta)sK}\sum_{\tau=0}^{t-1}(\frac{\gamma\beta}{1+\beta})^{t-1-\tau}\sum_{i\in\mathcal{S}_\tau}\bar{G}_\tau^i\right\|^2 + \frac{\alpha^2[1 - (\frac{\gamma\beta}{1+\beta})^t]^2}{(1 - \alpha'\beta)^2 sK^2}\sigma_l^2$$

$$\overset{(a)}{=} p_t^2\mathbb{E}_t\left\|\frac{1}{(1+\beta)sK}\sum_{\tau=0}^{t-1}p_t^{-1}(\frac{\gamma\beta}{1+\beta})^{t-1-\tau}\sum_{i\in\mathcal{S}_\tau}\bar{G}_\tau^i\right\|^2 + \frac{\alpha^2[1 - (\frac{\gamma\beta}{1+\beta})^t]^2}{(1 - \alpha'\beta)^2 sK^2}\sigma_l^2$$

$$\le \frac{p_t^2}{(1+\beta)^2}\sum_{\tau=0}^{t-1}p_t^{-1}(\frac{\gamma\beta}{1+\beta})^{t-1-\tau}\mathbb{E}_t\left\|\frac{1}{sK}\sum_{i\in\mathcal{S}_\tau}\bar{G}_\tau^i\right\|^2 + \frac{\alpha^2[1 - (\frac{\gamma\beta}{1+\beta})^t]^2}{(1 - \alpha'\beta)^2 sK^2}\sigma_l^2 \tag{9}$$

$$= \frac{1 - (\frac{\gamma\beta}{1+\beta})^t}{(1+\beta)(1 - \alpha'\beta)}\sum_{\tau=0}^{t-1}(\frac{\gamma\beta}{1+\beta})^{t-1-\tau}\mathbb{E}_t\left\|\frac{1}{sK}\sum_{i\in\mathcal{S}_\tau}\bar{G}_\tau^i\right\|^2 + \frac{\alpha^2[1 - (\frac{\gamma\beta}{1+\beta})^t]^2}{(1 - \alpha'\beta)^2 sK^2}\sigma_l^2$$

$$\le \frac{1}{(1+\beta)(1 - \alpha'\beta)}\sum_{\tau=0}^{t-1}(\frac{\gamma\beta}{1+\beta})^{t-1-\tau}\mathbb{E}_t\left\|\frac{1}{sK}\sum_{i\in\mathcal{S}_\tau}\bar{G}_\tau^i\right\|^2 + \frac{\alpha^2}{(1 - \alpha'\beta)^2 sK^2}\sigma_l^2.$$

The first inequality is based on Assumption 3.2 and Lemma D.3. In $(a)$, $p_t = \sum_{\tau=0}^{t-1} (\frac{\gamma\beta}{1+\beta})^{t-1-\tau}$.
Now we sum equation 9 over $t$ from 0 to $T-1$:

$$\sum_{t=0}^{T-1} \mathbb{E}_t \|v_t\|^2$$

$$\leq \frac{1}{(1+\beta)(1-\alpha'\beta)} \sum_{t=0}^{T-1} \sum_{\tau=0}^{t-1} (\frac{\gamma\beta}{1+\beta})^{t-1-\tau} \mathbb{E}_t \left\| \frac{1}{sK} \sum_{i\in\mathcal{S}_\tau} \bar{G}_\tau^i \right\|^2 + \frac{T\alpha^2}{(1-\alpha'\beta)^2 sK^2} \sigma_l^2$$

$$= \frac{1}{(1+\beta)(1-\alpha'\beta)} \sum_{\tau=0}^{T-2} \mathbb{E}_t \left\| \frac{1}{sK} \sum_{i\in\mathcal{S}_\tau} \bar{G}_\tau^i \right\|^2 \sum_{s=\tau+1}^{T-1} (\frac{\gamma\beta}{1+\beta})^{s-1-\tau} + \frac{T\alpha^2}{(1-\alpha'\beta)^2 sK^2} \sigma_l^2$$

$$\leq \frac{1}{(1-\alpha'\beta)^2} \sum_{t=0}^{T-1} \mathbb{E}_t \left\| \frac{1}{sK} \sum_{i\in\mathcal{S}_t} \bar{G}_t^i \right\|^2 + \frac{T\alpha^2}{(1-\alpha'\beta)^2 sK^2} \sigma_l^2.$$

$\square$

**Lemma D.10.** *Suppose the local learning rate satisfies $\eta_l \leq \frac{1}{KL}$, we can bound the client drift by:*

$$\frac{1}{m} \sum_{i\in[M]} \mathbb{E}_t \|x_{t,k}^i - x_t\|^2 \leq 36\eta_l^2 K^2 \mathbb{E}_t \|\nabla\hat{f}(x_t)\|^2 + 36\eta_l^2 K^2 \sigma_g^2 + 3\eta_l^2 K \sigma_l^2 + 18\beta^2 \eta_l^2 K^2 \mathbb{E}_t \|v_t\|^2.$$

*Proof.*

$$\mathbb{E}_t \|x_{t,k}^i - x_t\|^2$$
$$= \mathbb{E}_t \|x_{t,k-1}^i - x_t - \eta_l\lambda(x_{t,k-1}^i - x_t) - \eta_l \hat{g}_{t,k-1}^i - \beta\eta_l v_t\|^2$$
$$= \mathbb{E}_t \|(1-\eta_l\lambda)(x_{t,k-1}^i - x_t) - \eta_l \hat{g}_{t,k-1}^i - \beta\eta_l v_t\|^2$$
$$\leq \mathbb{E}_t \|(1-\eta_l\lambda)(x_{t,k-1}^i - x_t) - \eta_l \nabla\hat{F}_i(x_{t,k-1}^i) - \beta\eta_l v_t\|^2 + \eta_l^2 \sigma_l^2$$
$$\leq \mathbb{E}_t [(1+a)\|(1-\eta_l\lambda)(x_{t,k-1}^i - x_t)\|^2] + \eta_l^2 \sigma_l^2 + \mathbb{E}_t [(1+\frac{1}{a})\eta_l^2 \|\nabla\hat{F}_i(x_{t,k-1}^i) + \beta v_t\|^2].$$

The first inequality we use Lemma D.3 and the last inequality we use Assumption 3.1 and Lemma D.1. We further bound the third term as

$$\mathbb{E}_t \|\nabla\hat{F}_i(x_{t,k-1}^i) + \beta v_t\|^2 = \mathbb{E}\|\nabla\hat{F}_i(x_{t,k-1}^i) - \nabla\hat{F}_i(x_t) + \nabla\hat{F}_i(x_t) + +\beta v_t\|^2$$
$$\leq \mathbb{E}(3\|\nabla\hat{F}_i(x_{t,k-1}^i) - \nabla\hat{F}_i(x_t)\|^2 + 3\|\nabla\hat{F}_i(x_t)\|^2 + 3\beta^2 v_t)$$
$$\leq 3L^2 \mathbb{E}\|x_{t,k-1}^i - x_t\|^2 + 6\mathbb{E}\|\nabla\hat{f}(x_t)\|^2 + 6\sigma_g^2 + 3\beta^2 \mathbb{E}\|v_t\|^2.$$

Hence, we have

$$\mathbb{E}_t \|x_{t,k}^i - x_t\|^2$$
$$\leq [(1-\eta_l\lambda)^2(1+a) + 6(1+\frac{1}{a})\eta_l^2 L^2]\mathbb{E}_t \|x_{t,k-1}^i - x_t\|^2 + (1+\frac{1}{a})6\eta_l^2 (\mathbb{E}_t\|\nabla\hat{f}(x_t)\|^2 + \sigma_g^2)$$
$$+ \eta_l^2 \sigma_l^2 + (1+\frac{1}{a})3\beta^2 \eta_l^2 \mathbb{E}_t \|v_t\|^2.$$

For $K=1$, take $a=1$ and the lemma holds. Suppose that $K\geq 2$ thereafter then take $a = \frac{1}{2K-1}$. It
follows from $\eta_l \leq \frac{1}{15LK}$ that

$$\mathbb{E}_t \|x_{t,k}^i - x_t\|^2$$
$$\leq (1+\frac{1}{K-1})\mathbb{E}_t \|x_{t,k-1}^i - x_t\|^2 + 12\eta_l^2 K \mathbb{E}_t \|\nabla\hat{f}(x_t)\|^2 + 12\eta_l^2 K \sigma_g^2 + 6\beta^2 \eta_l^2 K \mathbb{E}_t \|v_t\|^2$$
$$+ \eta_l^2 \sigma_l^2.$$

Unrolling the recursion, noting that $\mathbb{E}_t \|x_{t,0}^i - x_t\|^2 = 0$ and $(k-1)[(1 + \frac{1}{k-1})^k - 1] \leq 3k$ for $k \geq 2$, we obtain

$$\frac{1}{m} \sum_{i \in [M]} \mathbb{E}_t \|x_{t,k}^i - x_t\|^2$$

$$\leq \sum_{k=0}^{K-1} (1 + \frac{1}{K-1})[12\eta_l^2 \mathbb{E}_t \|\nabla \hat{f}(x_t)\|^2 + 12\eta_l^2 K \sigma_g^2 + \eta_l^2 \sigma_l^2 + 6\beta^2 \eta_l^2 K \mathbb{E}_t \|v_t\|^2]$$

$$\leq 36\eta_l^2 K^2 \mathbb{E}_t \|\nabla \hat{f}(x_t)\|^2 + 36\eta_l^2 K^2 \sigma_g^2 + 3\eta_l^2 K \sigma_l^2 + 18\beta^2 \eta_l^2 K^2 \mathbb{E}_t \|v_t\|^2.$$

□

**Lemma D.11.** *Suppose that the local learning rate satisfies $n_l \leq \frac{1}{15KL}$, we can bound the sum of gradients by:*

$$\frac{\kappa_1}{mK} \sum_{t=0}^{T-1} \sum_{i \in [M]} \mathbb{E}_t \left\| \sum_{k=0}^{K-1} \alpha_k \nabla \hat{F}_i(x_{t,k}^i) \right\|^2$$

$$\leq (72\alpha^2 \eta_l^2 L^2 K + \frac{4\alpha^2}{K}) \sum_{t=0}^{T-1} \mathbb{E}_t \|\nabla \hat{f}(x_t)\|^2 + (72\alpha^2 \eta_l^2 L^2 K + \frac{4\alpha^2}{K}) T \sigma_g^2 + 6\alpha^2 \eta_l^2 L^2 T \sigma_l^2$$

$$+ T \frac{36\alpha^4 \beta^2 \eta_l^2 L^2}{(1 - \alpha'\beta) s K} \sigma_l^2.$$

*Proof.*

$$\frac{1}{mK} \sum_{i \in [M]} \mathbb{E}_t \left\| \sum_{k=0}^{K-1} \alpha_k \nabla \hat{F}_i(x_{t,k}^i) \right\|^2 = \frac{\alpha^2}{mK} \sum_{i \in [M]} \mathbb{E}_t \left\| \sum_{k=0}^{K-1} \frac{\alpha_k}{\alpha} \nabla \hat{F}_i(x_{t,k}^i) \right\|^2$$

$$\leq \frac{\alpha}{mK} \sum_{i \in [M]} \sum_{k=0}^{K-1} \alpha_k \mathbb{E}_t \|\nabla \hat{F}_i(x_{t,k}^i)\|^2$$

For all $i \in [M]$, we have

$$\sum_{k=0}^{K-1} \alpha_k \mathbb{E}_t \|\nabla \hat{F}_i(x_{t,k}^i)\|^2$$

$$= \sum_{k=0}^{K-1} \alpha_k \mathbb{E}_t \|\nabla \hat{F}_i(x_{t,k}^i) - \nabla \hat{F}_i(x_t) + \nabla \hat{F}_i(x_t)\|^2$$

$$\leq 2 \sum_{k=0}^{K-1} \alpha_k \mathbb{E}_t [\|\nabla \hat{F}_i(x_{t,k}^i) - \nabla \hat{F}_i(x_t)\|^2 + \|\nabla \hat{F}_i(x_t) - \nabla \hat{f}(x_t) + \nabla \hat{f}(x_t)\|^2]$$

$$\leq 2L^2 \sum_{k=0}^{K-1} \alpha_k \mathbb{E}_t \|x_{t,k}^i - x_t\|^2 + 4\alpha \sigma_g^2 + 4\alpha \mathbb{E}_t \|\nabla \hat{f}(x_t)\|^2$$

$$\leq (72\alpha \eta_l^2 L^2 K^2 + 4\alpha) \mathbb{E}_t \|\nabla \hat{f}(x_t)\|^2 + (72\alpha \eta_l^2 L^2 K^2 + 4\alpha) \sigma_g^2 + 6\alpha \eta_l^2 L^2 K \sigma_l^2$$

$$+ 36\alpha \beta^2 \eta_l^2 L^2 K^2 \mathbb{E}_t \|v_t\|^2.$$

Substituting it back into above inequality we get

$$\frac{1}{mK}\sum_{i\in[M]}\mathbb{E}_t\left\|\sum_{k=0}^{K-1}\alpha_k\nabla\hat{F}_i(x_{t,k}^i)\right\|^2$$

$$\leq [(72\alpha^2\eta_l^2L^2K+\frac{4\alpha^2}{K})\mathbb{E}_t\|\nabla\hat{f}(x_t)\|^2+(72\alpha^2\eta_l^2L^2K+\frac{4\alpha^2}{K})\sigma_g^2+6\alpha^2\eta_l^2L^2\sigma_l^2$$

$$+36\alpha^2\beta^2\eta_l^2L^2K\mathbb{E}_t\|v_t\|^2]$$

We sum the above inequality by using weight 1, we get

$$\frac{1}{mK}\sum_{t=0}^{T-1}\sum_{i\in[M]}\mathbb{E}_t\left\|\sum_{k=0}^{K-1}\alpha_k\nabla\hat{F}_i(x_{t,k}^i)\right\|^2$$

$$\leq (72\alpha^2\eta_l^2L^2K+\frac{4\alpha^2}{K})\sum_{t=0}^{T-1}\mathbb{E}_t\|\nabla\hat{f}(x_t)\|^2+(72\alpha^2\eta_l^2L^2K+\frac{4\alpha^2}{K})T\sigma_g^2+6\alpha^2\eta_l^2L^2T\sigma_l^2$$

$$+36\alpha^2\beta^2\eta_l^2L^2K\sum_{t=0}^{T-1}\mathbb{E}_t\|v_t\|^2$$

$$\leq (72\alpha^2\eta_l^2L^2K+\frac{4\alpha^2}{K})\sum_{t=0}^{T-1}\mathbb{E}_t\|\nabla\hat{f}(x_t)\|^2+(72\alpha^2\eta_l^2L^2K+\frac{4\alpha^2}{K})T\sigma_g^2+6\alpha^2\eta_l^2L^2T\sigma_l^2$$

$$+\frac{36\alpha^2\beta^2\eta_l^2L^2K}{(1-\alpha'\beta)^2}\sum_{t=0}^{T-1}\mathbb{E}_t\left\|\frac{1}{sK}\sum_{i\in\mathcal{S}_t}\bar{G}_t^i\right\|^2+T\frac{36\alpha^4\beta^2\eta_l^2L^2}{(1-\alpha'\beta)^2sK}\sigma_l^2.$$

We define a positive constant $\kappa_1 < 1$ such that $1-\frac{36\alpha^2\beta^2\eta_l^2L^2}{(1-\alpha'\beta)^2}\leq\kappa_1$ if $\eta_l\leq\frac{1-\alpha'\beta}{6\alpha\beta L}$. So we get

$$\frac{\kappa_1}{mK}\sum_{t=0}^{T-1}\sum_{i\in[M]}\mathbb{E}_t\left\|\sum_{k=0}^{K-1}\alpha_k\nabla\hat{F}_i(x_{t,k}^i)\right\|^2$$

$$\leq (72\alpha^2\eta_l^2L^2K+\frac{4\alpha^2}{K})\sum_{t=0}^{T-1}\mathbb{E}_t\|\nabla\hat{f}(x_t)\|^2+(72\alpha^2\eta_l^2L^2K+\frac{4\alpha^2}{K})T\sigma_g^2+6\alpha^2\eta_l^2L^2T\sigma_l^2$$

$$+T\frac{36\alpha^4\beta^2\eta_l^2L^2}{(1-\alpha'\beta)sK}\sigma_l^2.$$

$$\square$$

**Lemma D.12.** *Under Assumption 3.1-3.3 and loss function is non-convex, we define* $D_0:=\mathbb{E}(\hat{f}(z_0)-\hat{f}(z_T))$. *When the learning rate satisfies* $\eta_l\leq\frac{1}{16KL}$ *and*

$$\eta\eta_l\leq\frac{m(s-1)}{s(m-1)}min\left\{\frac{(1-\alpha'\beta)^2\sqrt{K}}{4\alpha\beta L\sqrt{3\alpha}},\frac{(1-\alpha'\beta)\eta}{6\beta L\sqrt{2\alpha}},\frac{1-\alpha'\beta}{2\alpha L}\right\}.$$

*Then the auxiliary sequence* $z_t$ *in equation equation 7 generated by executing the FedSAGD satisfies:*

$$\frac{1}{T}\sum_{t=0}^{T-1}\|\nabla f(x_t)\|\leq\mathcal{O}\left(\frac{LD_0}{\sqrt{sKT}}+\frac{\sigma_l^2}{\sqrt{sKT}}+\frac{1}{T}(\frac{\sigma_l^2}{K^2}+K\sigma_l^2+\frac{\sigma_l^2}{sK}+\frac{\sigma_g^2}{K})+\Psi_l+\Psi_g\right)$$

*where*

$$\Psi_l=(\frac{1}{T}+\frac{K}{sT})\frac{(m-s)}{m-1}(\frac{1}{T}+\frac{1}{sK^2T}+\frac{1}{\sqrt{sKT}})\sigma_l^2,$$

$$\Psi_g=(\frac{K}{T}+K)\frac{(m-s)}{m-1}(\frac{1}{T}+\frac{1}{sK^2T}+\frac{1}{\sqrt{sKT}})\sigma_g^2.$$

*Proof.* For the general non-convex case, according to the Assumptions and the smoothness of $\hat{f}$, we take the conditional expectation at round $t+1$ and expand the $\hat{f}(z_{t+1})$ as

$$\mathbb{E}_t[\hat{f}(z_{t+1})] \leq \hat{f}(z_t) + \langle \nabla \hat{f}(z_t), \mathbb{E}_t[z_{t+1} - z_t]\rangle + \frac{L}{2}\mathbb{E}_t\|z_{t+1} - z_t\|^2.$$

Note that the second term can be split into the following:

$$\langle \nabla \hat{f}(z_t), \mathbb{E}_t[z_{t+1} - z_t]\rangle = \langle \nabla \hat{f}(z_t) - \nabla \hat{f}(x_t) + \nabla \hat{f}(x_t), \mathbb{E}_t[z_{t+1} - z_t]\rangle$$
$$= \underbrace{\langle \nabla \hat{f}(z_t) - \nabla \hat{f}(x_t), \mathbb{E}_t[z_{t+1} - z_t]\rangle}_{A_1} + \underbrace{\langle \nabla \hat{f}(x_t), \mathbb{E}_t[z_{t+1} - z_t]\rangle}_{A_2}.$$

We first bound the $A_1$:

$$\langle \nabla \hat{f}(z_t) - \nabla \hat{f}(x_t), \mathbb{E}_t[z_{t+1} - z_t]\rangle$$
$$= \langle \nabla \hat{f}(z_t) - \nabla \hat{f}(x_t), -\frac{\eta\eta_l}{(1-\alpha'\beta)}\mathbb{E}_t[\frac{1}{m}\sum_{i\in[M]}G_t^i]\rangle$$
$$\langle \frac{\sqrt{2\eta\eta_l K}}{\sqrt{1-\alpha'\beta}}\nabla \hat{f}(z_t) - \nabla \hat{f}(x_t), -\frac{\sqrt{\eta\eta_l}}{\sqrt{2(1-\alpha'\beta)K}}\mathbb{E}_t[\frac{1}{m}\sum_{i\in[M]}\bar{G}_t^i]\rangle$$
$$\leq \frac{\eta\eta_l K}{1-\alpha'\beta}\|\nabla \hat{f}(z_t) - \nabla \hat{f}(x_t)\|^2 + \frac{\eta\eta_l}{4(1-\alpha'\beta)m^2 K}\mathbb{E}_t\|\sum_{i\in[M]}\bar{G}_{t,K}^i\|^2$$
$$\leq \frac{\eta\eta_l L^2 K}{1-\alpha'\beta}\|z_t - x_t\|^2 + \frac{\eta\eta_l}{4(1-\alpha'\beta)m^2 K}\mathbb{E}_t\|\sum_{i\in[M]}\bar{G}_{t,K}^i\|^2.$$

Then the $A_2$ can be bounded as:

$$\langle \nabla \hat{f}(x_t), \mathbb{E}_t[z_{t+1} - z_t]\rangle$$
$$= \langle \nabla \hat{f}(x_t), -\frac{\eta\eta_l}{1-\alpha'\beta}\mathbb{E}_t[\frac{1}{m}\sum_{i\in[M]}\bar{G}_t^i]\rangle$$
$$= \langle \frac{1}{1-\alpha'\beta}\nabla \hat{f}(x_t), -\eta\eta_l\mathbb{E}_t[\frac{1}{m}\sum_{i\in[M]}\bar{G}_t^i - \alpha\nabla \hat{f}(x_t) + \alpha\nabla \hat{f}(x_t)]\rangle$$
$$= -\frac{\eta\eta_l\alpha}{1-\alpha'\beta}\|\nabla \hat{f}(x_t)\|^2 + \langle \frac{\sqrt{\eta\eta_l\alpha}}{\sqrt{1-\alpha'\beta}}\nabla \hat{f}(x_t), -\frac{\sqrt{\eta\eta_l}}{\sqrt{(1-\alpha'\beta)\alpha}}\mathbb{E}_t[\frac{1}{m}\sum_{i\in[M]}\bar{G}_t^i - \alpha\nabla \hat{f}(x_t)]\rangle$$
$$= -\frac{\eta\eta_l\alpha}{2(1-\alpha'\beta)}\|\nabla \hat{f}(x_t)\|^2 + \frac{\eta\eta_l}{2(1-\alpha'\beta)\alpha m^2}\mathbb{E}_t(\|\sum_{i\in[M]}\bar{G}_t^i - \alpha\nabla \hat{f}(x_t)\|^2 - \|\sum_{i\in[M]}G_t^i\|^2)$$
$$= \frac{\eta\eta_l}{2(1-\alpha'\beta)\alpha m^2}\mathbb{E}_t\|\sum_{i\in[M]}\sum_{k=0}^{K-1}\alpha_k[\nabla F_i(x_{t,k}^i) - \nabla f(x_t)]\|^2 - \frac{\eta\eta_l\alpha}{2(1-\alpha'\beta)}\|\nabla f(x_t)\|^2$$
$$\quad - \frac{\eta\eta_l}{2(1-\alpha'\beta)\alpha m^2}\mathbb{E}_t\|\sum_{i\in[M]}\bar{G}_t^i\|^2$$
$$\leq \frac{\eta\eta_l L^2}{2(1-\alpha'\beta)\alpha m}\sum_{i\in[M]}\sum_{k=0}^{K-1}\alpha_k\mathbb{E}_t[\|x_{t,k}^i - x_t\|^2] - \frac{\eta\eta_l\alpha}{2(1-\alpha'\beta)}\|\nabla f(x_t)\|^2$$
$$\quad - \frac{\eta\eta_l}{2(1-\alpha'\beta)\alpha m^2}\mathbb{E}_t\|\sum_{i\in[M]}\bar{G}_t^i\|^2$$
$$\leq \frac{\eta\eta_l L^2}{(1-\alpha'\beta)}(36\eta_l^2 K^2\mathbb{E}_t\|\nabla \hat{f}(x_t)\|^2 + 36\eta_l^2 K^2\sigma_g^2 + 3\eta_l^2 K\sigma_l^2 + 18\beta^2\eta_l^2 K^2\mathbb{E}_t\|v_t\|^2)$$
$$\quad - \frac{\eta\eta_l\alpha}{2(1-\alpha'\beta)}\|\nabla \hat{f}(x_t)\|^2 - \frac{\eta\eta_l}{2(1-\alpha'\beta)\alpha m^2}\mathbb{E}_t\|\sum_{i\in[M]}\bar{G}_t^i\|^2.$$

Now we have

$$\mathbb{E}_t[\hat{f}(z_{t+1})]$$

$$\leq \hat{f}(z_t) + \langle \nabla \hat{f}(z_t), \mathbb{E}_t[z_{t+1} - z_t]\rangle + \frac{L}{2}\mathbb{E}_t\|z_{t+1} - z_t\|^2$$

$$\leq \hat{f}(z_t) + \frac{\eta\eta_l L^2 K}{1 - \alpha'\beta}\|z_t - x_t\|^2 + \frac{\eta\eta_l}{4(1 - \alpha'\beta)m^2 K}\mathbb{E}_t\left\|\sum_{i\in[M]}\bar{G}_t^i\right\|^2 - \frac{\eta\eta_l\alpha}{2(1 - \alpha'\beta)}\|\nabla\hat{f}(x_t)\|^2$$

$$+ \frac{\eta\eta_l L^2}{(1 - \alpha'\beta)}(36\eta_l^2 K^2 \mathbb{E}_t\|\nabla\hat{f}(x_t)\|^2 + 36\eta_l^2 K^2\sigma_g^2 + 3\eta_l^2 K\sigma_l^2 + 18\beta^2\eta_l^2 K^2\mathbb{E}_t\|v_t\|^2)$$

$$- \frac{\eta\eta_l}{2(1 - \alpha'\beta)\alpha m^2}\mathbb{E}_t\left\|\sum_{i\in[M]}\bar{G}_t^i\right\|^2 + \frac{L}{2}\mathbb{E}_t\|z_{t+1} - z_t\|^2$$

$$\leq \hat{f}(z_t) + \left(\frac{\alpha^2\beta^2(1 + \beta)^2\eta^3\eta_l^3 L^2 K}{(1 - \alpha'\beta)^3} + \frac{18\beta^2\eta\eta_l^3 L^2 K^2}{1 - \alpha'\beta}\right)\mathbb{E}_t\|v_t\|^2 - \frac{\eta\eta_l\alpha}{2(1 - \alpha'\beta)}\mathbb{E}_t\|\nabla\hat{f}(x_t)\|^2$$

$$+ \frac{36\eta\eta_l^3 L^2 K^2}{1 - \alpha'\beta}\mathbb{E}_t\|\nabla\hat{f}(x_t)\|^2 + (\frac{3\eta\eta_l^3 L^2 K}{(1 - \alpha'\beta)} + \frac{\alpha\eta^2\eta_l^2 L}{2(1 - \alpha'\beta)^2 s})\sigma_l^2 + \frac{36\eta\eta_l^3 L^2 K^2}{1 - \alpha'\beta}\sigma_g^2$$

$$+ \frac{\eta^2\eta_l^2 L}{2(1 - \alpha'\beta)^2 s^2}\mathbb{E}_t\left\|\sum_{i\in\mathcal{S}_t}\bar{G}_t^i\right\|^2 - \frac{\eta\eta_l}{4(1 - \alpha'\beta)\alpha m^2}\mathbb{E}_t\left\|\sum_{i\in[M]}\bar{G}_t^i\right\|^2.$$

Summing from $t = 0, \cdots, T - 1$, we get

$$\mathbb{E}_t[\hat{f}(z_T)] - \hat{f}(z_0)$$

$$\leq -\left(\frac{\eta\eta_l\alpha}{2(1 - \alpha'\beta)} - \frac{36\eta\eta_l^3 L^2 K^2}{1 - \alpha'\beta}\right)\sum_{k=0}^{T-1}\mathbb{E}_t\|\nabla\hat{f}(x_t)\|^2 + T(\frac{3\eta\eta_l^3 L^2 K}{(1 - \alpha'\beta)} + \frac{\alpha\eta^2\eta_l^2 L}{2(1 - \alpha'\beta)^2 s})\sigma_l^2$$

$$+ \left(\frac{\alpha^2\beta^2(1 + \beta)^2\eta^3\eta_l^3 L^2 K}{(1 - \alpha'\beta)^3} + \frac{18\beta^2\eta\eta_l^3 L^2 K^2}{1 - \alpha'\beta}\right)\sum_{k=0}^{T-1}\mathbb{E}_t\|v_t\|^2 + T\frac{36\eta\eta_l^3 L^2 K^2}{1 - \alpha'\beta}\sigma_g^2$$

$$+ \frac{\eta^2\eta_l^2 L}{2(1 - \alpha'\beta)^2 s^2}\sum_{k=0}^{T-1}\mathbb{E}_t\left\|\sum_{i\in\mathcal{S}_t}\bar{G}_t^i\right\|^2 - \frac{\eta\eta_l}{4(1 - \alpha'\beta)\alpha m^2}\sum_{k=0}^{T-1}\mathbb{E}_t\left\|\sum_{i\in[M]}\bar{G}_t^i\right\|^2$$

$$\leq -\left(\frac{\eta\eta_l\alpha}{2(1 - \alpha'\beta)} - \frac{36\eta\eta_l^3 L^2 K^2}{1 - \alpha'\beta}\right)\sum_{k=0}^{T-1}\mathbb{E}_t\|\nabla\hat{f}(x_t)\|^2 + T(\frac{3\eta\eta_l^3 L^2 K}{(1 - \alpha'\beta)} + \frac{\alpha\eta^2\eta_l^2 L}{2(1 - \alpha'\beta)^2 s})\sigma_l^2$$

$$+ T\left(\frac{4\alpha^4\beta^2\eta^3\eta_l^3 L^2}{(1 - \alpha'\beta)^5 sK} + \frac{18\alpha^2\beta^2\eta\eta_l^3 L^2}{(1 - \alpha'\beta)^3 s}\right)\sigma_l^2 + T\frac{36\eta\eta_l^3 L^2 K^2}{1 - \alpha'\beta}\sigma_g^2$$

$$+ \left(\frac{4\alpha^2\beta^2\eta^3\eta_l^3 L^2 K}{(1 - \alpha'\beta)^5} + \frac{18\beta^2\eta\eta_l^3 L^2 K^2}{(1 - \alpha'\beta)^3} + \frac{\eta^2\eta_l^2 L K^2}{2(1 - \alpha'\beta)^2}\right)\sum_{k=0}^{T-1}\mathbb{E}_t\left\|\frac{1}{sK}\sum_{i\in\mathcal{S}_t}\bar{G}_t^i\right\|^2$$

$$- \frac{\eta\eta_l}{4(1 - \alpha'\beta)\alpha m^2}\sum_{k=0}^{T-1}\mathbb{E}_t\left\|\sum_{i\in[M]}\bar{G}_t^i\right\|^2.$$

Then we use Lemma D.4 and let $\eta\eta_l$ satisfies $\eta\eta_l \leq \frac{m(s-1)}{s(m-1)} min \left\{ \frac{(1-\alpha'\beta)^2\sqrt{K}}{4\alpha\beta L\sqrt{3\alpha}}, \frac{(1-\alpha'\beta)\eta}{6\beta L\sqrt{2\alpha}}, \frac{1-\alpha'\beta}{2\alpha L} \right\}$, we have

$$\mathbb{E}_t[\hat{f}(z_T)] - \hat{f}(z_0)$$

$$\leq -\left(\frac{\eta\eta_l\alpha}{2(1-\alpha'\beta)} - \frac{36\eta\eta_l^3 L^2 K^2}{1-\alpha'\beta}\right)\sum_{k=0}^{T-1}\mathbb{E}_t\|\nabla\hat{f}(x_t)\|^2 + T\left(\frac{3\eta\eta_l^3 L^2 K}{(1-\alpha'\beta)} + \frac{\alpha\eta^2\eta_l^2 L}{2(1-\alpha'\beta)^2 s}\right)\sigma_l^2$$

$$+ T\left(\frac{4\alpha^4\beta^2\eta^3\eta_l^3 L^2}{(1-\alpha'\beta)^5 sK} + \frac{18\alpha^2\beta^2\eta\eta_l^3 L^2}{(1-\alpha'\beta)^3 s}\right)\sigma_l^2 + T\frac{36\eta\eta_l^3 L^2 K^2}{1-\alpha'\beta}\sigma_g^2$$

$$+ \frac{(m-s)}{ms(m-1)}\left(\frac{4\alpha^2\beta^2\eta^3\eta_l^3 L^2}{(1-\alpha'\beta)^5 K} + \frac{18\beta^2\eta\eta_l^3 L^2}{(1-\alpha'\beta)^3} + \frac{\eta^2\eta_l^2 L}{2(1-\alpha'\beta)^2}\right)\sum_{k=0}^{T-1}\sum_{i\in[M]}\mathbb{E}_t\|\bar{G}_t^i\|^2$$

$$\leq T\left(\frac{3\eta\eta_l^3 L^2 K}{(1-\alpha'\beta)} + \frac{\alpha\eta^2\eta_l^2 L}{2(1-\alpha'\beta)^2 s} + \frac{4\alpha^4\beta^2\eta^3\eta_l^3 L^2}{(1-\alpha'\beta)^5 sK} + \frac{18\alpha^2\beta^2\eta\eta_l^3 L^2}{(1-\alpha'\beta)^3 s} + (6\alpha^2\eta_l^2 L^2 K\right.$$

$$+ \frac{36\alpha^4\beta^2\eta_l^2 L^2}{(1-\alpha'\beta)s})\frac{(m-s)}{\kappa_1 s(m-1)}\left(\frac{4\alpha^2\beta^2\eta^3\eta_l^3 L^2}{(1-\alpha'\beta)^5 K} + \frac{18\beta^2\eta\eta_l^3 L^2}{(1-\alpha'\beta)^3} + \frac{\eta^2\eta_l^2 L}{2(1-\alpha'\beta)^2}\right)\right)\sigma_l^2$$

$$+ T\left(\frac{(72\alpha^2\eta_l^2 L^2 K^2 + 4\alpha^2)(m-p)}{\kappa_1 s(m-1)}\left(\frac{4\alpha^2\beta^2\eta^3\eta_l^3 L^2}{(1-\alpha'\beta)^5 K} + \frac{18\beta^2\eta\eta_l^3 L^2}{(1-\alpha'\beta)^3} + \frac{\eta^2\eta_l^2 L}{2(1-\alpha'\beta)^2}\right)\right.$$

$$+ \frac{36\eta\eta_l^3 L^2 K^2}{1-\alpha'\beta}\right)\sigma_g^2 - \sum_{k=0}^{T-1}\mathbb{E}_t\|\nabla\hat{f}(x_t)\|^2\left(\frac{\eta\eta_l\alpha}{2(1-\alpha'\beta)} - \frac{36\eta\eta_l^3 L^2 K^2}{1-\alpha'\beta}\right.$$

$$\left.- \frac{(72\alpha^2\eta_l^2 L^2 K^2 + 4\alpha^2)(m-s)}{\kappa_1 s(m-1)}\left(\frac{4\alpha^2\beta^2\eta^3\eta_l^3 L^2}{(1-\alpha'\beta)^5 K} + \frac{18\beta^2\eta\eta_l^3 L^2}{(1-\alpha'\beta)^3} + \frac{\eta^2\eta_l^2 L}{2(1-\alpha'\beta)^2}\right)\right).$$

We define a positive constant $c$ such that $0 < c \leq \frac{1}{2} - \frac{36\eta_l^2 L^2 K^2}{\alpha} - \frac{(72\alpha\eta_l^2 L^2 K^2 + 4\alpha)(m-s)}{\kappa_1 s(m-1)}\left(\frac{4\alpha^2\beta^2\eta^2\eta_l^2 L^2}{(1-\alpha'\beta)^4 K} + \frac{18\beta^2\eta_l^2 L^2}{(1-\alpha'\beta)^2} + \frac{\eta\eta_l L}{2(1-\alpha'\beta)}\right)$ and rearrange the above inequality, we have

$$\frac{1}{T}\sum_{k=0}^{T-1}\mathbb{E}_t\|\nabla\hat{f}(x_t)\|^2$$

$$\leq \left(\frac{3\eta_l^2 L^2 K}{\alpha} + \frac{\eta\eta_l L}{2(1-\alpha'\beta)s} + \frac{4\alpha^3\beta^2\eta^2\eta_l^2 L^2}{(1-\alpha'\beta)^4 sK} + \frac{18\alpha\beta^2\eta_l^2 L^2}{(1-\alpha'\beta)^2 s} + (6\alpha\eta_l^2 L^2 K\right.$$

$$+ \frac{36\alpha^3\beta^2\eta_l^2 L^2}{(1-\alpha'\beta)s})\frac{(m-s)}{\kappa_1 s(m-1)}\left(\frac{4\alpha^2\beta^2\eta^2\eta_l^2 L^2}{(1-\alpha'\beta)^4 K} + \frac{18\beta^2\eta_l^2 L^2}{(1-\alpha'\beta)^2} + \frac{\eta\eta_l L}{2(1-\alpha'\beta)}\right)\right)\frac{\sigma_l^2}{c}$$

$$+ \left(\frac{(72\alpha\eta_l^2 L^2 K^2 + 4\alpha)(m-s)}{\kappa_1 s(m-1)}\left(\frac{4\alpha^2\beta^2\eta^2\eta_l^2 L^2}{(1-\alpha'\beta)^4 K} + \frac{18\beta^2\eta_l^2 L^2}{(1-\alpha'\beta)^2} + \frac{\eta\eta_l L}{2(1-\alpha'\beta)}\right)\right.$$

$$+ \frac{36\eta_l^2 L^2 K^2}{\alpha}\right)\frac{\sigma_g^2}{c} + \frac{(1-\alpha'\beta)(\hat{f}(z_0) - \mathbb{E}_t[\hat{f}(z_T)])}{c\alpha\eta\eta_l T}$$

Then we set $\eta_l = \mathcal{O}(\frac{1}{LK\sqrt{T}})$ and $\eta = \mathcal{O}(\sqrt{sK})$, we can deduce that $\alpha = \mathcal{O}(K)$. Thus, we get:

$$\frac{1}{T}\sum_{k=0}^{T-1}\mathbb{E}_t\|\nabla\hat{f}(x_t)\|^2$$

$$\leq \frac{(1-\beta)L(\hat{f}(z_0) - \mathbb{E}_t[\hat{f}(z_T)])}{c\sqrt{sKT}} + \left(\frac{3}{K^2 T} + \frac{1}{2(1-\beta)\sqrt{sKT}} + \frac{4\beta^2 K}{(1-\beta)^4 T} + \frac{18\beta^2}{(1-\beta)^2 sKT}\right.$$

$$+ (\frac{6}{T} + \frac{36\beta^2 K}{(1-\beta)sT})\frac{(m-s)}{\kappa_1(m-1)}\left(\frac{4\beta^2}{(1-\beta)^4 T} + \frac{18\beta^2}{(1-\beta)^2 sK^2 T} + \frac{1}{2\sqrt{sKT}}\right)\right)\frac{\sigma_l^2}{c}$$

$$+ \left(\left(\frac{72K}{T} + 4K\right)\frac{(m-s)}{\kappa_1(m-1)}\left(\frac{4\beta^2}{(1-\beta)^4 T} + \frac{18\beta^2}{(1-\beta)^2 sK^2 T} + \frac{1}{2\sqrt{sKT}}\right) + \frac{36}{KT}\right)\frac{\sigma_g^2}{c}.$$

□

# E    THE USE OF LARGE LANGUAGE MODELS (LLMs)

In the preparation of this paper, a large language model (LLM) was used solely for minor text polishing and grammar corrections. The LLM did not contribute to research ideation, content generation, or any other significant aspect of the work. All content, including the final text, has been thoroughly reviewed and approved by the authors, who take full responsibility for its accuracy and originality.

