# OpenReview forum: "FedSAGD: federated learning with stable and accelerated client gradient descent"
_ICLR.cc/2026/Conference — ICLR 2026 Conference Withdrawn Submission_

### Official Review · Reviewer_46Nf · 2025-10-19

**Soundness:** 2
**Presentation:** 3
**Contribution:** 2
**Rating:** 4
**Confidence:** 3

**Summary:**

This paper aims to address the deteriorating generalization performance of FL caused by heterogeneous local data distributions and client-side overfitting. It proposes a global-update stability framework to better analyze generalization errors in FL. Additionally, the paper develops a novel FL algorithm named FedSAGD and demonstrates that this method can achieve superior stability and generalization. Finally, extensive empirical studies are conducted to verify that FedSAGD outperforms baseline methods in terms of faster convergence and better generalization.

**Strengths:**

The paper introduces and formalizes the concept of Global-Update Stability in FL, which is highly relevant for the practical setting of partial client participation.

**Weaknesses:**

1. The novelty is somewhat incremental. It appears to be a straightforward extension of traditional sample-level stability analysis to the client-level setting.
2. The form of the generalization error bound is uncommon — the effect of sample size is not clearly provided in the bound.
3. The notation and writing in the problem setup are unconventional and need improvement — for example,  key quantities (loss function, empirical and population risks) are needed to define more clearly.

**Questions:**

1. Please explain the meaning of the symbols $\gamma$, $s$, $K$, and $T$ in the abstract, so that readers unfamiliar with FL can immediately understand them.
2. In the *Related Work* section, I suggest splitting the discussion of "Generalization and stability" into two sub-sections:
   - “Generalization in centralized learning” — introduce the traditional uniform-convergence framework, algorithm-dependent generalization (e.g., algorithmic stability, PAC-Bayes), deep-learning generalization theory, out-of-distribution/domain generalization theory, etc.
   - “Generalization in federated learning” — discuss algorithmic stability and PAC-Bayes / information‐theoretic generalization for FL, federated domain generalization, unseen‐client participation scenarios, etc.

   This clearer separation will help readers situate your contributions more precisely.
3. The *Problem Formulation* part is somewhat non-standard. I recommend the following structure: first define the loss function (on a single sample), then define the local empirical risk (on the local dataset) based on that loss, then define the local population risk (on the local data distribution) which is used for the generalization analysis. I believe that explicitly define the local loss for a sample and the risks will improve the clarity of this paper.
4. Regarding Assumption 3.1: In FL contexts it is common to assume that the variance of the local stochastic gradient (when sampling mini-batches from the local training set) is bounded. More precisely, the bounded local variance refers to the variance between the mini-batch‐SGD gradient and the full local empirical risk gradient (not between the SGD gradient and the true expected risk gradient w.r.t. local data distribution).
5. Definition 3.4: What exactly does $A(\mathcal{S}^{i},\xi_j)$ denote? The notation is unclear—please provide a precise definition.
6. The generalization analysis in this paper appears unconventional. I suggest decomposing Equation (2) further to introduce the empirical risk explicitly, which will help expose how the sample size influences the generalization bound.
7. On the theoretical novelty/challenge: The concept of *Global-update stability* seems to mirror sample-level stability analysis by replacing “samples” with “clients”. Please clarify how this extension is non-trivial and what new technical difficulties you overcome.
8. Could the authors provide an intuitive explanation for *why* the proposed hybrid proximal term improves stability more than the standard FedProx regularization? A more intuitive commentary would strengthen the presentation.
9. I recommend presenting the generalization bound of your algorithm as a clearly stated theorem (rather than embedding it in the text). This helps readers locate and understand the key theoretical result.
10. For the experiments:
    - It would be beneficial to evaluate the robustness of the proposed method under feature skew (not only label skew), since real-world FL often suffers from heterogeneity in feature distributions.
    - Please report results averaged over multiple runs (e.g., mean ± standard error or provide error bars) to show statistical reliability of improvements.

---

> ### Author Response · Authors · 2025-11-20
> **Author Response (part 1 of 3)**
>
> We sincerely appreciate the reviewer for recognizing our contributions and for the constructive comments. Our point-to-point responses to concerns on Weaknesses and Questions are given below.
>
> **Q1:Please explain the meaning of the symbols $\gamma$,$s$,$K$ and $T$ in the abstract, so that readers unfamiliar with FL can immediately understand them.**
>
> Thank you for the writing suggestion. We will clarify.
>
> **Q2:In the Related Work section, I suggest splitting the discussion of "Generalization and stability" into two sub-sections.**
>
> Thank you for the suggestion. We have split the “Generalization and Stability” part of the related work into two separate sub-sections, and these updates have been reflected in the latest PDF.
>
> **Q3 & W3:The Problem Formulation part is somewhat non-standard.**
>
> Thank you for the suggestion. We have revised the definitions before the generalization analysis, and the updates are reflected in the PDF.
>
> **Q4: Assumption 3.1 is misused. In FL, the bounded local variance refers to the variance between the mini-batch‐SGD gradient and the full local empirical risk gradient (not between the SGD gradient and the true expected risk gradient w.r.t. local data distribution).**
>
> Thank you for the reminder; we will clarify. We follow the standard “bounded variance” assumption commonly adopted in FL analyses, including FedAdam, FedACG [1], and FedANAG [2]. It is true that some FL papers [3] define $F_i(x) \triangleq \frac{1}{n_i}\sum_{j=1}^{n_i} F_i(x,\xi_j),$ which explicitly writes the full local empirical risk as a finite sum. In contrast, our definition—shared by FedAdam, FedACG, and FedANAG—uses $F_i(x)=\mathbb{E}_{\xi_i\sim \mathcal{P}_i}[F_i(x,\xi_i)],$ where $\mathcal{P}_i$ denotes the local empirical distribution on client $i$. Therefore, $\nabla F_i(x)$ is still the full local empirical risk gradient, so the two formulations are equivalent in meaning.
>
> [1]Kim, Geeho, et al. Communication-efficient federated learning with accelerated client gradient. CVPR, 2024.
>
> [2]Zhang, Hao, et al. Stabilizing and accelerating federated learning on heterogeneous data with partial client participation. TPAMI 2024.
>
> [3]Li X, Huang K, Yang W, et al. On the convergence of fedavg on non-iid data. ICLR, 2020.
>
> **Q5: Definition 3.4: What exactly does $\mathcal{A}(\mathcal{S},\xi_j)$ denote?**
>
> This is a typographical error; the correct definition for Definition 3.4 should be:
>
> $\begin{align*}
> \mathop{max} \limits_{i\in[M]}\mathbb{E}_{\mathcal{A},\mathcal{S},\xi_j}\big[|F_j(\mathcal{A}(\mathcal{S}),\xi_j)-F_j(\mathcal{A}(\mathcal{S}^{(i)}),\xi_j)|\big]\leq \epsilon
> \end{align*}$
>
> **Q6&W2: The generalization analysis in this paper appears unconventional. I suggest decomposing Equation (2) further to introduce the empirical risk explicitly, which will help expose how the sample size influences the generalization bound.**
>
> Explanation of $\epsilon_{gen}$: In practical FL scenarios, client participation is partial and random, the model needs to generalize to unseen clients whose data distributions may differ significantly. To better capture this setting, we adopt the concept of semi-empirical risk from [2,4] to define , which replaces the empirical loss within clients with the expected loss on the local dataset. Traditional generalization error measures the performance of a model on unseen data, while the generalization error we define here measures the performance of the model on the data of unseen clients. To better assess the generalization error for unparticipating clients, we adopt this definition, which is also consistent with the work in [2,4].
>
> Thank you，we will decomposing Equation (2) to expected risk and semi-empirical risk. In our setting, the generalization error is defined primarily in terms of the model’s performance on the data of unseen clients, which differs from the conventional notion of generalization error. Therefore, our analysis primarily focuses on how the update rules influence stability, without considering the effect of sample size; this can be explored in future work.
>
> [4]Yuan, Honglin, et al. What Do We Mean by Generalization in Federated Learning?. ICML 2021.

---

> ### Author Response · Authors · 2025-11-20
> **Author Response (part 2 of 3)**
>
> **W1&Q7：Global-update stability looks like sample-level stability with “samples → clients”. Please explain what is actually new or difficult in this extension.**
>
> Our work is not a simple extension. The proposed stability framework is designed to reflect practical FL scenarios characterized by low client participation, data heterogeneity, and non-convex loss functions, which introduce substantial technical challenges.
>
> First, the stability analysis in [5] relies on the assumption that when a client’s data point is replaced, the new data comes from the **same distribution**. This assumption is unrealistic in practical cross-device FL, where the number of clients is large and per-round participation is sparse. When the replaced data no longer follows the same distribution—as is typical under heterogeneity—the treatment of gradient differences used in [5] breaks down, and a bounded heterogeneity term  $\sigma_g$ must be introduced to make the analysis feasible.
>
> Second, the sample-replacement stability framework in [5] fundamentally depends on the convexity of the loss function and cannot be directly extended to non-convex optimization. In particular, the construction of Lemma 2 in [5] does not hold in the non-convex setting. In our analysis, we address this issue by establishing Lemma 1, which enables the necessary inequality relaxation in the non-convex case.
>
>
> [5]Sun, Zhenyu, Xiaochun Niu, and Ermin Wei. Understanding generalization of federated learning via stability: Heterogeneity matters. AISTATS, 2024.
>
> **Q8:Could the authors provide an intuitive explanation for why the proposed hybrid proximal term improves stability more than the standard FedProx regularization? A more intuitive commentary would strengthen the presentation.**
>
> According to our derivation, the proximal term used in FedProx constrains the deviation of the local model from the global model, effectively introducing a form of weight regularization into the local update. Specifically, the model coefficient in its local update is less than 1, which leads to improved local-update stability. The formulation is as follows:
> $\begin{align*}
>     &\mathbb{E}\Vert x^i_{t,k}-x^j_{t,k} \Vert & \\
>     & \leq (1-\lambda\eta_l)\mathbb{E}\Vert x^i_{t,k-1}-x^j_{t,k-1} \Vert + 2(\sigma^2_l + 2\sigma^2_g + 2\Phi)^{\frac{1}{2}}\eta_l+\lambda\eta_l\mathbb{E}\Vert x_t-x'_t \Vert.   &\\
> \end{align*}$
>
>
> When the iteration reaches the global update, this effect disappears because the previous global model is subtracted with the same coefficient. Consequently, the update coefficient of the global model remains 1, and its global-update stability is unaffected. The corresponding expression is as follows:
> $\begin{align*}
>     \mathbb{E}\Vert x_{t+1}-x'_{t+1}\Vert
>     & \leq \mathbb{E}\Vert x_t-x'_t \Vert +\frac{2}{\lambda}(\sigma^2+2\sigma^2_g+2\Phi)^{\frac{1}{2}}[1-(1-\lambda\eta_l)^K].
> \end{align*}$
>
> The hybrid proximal term we introduce assigns asymmetric coefficients to the local model and the previous global model. This not only improves local update stability during local iterations, but also induces an additional weight-decay factor when the update reaches the global model. This acts as a form of approximate weight regularization on the global model, thereby yielding higher global update stability. The corresponding formulation is as follows:
>
> $\begin{align*}
>     \mathbb{E}\Vert x_{t+1}-x'_{t+1}\Vert \leq(1- \Gamma)\mathbb{E}\Vert x_t-x'_t\Vert+\frac{2+2\beta}{\lambda+\mu}(\sigma^2_l+2\sigma^2_g+2\Phi)^{\frac{1}{2}}.
> \end{align*}$
>
> In summary, our hybrid proximal term alters the global update by explicitly re-weighting the contribution of the previous global model. This induces an approximate form of weight regularization (or contraction) on the global model: when comparing two runs with neighboring client sets, the proximal term attenuates the discrepancy between their global updates, yielding a finite stability bound. In contrast, the proximal term in FedProx operates only on local updates and does not affect the global model. Consequently, it cannot diminish the difference between global updates when comparing two runs that differ only in their client sets.

---

> ### Author Response · Authors · 2025-11-20
> **Author Response (part 3 of 3)**
>
> **Q9: I recommend presenting the generalization bound of your algorithm as a clearly stated theorem.**
>
> There may be a misunderstanding. We compute the global-update stability bound of the algorithm, but we do not obtain a generalization bound. If the reviewer is suggesting that the stability bound of our algorithm should be stated explicitly as a theorem, we appreciate the suggestion. We state the stability result of our algorithm as Theorem 3.6. We have revised the PDF accordingly.
>
>
> **Q10: For the experiments:feature skew;results averaged over multiple runs.**
>
> (1) Thanks for this suggestion. We have added experiments under feature skew, following the experimental setup in [1]. We conduct the evaluation on the real-world LEAF dataset FEMNIST [6], which naturally contains feature-skewed client distributions. In this experiment, we set the number of clients to 2000, use user-level partitioning, randomly sample 5 clients per communication round, and adopt a two-layer CNN architecture as in [6]. The experimental results are shown below：
>
> | method             | Acc. (%, ↑) 250R | Acc. (%, ↑) 500R | Rounds (↓) 75% | Rounds (↓) 80% |
> |--------------------|-------|-------|------|-------|
> | FedSAGD(ours)               | 79.82 | 81.14 | 80   | 267   |
> | FedACG             | 79.5  | 80.62 | 82   | 285   |
> | FedANAG       | 79.37 | 80.54 | 86   | 288   |
> | FedAdam            | 76.79 | 79.01 | 180  | 500+  |
> | SCAFFOLD           | 78.58 | 79.86 | 175  | 500+  |
> | FedLNAG  | 74.46 | 78.31 | 346  | 500+  |
> | FedAvg             | 77.29 | 78.74 | 192  | 500+  |
> | MimeLite               | 78.78 | 80.18 | 179  | 372   |
> | FedAvgM            | 78.73 | 80.07 | 166  | 467   |
> | FedCM             | 75.58 | 79.05 | 226  | 500+  |
> | FedProx            | 77.79 | 79.33 | 187  | 500+  |
> | FedDyn             | 77.31 | 79.7  | 172  | 487   |
>
> Results demonstrates that our method also outperforms the baselines under feature-skew settings, highlighting its robustness to feature-skew heterogeneity.
>
> (2) As shown in Appendix A.6, we provide the results averaged across multiple runs. We still beat all the baselines (refer to Appendix A.6).
>
> [6] Caldas S, Duddu S M K, Wu P, et al. Leaf: A benchmark for federated settings. arXiv preprint arXiv:1812.01097, 2018.

---

### Official Review · Reviewer_BxwB · 2025-10-28

**Soundness:** 2
**Presentation:** 2
**Contribution:** 1
**Rating:** 2
**Confidence:** 4

**Summary:**

This paper stuided federated learning and proposed a new federated optimization algroithm. The authors cliamed that FL suffers from degraded generalization performance due to the inconsistency between local and global optimization objectives and client-side overfitting. They introduced global-update stability as an analytical tool to study generalization error and derive the stability bounds of mainstream FL optimization algorithms under non-convex settings.

**Strengths:**

The authors performed extensive analyses, covering both their method and existing baselines. The experimental section includes a wide range of simulation results demonstrating the effectiveness of the methods.

**Weaknesses:**

1. This paper addresses a well-studied problem, but I did not observe a clear breakthrough over existing works. In fact, some of the results appear weaker than those of prior studies. For instance, the theoretical analysis in this paper relies on a **bounded heterogeneity assumption**, whereas existing methods such as **FedDyn** and **SCAFFOLD** do not require this restriction. Moreover, recent advances in **momentum-based optimization** [R1, R2] have established convergence guarantees **without assuming bounded data heterogeneity**. This raises concerns about the originality and strength of the theoretical contribution.


2. The authors claim that their work addresses the stability issue under partial client participation. However, the presented analysis is not convincing. Specifically, the convergence proof is established under **uniform random client participation** and measures convergence using the **gradient norm**, while the stability analysis switches to the **parameter distance** metric without justification. For non-convex problems, existing algorithms have already demonstrated convergence under uniform random participation, so it is unclear why this particular metric is introduced or necessary.

Additionally, the result presented in **Line 1468** appears counterintuitive. It is unclear how one can meaningfully characterize the parameter property in the context of **non-convex optimization**. Furthermore, note that $\Phi$ could be unbounded (infinite) under the definition in **Line 1002**, making the derivations there mathematically questionable and potentially invalid.

[R1] Cheng, Ziheng, et al. "Momentum benefits non-iid federated learning simply and provably." arXiv preprint arXiv:2306.16504 (2023).

[R2] Cheng, Ziheng, and Margalit Glasgow. "Convergence of Distributed Adaptive Optimization with Local Updates." arXiv preprint arXiv:2409.13155 (2024).

**Questions:**

See the weakness

---

> ### Author Response · Authors · 2025-11-20
> **Author Response (part 1 of 2)**
>
> We sincerely appreciate the reviewer for recognizing our contributions and for the constructive comments. Our point-to-point responses to concerns on Weaknesses and Questions are given below.
>
> **Q1:The problem is well-studied, and there's no clear breakthrough. Some results are weaker. The theory needs a bounded heterogeneity assumption, but FedDyn, SCAFFOLD, and newer momentum methods [R1, R2] do not. This reduces originality and strength.**
>
> Our work focuses on the stability of FL algorithms rather than their convergence. Note that [R1] and [R2] cited in the review studies convergence, different from our work. Stability analysis aims to control the generalization error by examining the algorithm’s sensitivity to input data, whereas convergence analysis concerns the rate at which the optimization error decays during iterative updates. These two analyses differ fundamentally in both their goals and methodologies. While convergence in FL has already been well-studied in the literature, stability has only been studied in two prior works[1][3]. However, stability is crucial due to it can characterize how sensitive the global model is to changes in the data of local client or randomly selected client set, which helps analyze the generalization performance of FL algorithms. Our Theorem 3.5 also shows that a smaller upper bound on global stability implies a smaller generalization error, indicating that the model will perform better on unseen clients. In fact, many studies[6][7][8] have already used stability analysis to examine the relationship between an algorithm’s stability and its generalization error, and their empirical results similarly demonstrate that higher stability leads to better generalization.
>
> The concrete differences between our work and existing FL stability studies, as well as the associated technical challenges, are as follows:
>
> First, the stability analysis in [1] relies on the assumption that when a client’s data point is replaced, the new data comes from the **same distribution**. This assumption is unrealistic in practical cross-device FL, where the number of clients is large and per-round participation is sparse. When the replaced data no longer follows the same distribution—as is typical under heterogeneity—the treatment of gradient differences used in [1] breaks down, and a bounded heterogeneity term  $\sigma_g$ must be introduced to make the analysis feasible.
>
> Second, the sample-replacement stability framework in [1] fundamentally depends on the convexity of the loss function and cannot be directly extended to non-convex optimization. In particular, the construction of Lemma 2 in [1] does not hold in the non-convex setting. In our analysis, we address this issue by establishing Lemma 1, which enables the necessary inequality relaxation in the non-convex case.
>
> Third, the stability analysis in [3] only considers the local update process, without accounting for the influence of global updates. Therefore, similar to centralized stability analyses, it essentially examines the stability of the local update process. Their proposed algorithm also does not improve stability compared with FedAvg.
>
>
> We agree that developing algorithms capable of adapting to heterogeneity is a valuable direction—for example, approaches based on control variates such as SCAFFOLD. However, we focus on enhancing algorithmic stability, with the goal of achieving better generalization performance under a certain level of heterogeneity. Addressing robustness to arbitrary heterogeneous data is not the objective of our work.
> Moreover, existing studies have shown that scenarios with extreme heterogeneity are relatively uncommon in most practical FL applications [4][5]. Therefore, investigating the generalization performance of algorithms under a certain level of heterogeneity is both meaningful and necessary.
>
> Moreover, although [R1] and [R2] do not require the bounded heterogeneity assumption, this comes at the cost of restricting the momentum update: the momentum coefficient and gradient coefficient must sum to 1, which severely limits the expressive power of momentum. Many existing momentum-based FL algorithms [2][3] adopt new momentum update formulations, and their convergence analyses do rely on the bounded heterogeneity assumption. Nonetheless, these algorithms achieve better convergence and generalization performance than methods such as SCAFFOLD and FedDyn. In addition, as shown in Table 3, we have conducted experiments under varying levels of heterogeneity (Dirichlet distributions with parameters ranging from 0.3 to 0.8), and our method consistently achieves the best performance. Table 14 in Appendix A.6 reports the results under higher heterogeneity.

---

> ### Author Response · Authors · 2025-11-20
> **Author Response (part 2 of 2)**
>
> **Q2:They claim to handle stability with partial participation, but the analysis is unconvincing. Convergence uses gradient norm, while stability suddenly uses parameter distance without reason. For non-convex cases, existing methods already converge under random participation, so the new metric seems unnecessary.**
>
> It is stability, rather than convergence, that constitutes the primary focus of our study. Regarding the comment that “existing algorithms have already demonstrated convergence under uniform random participation,” we fully agree that convergence (measured via the gradient norm) has been analyzed for many FL algorithms. However, convergence alone does not characterize how sensitive the final model is to changes in the randomly selected client set, nor how well it generalizes to unparticipating clients in heterogeneous settings. Our analysis complements prior convergence results by studying global-update stability: we show that for existing momentum-based methods the model discrepancy between neighboring client sets grows as $O(T)$ while FedSAGD achieves a finite upper bound. under the same partial participation and non-convex assumptions. Combined with Theorem 3.5, this yields a strictly tighter generalization guarantee under partial participation and data heterogeneity.
>
> We would like to reiterate that the purpose of stability analysis is to control the generalization error bound by examining the algorithm's sensitivity to input data, while convergence analysis focuses on how an optimization algorithm approaches the optimal solution during the iterative process. These two types of analysis differ fundamentally in both objective and methodology.
> Moreover，We establish a connection between the stability upper bound and Definition 2.5 through the Lipschitz continuity of the loss function, as explained in line 208 of the paper.
>
> **Q3:The result in Line 1468 seems counterintuitive. In non-convex settings, the parameter property is unclear, and the quantity defined in Line 1002 can be unbounded, making the math questionable.**
>
> (1)The statement at Line 1468 is not intended as a property of a global optimum in a non-convex landscape, but as a property of the update dynamics of our algorithm under neighboring client sets. In particular, our hybrid proximal term modifies the global update by explicitly re-weighting the contribution of the previous global model. This has the effect of introducing an approximate weight-regularization (or contraction) on the global model: when we compare two runs with neighboring client sets, the proximal term dampens the discrepancy between their global updates, leading to the finite stability bound reported at Line 1468. Thus, even in a non-convex setting, the result simply characterizes how our specific update rule controls the parameter distance between two trajectories induced by neighboring client sets, rather than asserting any convex-like optimality property of the loss surface itself. We will clarify this interpretation in the revised version.(In the updated PDF, Line 1468 → Line 1629.)
>
> (2) Regarding $\Phi$ in Line 1002: including the gradient norm in a stability upper bound is a standard practice, which implicitly assumes that the gradient norm is bounded. This is consistent with the analyses in [1][3]. Bounds involving gradient norms are also common in non-FL stability analyses [9][10]. For clarity, one may further assume bounded stochastic gradients here. In addition, convergence analyses already indicate that the gradient norm does not grow unbounded, and our convergence analysis does not rely on a bounded-gradient assumption. (In the updated PDF, Line 1002 → Line 1164.)
>
> [1]Sun, Zhenyu, Xiaochun Niu, and Ermin Wei. Understanding generalization of federated learning via stability: Heterogeneity matters. AISTATS, 2024.
>
> [2]Kim, Geeho, et al. Communication-efficient federated learning with accelerated client gradient. CVPR, 2024.
>
> [3]Zhang, Hao, et al. Stabilizing and accelerating federated learning on heterogeneous data with partial client participation. TPAMI, 2024.
>
> [4]Li, Tian, et al. "Federated optimization in heterogeneous networks." Proceedings of Machine learning and systems (2020).
>
> [5]Ma, Xiaodong, et al. A state-of-the-art survey on solving non-iid data in federated learning. Future Generation Computer Systems, 2022.
>
> [6]Kuzborskij I, Lampert C. Data-dependent stability of stochastic gradient descent. ICML, 2018.
>
> [7]Hardt M, Recht B, Singer Y. Train faster, generalize better: Stability of stochastic gradient descent. ICML, 2016.
>
> [8]Chandramoorthy N, Loukas A, Gatmiry K, et al. On the generalization of learning algorithms that do not converge. NIPS, 2022.
>
> [9]Sun T, Li D, Wang B. Stability and generalization of decentralized stochastic gradient descent. AAAI 2021.
>
> [10]Zhu T, He F, Zhang L, et al. Topology-aware generalization of decentralized sgd. ICML, 2022.

---

> ### Comment · Reviewer_BxwB · 2025-11-26
>
> Thanks to the authors for the reply.
>
> **R1.** The authors claimed that the focus is the stability, but the section 4 is all about convergence. My criticise is that the results of convergence relying on the restrictive assumption, which is **weaker than** the existing result in the theoretical perspective. The authours rebuttal doesn't address my concern on this.
>
> The authours argued that ***Moreover, although [R1] and [R2] do not require the bounded heterogeneity assumption, this comes at the cost of restricting the momentum update: the momentum coefficient and gradient coefficient must sum to 1, which severely limits the expressive power of momentum.**
>
> I don't understand why the authors make such a argument. It is actually a very common way in optimization and it is quite flexible in my opinion. Specifically:
>
> $x= x - \eta g, ~~ g = \beta \nabla F(x) + (1-\beta) m $.
>
> You can adjust $\beta$ to adjust the weight of the momentum. Why does it severely limit the expressive power of momentum?

---

> ### Comment · Reviewer_BxwB · 2025-11-26
>
> Thanks to the authors for the reply.
>
> **R2 and R3**. It is still hard for me to understand how we can bound the parameter distance in non-convex optimization. The results obtained for the baselines are **very weak**, which don't truely reflect the potential of them. I think the reason is that the authors picked a wrong metric, parameter distance.
>
> I am also skeptism to their own results. As I wrote in the first round of the review, you can't leave the gradient in the final result. If you need bounded gradient assumption, please **explicitly** clarify that. And use the same assumption for the baselines.
>
> Second, I apart from the first issue, I also have some other doubts about your **proof of Theorem C.12**. I think Line 1623 doesn't hold as $\sum_{i\in S}x_{t,0}^i \neq \sum_{i\in S^{\prime}}x_{t,0}^i$. Moreover, the derivation from line 1622 to Line 1625 is missed. I feel the authors made a mistake in this derivation. **If I'm wrong, please correct me.** In addition, Lemma C.1 is very loose as well as the derivation from Line 1604 to line 1607.

---

> ### Author Response · Authors · 2025-11-27
> **Author Response (part 1 of 2)**
>
> We sincerely appreciate the reviewer’s prompt response.
>
> Q1: The authors claimed that the focus is the stability, but the section 4 is all about convergence. My criticise is that the results of convergence relying on the restrictive assumption, which is weaker than the existing result in the theoretical perspective. The authours rebuttal doesn't address my concern on this.
>
> There may be a misunderstanding here. Our main theoretical contribution is the stability and generalization analysis under partial client participation. The convergence analysis in Section 4 is not intended to provide a stronger convergence result than prior work. Instead, Section 4 primarily introduces the construction of our algorithm and clarifies how each algorithmic component affects stability; the convergence result at the end of Section 4 only serves to give a basic convergence guarantee for the algorithm.
>
> The “heterogeneity-free” convergence results referred to by the reviewer are not directly comparable to our setting. The existing method [R1] removes the bounded-heterogeneity assumption by adding momentum and proving convergence. However, it only works when the momentum has the Adam-style form $\beta v+(1-\beta)g$. Specifically, the heterogeneity-bounded assumption can be removed only under the following condition: the momentum use $\beta v +(1-\beta)g$, and the server must update momentum as $v=\frac{1}{\eta N K}\sum^K_{i=1}(x_t-x^i_{t,K})$. The detailed derivations can be found in Lemmas 8–10 of [R1]. Therefore, using momentum to remove the bounded-heterogeneity assumption puts strict constraints on the momentum update rule.
>
> Besides the Adam-style momentum $\beta v + (1-\beta)g$, mainstream momentum also include the classical Heavy Ball form $\beta v + g$ and Nesterov Accelerated Gradient (NAG).
> In centralized optimization, $\beta v + g$ and $\beta v + (1-\beta)g$ can be made equivalent by rescaling the learning rate. However, this equivalence does not hold in FL, where momentum affects not only the global update on the server but also the local updates and proximal terms on the clients. As a result, a simple rescaling is no longer applicable. Therefore, momentum other than Adam-style momentum have not been shown to remove the bounded-heterogeneity assumption in FL.
>
> In fact, the recent momentum-based FL SOTA methods [1] and [2] also do not use the Adam-style momentum update and both rely on the bounded-heterogeneity assumption. We also experimented with both the Adam-style momentum and the Heavy Ball momentum update. Our results show that the Heavy Ball momentum yields better performance in our setting. Therefore, we adopt the Heavy Ball momentum update in our method.
>
> Overall, we do not weaken the conclusions of existing studies. Prior work only shows that, under Adam-style momentum $\beta v + (1-\beta)g$, the bounded heterogeneity assumption in FL can be removed. These studies do not establish convergence results for other types of momentum updates（such as Heavy Ball or Nesterov Accelerated Gradient (NAG)） under unbounded heterogeneity.
>
> We acknowledge that exploring how other momentum forms(such as Heavy Ball and Nesterov’s Accelerated Gradient (NAG)) can remove the bounded-heterogeneity assumption in FL is an important and meaningful direction for future research.
> However, this is not the focus of our work. Section 4 mainly presents the structure of our algorithm and analyzes how each component affects stability, followed by a discussion of its convergence. The convergence proof serves only to provide the necessary theoretical guarantee for our method.
>
> [1] Kim, Geeho, et al. Communication-efficient federated learning with accelerated client gradient. CVPR, 2024.
>
> [2] Zhang, Hao, et al. Stabilizing and accelerating federated learning on heterogeneous data with partial client participation. TPAMI, 2024.
>
> [3] Sun, Zhenyu, Xiaochun Niu, and Ermin Wei. Understanding generalization of federated learning via stability: Heterogeneity matters. AISTATS, 2024.
>
> [4] Train faster, generalize better: Stability of stochastic gradient descent. ICML, 2016.
>
> [5] Fine-grained analysis of stability and generalization for stochastic gradientdescent.ICML, 2020.

---

> ### Author Response · Authors · 2025-11-27
> **Author Response (part 2 of 2)**
>
> 2.R2 and R3. It is still hard for me to understand how we can bound the parameter distance in non-convex optimization. The results obtained for the baselines are very weak, which don't truely reflect the potential of them. I think the reason is that the authors picked a wrong metric, parameter distance.
>
> We would like to clarify that using parameter distance as a stability measure is not introduced by us. In prior stability studies—both in non-federated learning (non-FL) [4][5] and federated learning (FL) [2][3]—parameter distance has been widely used as the primary metric.
>
> In non-FL settings, for example:
> [4] (ICML 2016, 1698 citations), Eq. (3.1)；[5] (ICML 2020), Theorem 3.
>
> In FL settings, for example:
>
> [3] (AISTATS 2024), Appendix C.1, Lemma 4;[2] (TPAMI 2024), Eq.(4).
>
> Among them, [4] and [5] explicitly use Lipschitz continuity to relate the difference in loss values to the parameter distance, and then adopt the parameter distance as the stability metric.
> Although [3] defines stability in terms of the loss difference in the main text, its proofs rely on parameter distance.
> In [2], the definition directly takes the parameter distance as the stability measure.
>
> In summary, parameter distance has been the standard metric for stability analysis in prior work, and it is not an assumption or design introduced by our paper.
>
> 3.I am also skeptism to their own results. As I wrote in the first round of the review, you can't leave the gradient in the final result. If you need bounded gradient assumption, please explicitly clarify that. And use the same assumption for the baselines.
>
> In this part, we rely on the Lipschitz continuity assumption, which is also widely used in prior stability analyses [4][5].
> We have explicitly included the Z-Lipschitz continuity assumption in the Assumption 3.4 of the paper: $\Vert F_i(x)-F_i(y) \Vert\leq Z\Vert  x-y \Vert$. From this, we can naturally obtain that $\Vert f(x)-f(y) \Vert\leq Z\Vert  x-y \Vert$ and $\Vert \nabla f \Vert\leq Z$. We have updated the PDF to incorporate this assumption. Moreover, we apply the same assumption to all baseline methods in our computations, and their results also involve $Z$ under this assumption.
>
> 4.Some concerns regarding the proofs.
>
> (1)We appreciate the reviewer’s observation regarding the issue in line 1623. This is a typographical error. The correct expression should be $\mathbb{E}\Vert x_{t+1}-x_{t+1}' \Vert = \frac{1}{s} \mathbb{E}\Vert \sum_{i\in \mathcal{S}}x_{t,K}^i-\sum_{j\in\mathcal{S}'}x_{t,K}^j \Vert$. However, the result presented in line 1625 remains correct. The derivation is as follows: \begin{align*}
>     \mathbb{E}\Vert x_{t+1}-x_{t+1}'\Vert = \frac{1}{s}\mathbb{E}\Vert \sum_{i\in \mathcal{S}}x_{t,K}^i-\sum_{j\in\mathcal{S}'}x_{t,K}^j \Vert \leq(1- \Gamma)\mathbb{E}\Vert x_t-x_t'\Vert+\frac{2+2\beta}{\lambda+\mu}(\sigma^2_l+2\sigma^2_g+2Z^2)^{\frac{1}{2}}.
> \end{align*}
> Here, the result follows directly from the relation already established in line 1612. Specifically, for any pair of local models $x^i_{t,K}$ and $x^j_{t,K}$, line 1612 has shown that \begin{align*}
>     \mathbb{E}\Vert x^i_{t,K}-x^j_{t,K}\Vert \leq (1-\Gamma)\mathbb{E}\Vert x_t-x'_t\Vert+\frac{1-(1-\eta_l(\lambda+\mu))^K}{\lambda+\mu}(2+2\beta)(\sigma^2_l+2\sigma^2_g+2Z^2)^{\frac{1}{2}},
> \end{align*} and $1-(1-\eta_l(\lambda+\mu))^K<1$.
> We have updated the PDF.
>
> (2)After explicitly adding Assumption 3.4, we also updated Lemma C.1 accordingly. With the added Lipschitz assumption, the gradient norm no longer appears in the result. In fact, this type of scaling bound in Lemma C.1 is standard in stability analysis. For example, see Lemma 4 and Lemma 5 in [2], and Lemma 1 in [3].
>
> If there are any further questions, we would be very happy to continue the discussion with the reviewers. Thank you.

---

> ### Comment · Reviewer_BxwB · 2025-11-28
>
> I really think the derivation from Line 1620 to 1630 is incorrect. You can derive it step by step. Don't omit the derivation from Line 1620 to 1622. The first step is important.

---

> > ### Author Response · Authors · 2025-11-28
> > **Author Response (part 1 of 2)**
> >
> > Thank you for the reviewer’s prompt response. We will provide a detailed clarification below.
> >
> > First, $\frac{1}{s}\sum _ {j\in \mathcal{S}}x^j _ {t,K}≠\frac{1}{s}\sum _ {j\in\mathcal{S}'}x^j _ {t,K}$. In our original proof, all terms $x^j _ {t,K}$ correspond to indices $j\in\mathcal{S}'$, which may have caused some confusion.
> > To make the derivation clearer, we have rewritten the proof by denoting $x^j _ {t,K}$ with $j\in\mathcal{S}'$ as $x'^j _ {t,K}$.
> >
> > The detailed derivation is as follows:
> >
> > $x _ {t+1}=x _ t+\frac{1}{s}\sum _ {i\in\mathcal{S}}(x^i _ {t,K}-x^i _ {t,0})$,
> >
> > $x' _ {t+1}=x' _ t+\frac{1}{s}\sum _ {j\in\mathcal{S}'}(x'^j _ {t,K}-x'^j _ {t,0})$.
> >
> > The global model broadcast by the server in round t is identical to the initial model used by the clients in round $t$.
> >
> > $x^i _ {t,0} = x _ t$ , $x'^i _ {t,0} = x' _ t$.
> >
> > Therefore, $x _ {t+1} = \frac{1}{s}\sum _ {i\in\mathcal{S}}x^i _ {t,K}$, $x' _ {t+1} = \frac{1}{s}\sum _ {i\in\mathcal{S}'}x'^i _ {t,K}$.
> >
> > \begin{align*}
> >     \mathbb{E}\Vert x _ {t+1}-x' _ {t+1}\Vert &= \frac{1}{s}\mathbb{E}\Vert \sum _ {i\in \mathcal{S}}x^i _ {t,K}-\sum _ {j\in\mathcal{S}'}x'^j _ {t,K}  \Vert\\\\
> >     &\leq(1- \Gamma)\mathbb{E}\Vert x _ t-x' _ t\Vert+\frac{2+2\beta}{\lambda+\mu}(\sigma^2 _ l+2\sigma^2 _ g+2Z^2)^{\frac{1}{2}}.
> > \end{align*}
> >
> > Since $|\mathcal{S}|=|\mathcal{S}'| = s$, the term $\sum _ {i\in \mathcal{S}}x^i _ {t,K}-\sum _ {j\in\mathcal{S}'}x'^j _ {t,K}$ is equivalent to the sum of $s$ paired terms of the form $x^i _ {t,K} - x'^j _ {t,K}$. Let $\mathcal{S}=\{i _ 1,\dots,i _ s\},\mathcal{S}'=\{j _ 1,\dots,j _ s\}$.
> > Then, we can rewrite the difference as a sum of $s$ paired terms, $\frac{1}{s}\mathbb{E}\Vert \sum _ {i\in \mathcal{S}}x^i _ {t,K}-\sum _ {j\in\mathcal{S}'}x'^j _ {t,K}  \Vert=\frac{1}{s}\mathbb{E}\Vert\sum _ {l=1}^{s}(x^{i _ l} _ {t,K}-x'^{j _ l} _ {t,K})\Vert \leq\frac{1}{s}\sum^s _ {l=1}\mathbb{E}\Vert x^{i _ l} _ {t,K}-x'^{j _ l} _ {t,K} \Vert.$
> >
> > Therefore, we need to bound $\mathbb{E}\Vert x^i _ {t,K} - x'^j _ {t,K}\Vert$. This term $\mathbb{E}\Vert x^i _ {t,K}-x'^j _ {t,K}\Vert$ can be obtained recursively from the bound on $\mathbb{E}\Vert x^i _ {t,k}-x'^j _ {t,k} \Vert$.
> >
> > From line 1600，we have:
> >
> > \begin{align*}
> >     &\mathbb{E}\Vert x^i _ {t,k}-x'^j _ {t,k} \Vert  & \\\\
> >     &=\mathbb{E}\Vert x^i _ {t,k-1}-\eta _ l\nabla F _ i(x^i _ {t,k-1},\xi _ i)-\eta _ l(\lambda+\mu) x^i _ {t,k-1} -\beta\eta _ lv _ t-(x'^j _ {t,k}-\eta _ l\nabla F _ j(x'^j _ {t,k-1},\xi _ j)\\\\
> >     &\ \ \ \ -\eta _ l(\lambda+\mu)x'^j _ {t,k-1}-\beta\eta _ lv' _ t)+\eta _ l\lambda(x _ t-x' _ t)\Vert &\\\\
> >     & \leq \mathbb{E}\Vert (1-\eta _ l(\lambda+\mu))(x^i _ {t,k-1} -x'^j _ {t,k-1})\Vert+ \eta _ l \mathbb{E}\Vert \nabla F _ i(x^i _ {t,k-1},\xi _ i)-\nabla F _ j(x'^j _ {t,k-1},\xi _ j) \Vert & \\\\
> >     & \ \ \ \ +\beta\eta _ l\mathbb{E}\Vert v _ t-v' _ t \Vert +\eta _ l\lambda\mathbb{E}\Vert x _ t-x' _ t \Vert&\\\\
> >     & \leq \mathbb{E}\Vert (1-\eta _ l(\lambda+\mu))(x^i _ {t,k-1} -x'^j _ {t,k-1})\Vert+ \eta _ l \mathbb{E}\Vert \nabla F _ i(x^i _ {t,k-1},\xi _ i)\Vert+\mathbb{E}\Vert\nabla F _ j(x'^j _ {t,k-1},\xi _ j) \Vert & \\\\
> >     & \ \ \ \ +\beta\eta _ l\mathbb{E}\Vert v _ t\Vert+\beta\eta _ l\mathbb{E}\Vert v' _ t \Vert +\eta _ l\lambda\mathbb{E}\Vert x _ t-x' _ t \Vert&\\\\
> >     & \leq  (1-\eta _ l(\lambda+\mu))\mathbb{E}\Vert x^i _ {t,k-1}-x'^j _ {t,k-1} \Vert + \eta _ l\lambda\mathbb{E}\Vert x _ t-x' _ t\Vert+(2+2\beta)\eta _ l(\sigma^2 _ l+2\sigma^2 _ g+2Z^2)^{\frac{1}{2}},
> > \end{align*}
> >
> > Here, we use the triangle inequality and Lemma C.1. Moreover, the momentum term is a weighted combination of past gradients and is bounded above by the gradient norm.
> >
> > When iterated to $k=K$, we obtain (from line 1612)：
> >
> > $$
> > \begin{align*}
> >     \mathbb{E}\Vert x^i _ {t,K}-x'^j _ {t,K}\Vert &\leq (1-\eta _ l(\lambda+\mu))^K\mathbb{E}\Vert x _ t-x' _ t\Vert+\frac{\lambda-\lambda(1-\eta _ l(\lambda+\mu))^K}{\lambda+\mu}\mathbb{E}\Vert x _ t-x' _ t \Vert\\\\
> >     &\ \ \ \ +\frac{1-(1-\eta _ l(\lambda+\mu))^K}{\lambda+\mu}(2+2\beta)(\sigma^2 _ l+2\sigma^2 _ g+2Z^2)^{\frac{1}{2}}\\\\
> >     &=\left[\frac{\lambda}{\lambda+\mu}+\frac{\mu}{\lambda+\mu}(1-\eta _ l(\lambda+\mu))^K\right]\mathbb{E}\Vert x _ t-x' _ t\Vert\\\\
> >     &\ \ \ \ +\frac{1-(1-\eta _ l(\lambda+\mu))^K}{\lambda+\mu}(2+2\beta)(\sigma^2 _ l+2\sigma^2 _ g+2Z^2)^{\frac{1}{2}}.
> > \end{align*}
> > $$
> >
> > Let $\Gamma=1-\left[\frac{\lambda}{\lambda+\mu}+\frac{\mu}{\lambda+\mu}(1-\eta _ l(\lambda+\mu))^K\right]\in(0,1)$. Since $1-(1-\eta _ l(\lambda+\mu))^K<1$, we obtain:
> > \begin{align*}
> > \mathbb{E}\Vert x^i _ {t,K}-x'^j _ {t,K}\Vert
> > &\leq(1- \Gamma)\mathbb{E}\Vert x _ t-x' _ t\Vert+\frac{2+2\beta}{\lambda+\mu}(\sigma^2 _ l+2\sigma^2 _ g+2Z^2)^{\frac{1}{2}}
> > \end{align*}

---

> > ### Author Response · Authors · 2025-11-28
> > **Author Response (part 2 of 2)**
> >
> > In summary, we obtain the following result:
> >
> > \begin{align*}
> >     \mathbb{E}\Vert x _ {t+1}-x' _ {t+1}\Vert &= \frac{1}{s}\mathbb{E}\Vert \sum _ {i\in \mathcal{S}}x^i _ {t,K}-\sum _ {j\in\mathcal{S}'}x'^j _ {t,K}  \Vert\\\\
> >     &=\frac{1}{s}\mathbb{E}\Vert\sum _ {l=1}^{s}(x^{i _ l} _ {t,K}-x'^{j _ l} _ {t,K})\Vert\\\\
> >     &\leq\frac{1}{s}\sum^s _ {l=1}\mathbb{E}\Vert x^{i _ l} _ {t,K}-x'^{j _ l} _ {t,K} \Vert\\\\
> >     &\leq(1- \Gamma)\mathbb{E}\Vert x _ t-x' _ t\Vert+\frac{2+2\beta}{\lambda+\mu}(\sigma^2 _ l+2\sigma^2 _ g+2Z^2)^{\frac{1}{2}},
> > \end{align*}
> >
> > $\mathcal{S}=${$i _ 1,\dots,i _ s $},$\mathcal{S}'=${$j _ 1,\dots,j _ s$}.
> >
> > When $t=0$, we have $x _ 0 = x' _ 0$. Therefore, we can iterate to obtain:
> > \begin{align*}
> >     \mathbb{E}\Vert x _ T-x' _ T\Vert &\leq \frac{1-(1- \Gamma)^T}{\Gamma}\frac{2+2\beta}{\lambda+\mu}(\sigma^2 _ l+2\sigma^2 _ g+2Z^2)^{\frac{1}{2}}.
> > \end{align*}
> >
> > We hope these responses address your concerns.
> >
> > If there are any further questions, we would be very happy to continue the discussion with the reviewers. Thank you.

---

### Official Review · Reviewer_yxXG · 2025-10-30

**Soundness:** 3
**Presentation:** 3
**Contribution:** 3
**Rating:** 8
**Confidence:** 4

**Summary:**

The paper proposes FedSAGD, a FL algorithm that integrates global momentum and a hybrid proximal term to enhance both convergence and stability. By formalizing a new concept of global-update stability, the authors connect generalization guarantees to the sensitivity of global updates under partial participation. They provide rigorous theoretical proofs showing improved stability convergence, and validate the method through experiments on common FL benchmarks.

**Strengths:**

1. The paper introduces the novel concept of global-update stability and leverages it to design the FedSAGD algorithm, providing deep theoretical insights and improved generalization guarantees in federated learning, especially under client heterogeneity.

2. This provides a principled theoretical foundation linking generalization ability to stability, distinguishing it from previous works that focused primarily on convergence or variance reduction.

**Weaknesses:**

1. Sensitivity and Ablation Analysis Could Be Deeper: Although the paper examines parameter sensitivity for β, λ, and μ, it lacks detailed ablations isolating the effects of each component (momentum vs. proximal term). It’s unclear how much each part independently contributes to stability or convergence gains.

**Questions:**

1. Methods like SCAFFOLD also improves global stability with the control variates, and they do not require tuning on proximal terms. Would you think such design can also be incorporated to FedSAGD for improved results?

**Details Of Ethics Concerns:**

No concern.

---

> ### Author Response · Authors · 2025-11-20
> **Author Response**
>
> We sincerely appreciate the reviewer for recognizing our contributions and for the constructive comments. Our point-to-point responses to concerns on Weaknesses and Questions are given below.
>
> **Q1:There are no clear ablations to show what momentum or the proximal term each contributes.**
>
> We further add ablation studies to separate the contributions of the momentum and the proximal term. The results are as follows:
> | momentum | proximal term | Accuracy |
> |:-:|:-:|-:|
> |$\checkmark$| $\times$ | 82.84|
> | $\times$  |$\checkmark$ | 80.54|
> |$\checkmark$|$\checkmark$ | 84.05|
>
>
> **Q2:SCAFFOLD improves stability with control variates and avoids proximal tuning. Could a similar design be added to FedSAGD to further improve it?**
>
> In Appendix C.3, we show that control variates do not improve the global update stability order of FL.
>
>
> We incorporate the control variate design into FedSAGD, denoted as FedSAGD-CV, and conduct a stability analysis of this variant. The results are as follows:
>
> local update rule：$x^i_{t,k+1}=x^i_{t,k}-\eta_l(\nabla F_i(x^i_{t,k},\xi_i)+\
> v_t+\lambda(x^i_{t,k}-x^i_{t,0})-\mu x^i_{t,k}-c_i+c)$
>
>
> local model difference：
> $\begin{align*}\mathbb{E}\Vert x^i_{t,k} - x^j_{t,k} \Vert &\leq(1-\eta_l(\lambda+\mu))\mathbb{E}\Vert x^i_{t,k-1}-x^j_{t,k-1}\Vert+\eta_l\lambda\mathbb{E}\Vert x_t-x'_t \Vert
> \\
> &+(6+2\beta)\eta_l(\sigma^2_l+2\sigma^2_g+2\Phi)^{\frac{1}{2}}
> \end{align*}$
>
> global model difference:
> $\mathbb{E}\Vert x_{t+1} - x'_{t+1} \Vert\leq(1-\Gamma)\mathbb{E}\Vert x_t-x'_t\Vert +\frac{6+2\beta}{\lambda+\mu}(\sigma^2_l+2\sigma^2_g+2\Phi)^{\frac{1}{2}}$
>
> $\mathbb{E}\Vert x_T-x'_T\Vert \leq \frac{1-(1- \Gamma)^T}{\Gamma}\frac{6+2\beta}{\lambda+\mu}(\sigma^2_l+2\sigma^2_g+2\Phi)^{\frac{1}{2}}$
>
> After introducing control variates, FedSAGD-CV achieves the same stability order as FedSAGD, but with a larger constant. Our analysis shows that this is because the global control variate changes whenever the participating clients change, which enlarges the discrepancy between global models and harms global stability. In addition, control variates introduce more complex computations and incur higher communication overhead.

---

### Official Review · Reviewer_EQAa · 2025-10-31

**Soundness:** 2
**Presentation:** 3
**Contribution:** 2
**Rating:** 4
**Confidence:** 4

**Summary:**

The paper starts from the observation that existing momentum-based acceleration methods in FL don’t necessarily improve stability. Building on this, it proposes FedSAGD, a new optimization framework designed to make FL training more stable under partial participation and heterogeneous data. The paper proposes a novel notion called global update stability,  which measures how sensitive the global model is to variations in the participating client set. Theoretical results show the link between this stability and generalization error, and suggest that FedSAGD achieves optimal convergence rates for non-convex objectives. Empirical evaluations on several standard FL benchmarks (CIFAR-10/100, EMNIST-L, Shakespeare) show consistent improvements over popular baselines such as FedAvg, FedProx, and SCAFFOLD.

**Strengths:**

1. The introduction of global update stability as a metric to quantify the effect of client sampling randomness is original and potentially impactful. It provides a new way to reason about generalization in FL beyond the traditional sample-level stability.
2. The paper is well-written, with clear definitions, lemmas. Most proofs are easy to follow.

**Weaknesses:**

1. Theoretical assumptions are overly idealized.
2. Communication and computation costs are not reported.

3. Security and privacy concerns are ignored.
4. Experimental evaluation lacks ablation and statistical rigor.

**Questions:**

1. The convergence and stability analyses rely on a set of strong assumptions—smoothness, bounded gradients, limited client heterogeneity, synchronous updates, and i.i.d. client sampling. These are rarely satisfied in real-world FL systems, especially in cross-device settings. As a result, the theoretical results cover only a small fraction of realistic scenarios, and it’s not obvious how much they translate to practical improvements.

2. The paper does not report communication or computation costs, even though FedSAGD adds extra components like the hybrid proximal term and global momentum updates. Without such analysis, it’s hard to judge the real efficiency gains. Is there any comparison of communication and computation overhead with baselines such as FedProx or SCAFFOLD? It would be useful to know how much extra cost the hybrid proximal term adds per round.

3. There’s no empirical evidence showing that smaller global stability actually leads to better generalization. It would be helpful to see a quantitative analysis or correlation study to support this claim.

4. The method requires the server to keep an exponential moving average of historical gradients.

   Without secure aggregation, this can expose gradient information and potentially lead to data reconstruction risks, an issue that runs counter to FL’s privacy goals.

5. How sensitive is FedSAGD to the choice of the global momentum coefficient $\beta$ and proximal weights $(\mu, \lambda)$ under strongly non-IID settings?

   Would performance drop significantly if the heterogeneity parameter \alpha (in Dirichlet partitioning) were smaller?

6. How does the proposed global update stability correlate with the empirical generalization gap in experiments? Was this relationship measured or analyzed quantitatively?

---

> ### Author Response · Authors · 2025-11-20
> **Author Response (part 1 of 3)**
>
> We sincerely appreciate the reviewer for recognizing our contributions and for the constructive comments. Our point-to-point responses to concerns on Weaknesses and Questions are given below.
>
> PS:The line numbers mentioned in our response refer to the line numbers in the revised PDF.
>
> **W1&Q1: Theoretical assumptions are overly idealized.**
>
> In our convergence and stability analyses, we adopt the standard assumptions of L-smooth, bounded heterogeneity, and unbiased stochastic gradients. Importantly, we do not require the client data to be i.i.d., nor do we assume bounded gradients. In addition, synchronous and asynchronous updates represent two fundamentally different settings. Our work focuses exclusively on the synchronous setting, while asynchronous updates can be investigated in future work.
> To the best of our knowledge, these assumptions are standard in theoretical studies. For example, existing convergence analyses of momentum-accelerated FL algorithms—such as FedAcg (CVPR 2024) [1], FedANAG (TPAMI 2025) [2], and FedAdam (ICLR 2021) [3]—all rely on assumptions of L-smooth, bounded heterogeneity, and unbiased gradient estimators. Similarly, recent studies on stability also adopt these assumptions [4],[5]. Some works [6] avoid the bounded heterogeneity assumption, but only under a very specific momentum form: specifically, the momentum coefficient and the gradient coefficient must sum to 1, i.e., the Adam-style form $\beta v+(1-\beta)g$. Therefore, using momentum to remove the bounded-heterogeneity assumption puts strict constraints on the momentum update rule. Besides the Adam-style momentum, mainstream momentum also include the classical Heavy Ball form $\beta v + g$ and Nesterov Accelerated Gradient.  In fact, the recent momentum-based FL SOTA methods[1][2] also do not use the Adam-style momentum and both rely on the bounded-heterogeneity assumption. We also experimented with different momentum forms, and our results show that the Heavy Ball momentum better performance in our setting.
>
> [1]Kim, Geeho, et al. "Communication-efficient federated learning with accelerated client gradient." Proceedings of the IEEE/CVF Conference on Computer Vision and Pattern Recognition (2024).
>
> [2]Zhang, Hao, et al. "Stabilizing and accelerating federated learning on heterogeneous data with partial client participation." IEEE Transactions on Pattern Analysis and Machine Intelligence (2024).
>
> [3]Reddi, Sashank J., et al. "Adaptive Federated Optimization." International Conference on Learning Representations (2021).
>
> [4]Sun, Zhenyu, Xiaochun Niu, and Ermin Wei. "Understanding generalization of federated learning via stability: Heterogeneity matters." International conference on artificial intelligence and statistics (2024).
>
> [5]Liu, Yingqi, et al. "Understanding the Stability-based Generalization of Personalized Federated Learning." International Conference on Learning Representations (2025).
>
> [6]Cheng, Ziheng, et al. "Momentum Benefits Non-iid Federated Learning Simply and Provably." International Conference on Learning Representations (2024).
>
> **W2&Q2: Communication costs are not reported.**
>
> Communication Cost. Lines 336–342 of the paper analyze the communication cost of the algorithm under wireless communication, and the results are reported in the Communication column of Table 1. Here, we provide a more detailed explanation.
> In FL, the server broadcasts the global model $x^t$ once to all clients in the downlink, incurring a communication cost equivalent to one model size. In the uplink, $s$ clients unicast their local updates to the server, resulting in a total cost of $s$ model sizes.
> Therefore, in FedProx only model parameters are uploaded and downloaded. In contrast, SCAFFOLD require uploading and downloading information of size $2s$ times the model parameters (e.g., momentum vectors or control variates).
> For FedSAGD (ours), the server downloads both the model parameters and the momentum, while each client uploads only the model parameters (since the momentum and control variates have the same dimensionality as the model parameters).
> Under a network with broadcast support (e.g., wireless communication), the communication cost of FedProx is (s + 1) times the model size; the communication cost of SCAFFOLD is (2s+2) times the model size; the communication cost of FedSAGD (ours) is (s + 2) times the model size.
>
> Therefore, SCAFFOLD has the highest communication cost of $(2s+2)$.
> For FedProx, the communication cost is $(s+1)$, while ours is $(s+2)$. As the number of clients increases, the gap between the two becomes small. Specifically, under our cross-device FL setup with 50 clients, our method requires only about 2\% more communication overhead than FedProx, while achieving an average improvement of about 19\% on CIFAR-10 and CIFAR-100.
> In contrast, compared with SCAFFOLD, our method reduces the communication cost by about 48\%. The results for more algorithms are summarized in the “Communication” column of Table 1.

---

> ### Author Response · Authors · 2025-11-20
> **Author Response (part 2 of 3)**
>
> **W2&Q2: Computation costs are not reported.**
>
> Computation cost. The additional computation introduced by our method compared with FedAvg, due to momentum primarily occurs in three places: (1) momentum updates on the server side, (2) gradient correction on the client side, and (3) the computation of the proximal term. However, both operations have a computational complexity of $\mathcal{O}(d)$, where
>  is the dimensionality of the model parameters. We also recorded the average computation time per round for each algorithm in the CIFAR-100 experiments with ResNet-18 (including the total computation time on both the server and clients). The results are as follows:
> |Method|FedAvg|FedProx|SCAFFOLD|FedDyn|FastSlowMo|FedLNAG|FedAdam|FedCM|MimeLite|FedACG|FedANAG|FedSAGD(ours)|
> |-|-|-|-|-|-|-|-|-|-|-|-|-|
> |Time(s)|0.358|0.401|0.424|0.467|1.095|0.521|0.366|0.415|0.485|0.404|0.416|0.411|
>
> **Q3:There’s no empirical evidence showing that smaller global stability actually leads to better generalization. It would be helpful to see a quantitative analysis or correlation study to support this claim.**
>
> Thank you for the reviewer’s suggestion. The purpose of stability analysis is to control the generalization error by examining the algorithm’s sensitivity to input data. Theorem 3.5 provides a quantitative result derived from stability: a smaller upper bound on global stability implies a smaller generalization error, indicating better performance on the data of unseen clients.
>
> In fact, numerous works in both non-FL settings [6][7] and FL settings [4][5] have used stability analysis to study the relationship between algorithmic behavior and generalization error, and their empirical results[6][8][9] similarly demonstrate that higher stability leads to better generalization. We have clarified this in the Introduction and Related Work sections.
>
> [6]Hardt M, Recht B, Singer Y. Train faster, generalize better: Stability of stochastic gradient descent. ICML, 2016.
>
> [7]Zhu, Tongtian, et al. Topology-aware generalization of decentralized sgd. ICML, 2022.
>
> [8]Kuzborskij I, Lampert C. Data-dependent stability of stochastic gradient descent. ICML, 2018.
>
> [9]Chandramoorthy N, Loukas A, Gatmiry K, et al. On the generalization of learning algorithms that do not converge. NIPS, 2022.
>
> **W3&Q4: Security and privacy concerns are ignored.**
>
> First, our method uploads only gradients. Second, the exponential moving average of historical gradients is computed based on the aggregated global gradient. We can still employ secure aggregation to obtain the aggregated global gradient securely.
> Therefore, if secure aggregation is not used, our method exposes only client-side gradients to the server, the same information exposed by FedAvg and FedProx. If secure aggregation is applied, our method exposes only the aggregated global gradient to the server, again matching what is exposed by FedAvg and FedProx. Thus, no additional privacy risks are introduced. Moreover, maintaining an exponential moving average of historical gradients is a standard practice in global-momentum federated learning methods, such as FedAdam [3], FedANAG [2], and FedACG [1].

---

> ### Author Response · Authors · 2025-11-20
> **Author Response (part 3 of 3)**
>
> **Q5: How sensitive is FedSAGD to the choices of global momentum β and proximal weights (μ, λ) in strongly non-IID cases?**
>
> Our sensitivity experiments are already presented in Appendix A.5.
> To further demonstrate parameter sensitivity under strongly non-IID settings, we additionally include the results for $\alpha = 0.1$ below (all other settings follow Appendix A.5). The performance of FedSAGD remains stable across a wide range of values for the parameters $\beta$, $\mu$, and $\lambda$. Specifically, for $\beta$ values ranging from 0.6 to 0.95, FedSAGD maintains consistently high accuracy with slight fluctuations, peaking at $\beta = 0.9$. For $\mu$ values ranging from 0.01 to 0.0001, the accuracy remains high and stable, with the highest performance observed at $\mu = 0.001$. For $\lambda$ values ranging from 0.1 to 0.0001, FedSAGD shows stable performance with consistent accuracy, reaching its peak at $\lambda = 0.01$.
>
> $\lambda  = 0.01, \mu =0.001:$
> | $\beta$ | 0.99 | 0.95 | 0.9 | 0.8 | 0.7 | 0.6 | 0.1 |
> |-|-|-|-|-|-|-|-|
> | Acc.(%) | 77.31 | 78.44 | 78.65 | 78.24 | 77.86 | 77.29 | 74.78 |
>
> $\lambda  = 0.01, \beta =0.9:$
> | $\mu$ | 0.1 | 0.01 | 0.005 | 0.001 | 0.0005 | 0.0001 | 0.00001 |
> |-|-|-|-|-|-|-|-|
> | Acc.(%) | 77.22 | 80.09 | 80.93 | 81.34 | 81.14 | 80.08 | 78.72 |
>
> $\mu =0.001, \beta =0.9:$
> | $\lambda$ | 1 | 0.1 | 0.05 | 0.01 | 0.005 | 0.001 | 0.0001 |
> |-|-|-|-|-|-|-|-|
> | Acc.(%) | 70.97 | 76.62 | 77.15 | 78.65 | 78.54 | 78.31 | 78.12 |
>
> **Q5: Would performance drop significantly if the heterogeneity parameter \alpha (in Dirichlet partitioning) were smaller?**
>
> The performance variation with respect to the heterogeneity parameter $\alpha$ is already reported in Table 3.
> To further illustrate the behavior under smaller $\alpha$, we additionally include the results for $\alpha = 0.1$ and $\alpha = 0.2$ below (with all other settings identical to those in Table 3).
>
> $\alpha = 0.1:$
> |Method|FedSAGD(ours)|FedACG|FedANAG|SCAFFOLD|FedDyn|FedLNAG|FedAvgM|FedAdam|FedCM|
> |-|-|-|-|-|-|-|-|-|-|
> |Accuracy(%)|81.01|76.64|80.56|79.18|76.65|76.52|78.7|74.3|73.39|
>
> $\alpha = 0.2:$
> |Method|FedSAGD(ours)|FedACG|FedANAG|SCAFFOLD|FedDyn|FedLNAG|FedAvgM|FedAdam|FedCM|
> |-|-|-|-|-|-|-|-|-|-|
> |Accuracy(%)|82.37|80.97|81.86|80.35|77.51|78.05|80.09|78.31|78.82|
>
> W4:Experimental evaluation lacks ablation and statistical rigor.
>
> Statistical rigor:
> As shown in Appendix A.6, we provide the results averaged across multiple runs.
>
> Ablation:
> We further add ablation studies to separate the contributions of the momentum and the proximal term. The results are as follows:
> | momentum | proximal term | Accuracy |
> |:-:|:-:|-:|
> |$\checkmark$| $\times$ | 82.84|
> | $\times$  |$\checkmark$ | 80.54|
> |$\checkmark$|$\checkmark$ | 84.05|
>
> **Q6:How does the proposed global update stability correlate with the empirical generalization gap in experiments? Was this relationship measured or analyzed quantitatively?**
>
> We quantitatively analyze the relationship between the proposed global update stability and the generalization error in Theorem 3.5. In general, a smaller generalization error bound implies better generalization performance on the data of unseen clients.
> We also quantitatively measure the stability of each algorithm in our experiments(see Appendix A.4). We further include experiments measuring the empirical generalization gap.
> Using the CIFAR-10 task as the benchmark, we report the absolute accuracy gap and the absolute loss gap between training and testing at $epoch=300$. We will add the curves of the empirical generalization gap in the appendix of the revised version of the paper.
> |Metric|FedLNAG|FedDyn|FedAvg|FedANAG|FedProx|FedAdam|FedACG|FedSAGD|
> |-|-|-|-|-|-|-|-|-|
> | abs(train loss - test loss)  | 0.4763  | 0.24022 | 0.3338 | 0.3035 | 0.2333 | 0.3107 | 0.2456 | 0.1393 |
> | abs(train acc - test acc)    | 8.332   | 6.518  | 7.532  | 7.608   | 6.511    | 7.708  | 7.128  | 4.278   |
>
> The experimental results show that the empirical generalization gap is highly consistent with the stability results reported in Appendix A.4. This further indicates that the observation aligns well with the theoretical conclusion in Theorem 3.5.

---

### Note · Authors · 2026-01-26

I have read and agree with the venue's withdrawal policy on behalf of myself and my co-authors.

---

### Meta-Review · Area_Chair_Uaeg · 2025-12-16

**Summary:**

The reviewers concur that the paper addressed the important stability topic in local optimization methods. However, the general mood of the reviewers is that the paper is not fully convincing since it requires additional assumptions and provides a weaker theory compared to the previous methods, such as SCAFFOLD.

The main paper's contribution is the new stability bound $O(1 - (1 - \Gamma)^T),$ which improves $O(\sqrt{T})$ of the previos methods (claimed near line 252). I would agree that this is an important result if it were more transparent. Unfortunately, the new results seem to be no better than $O(\sqrt{T}).$ Indeed, their full stability bound in Theorem 3.7 is
$$(A) := O\left(\frac{(1 - (1 - \Gamma)^T)}{\Gamma}\right),$$
where $\Gamma$ in the denominator can be as small as $1 / \sqrt{T}$. Note that
$$\Gamma = 1 - \left(\frac{\lambda}{\lambda + \mu} + \frac{\mu}{\lambda + \mu} (1 - \eta_{l} (\lambda + \mu))^K\right).$$
Since $\eta_{l} = O(1 / (L K \sqrt{T}))$ (Remark 4.2), for $T$ large enough,
$$\Gamma \approx 1 - \left(\frac{\lambda}{\lambda + \mu} + \frac{\mu}{\lambda + \mu} (1 - K \eta_{l} (\lambda + \mu))\right) \approx O(1 / \sqrt{T}),$$
where I use the classical result $(1 + x)^k = 1 + k x + o(x).$ Substituting this to $(A),$ one can see that the new stability bound is also at least $O(\sqrt{T}).$ Thus, the main contribution is not convincing and potentially might be incorrect. Considering this and the reviewers' weaknesses, I recommend rejection.

**Reviewer Concerns:**

The main weaknesses regarding the stronger assumptions are not fully addressed. The authors claim that the novelty lies in the better stability, which I believe is not true due to the above derivations.

**Reviewer Scores:**

I am not sure that the reviewers would change their score.

---

### Decision · Program_Chairs · 2026-01-26

Reject